# Vein fate determined by flow-based but time-delayed integration of network architecture

**Sophie Marbach**[1†], **Noah Ziethen**[2†], **Leonie Bastin**[2], **Felix K Bäuerle**[2], **Karen Alim**[2,3*]

[1]Courant Institute of Mathematical Sciences, New York University, New York, United States; [2]Max Planck Institute for Dynamics and Self-Organization, Göttingen, Germany; [3]Center for Protein Assemblies and Department of Bioscience, School of Natural Sciences, Technical University of Munich, Garching, Germany

**Abstract** Veins in vascular networks, such as in blood vasculature or leaf networks, continuously reorganize, grow or shrink, to minimize energy dissipation. Flow shear stress on vein walls has been set forth as the local driver for a vein's continuous adaptation. Yet, shear feedback alone cannot account for the observed diversity of vein dynamics – a puzzle made harder by scarce spatio-temporal data. Here, we resolve network-wide vein dynamics and shear rate during spontaneous reorganization in the prototypical vascular networks of *Physarum polycephalum*. Our experiments reveal a plethora of vein dynamics (stable, growing, shrinking) where the role of shear is ambiguous. Quantitative analysis of our data reveals that (a) shear rate indeed feeds back on vein radius, yet, with a time delay of 1–3 min. Further, we reconcile the experimentally observed disparate vein fates by developing a model for vein adaptation within a network and accounting for the observed time delay. The model reveals that (b) vein fate is determined by parameters – local pressure or relative vein resistance – which integrate the entire network's architecture, as they result from global conservation of fluid volume. Finally, we observe avalanches of network reorganization events that cause entire clusters of veins to vanish. Such avalanches are consistent with network architecture integrating parameters governing vein fate as vein connections continuously change. As the network architecture integrating parameters intrinsically arise from laminar fluid flow in veins, we expect our findings to play a role across flow-based vascular networks.

**\*For correspondence:**
k.alim@tum.de

†These authors contributed equally to this work

## Editor's evaluation

This fundamental work elucidates the physical forces that shape rearrangement of vascular networks using the model system slime mold. The authors provide compelling theoretical and experimental evidence to demonstrate how the fluid flow locally deforms the veins and ultimately dictates a global remodelling of network architecture. This profound and experimentally validated theory will be of great interest for many readers working on dynamically rearranging networks, which are ubiquitous in living systems.

## Introduction

Veins interwebbed in networks distribute resources across numerous forms of life, from the blood vasculature in animals (*Kurz, 2000*; *Hove et al., 2003*; *Chen et al., 2012*; *Zhou et al., 1999*), via the leaf venation in plants (*Corson et al., 2009*; *Ronellenfitsch and Katifori, 2016*) to the vein networks entirely making up fungi and slime molds (*Tero et al., 2010*; *Alim et al., 2013*). Continuous reorganization is integral to a network's success: veins perpetually grow and shrink (*Lucitti et al., 2007*; *Chen*

et al., 2012; *Hu and Cai, 2013*). While vein dynamics are usually observed for individual veins (*Kurz, 2000*), reorganization patterns at the network scale remain a puzzle. Yet, understanding network reorganization is crucial to shed light on the mechanics of development (*Chen et al., 2012*) and wide-spread diseases (*Meyer et al., 2008*; *Pries et al., 2009*).

While the biological makeup of vasculature systems is quite diverse, the physics that governs pervading and laminar fluid flows is the same (*Alim, 2018*). Already almost a century ago, Murray introduced the idea that shear stress exerted by fluid flows on a vein wall determines vein radius size (*Murray, 1926*). Within his framework, at steady state, veins minimize viscous dissipation while constrained by a constant metabolic cost to sustain the vein. Solving the minimization problem yields that shear stress, driver of viscous dissipation, should be constant among veins. Since Murray derived his hypothesis, studies have focused on *static* networks (*Price and Enquist, 2007*; *Ronellenfitsch and Katifori, 2016*; *Mentus and Roper, 2021*). Data on optimal static network morphologies agrees very well with Murray's predictions, strikingly across very different forms of life; from animals (*West et al., 1997*; *Kassab, 2006*), to plants (*West et al., 1997*; *McCulloh et al., 2003*) and slime molds (*Akita et al., 2017*; *Fricker et al., 2017*). Fluid flow physics is, therefore, key to understanding vascular morphologies.

Beyond steady state, during reorganization, how do flows shape network morphologies? Data on vein *dynamics* (*Chen et al., 2012*; *Baumgarten and Hauser, 2013*; *Rosenfeld et al., 2016*; *Chang and Roper, 2019*; *Sugden et al., 2017*), even during spontaneous reorganization, is limited due to the difficulty of acquiring time-resolved data covering entire networks. Observation of network excerpts suggests that flow shear stress alone can not account for the diversity of observed dynamics (*Chang et al., 2017*). In light of scarce experimental observations, a number of vein adaptation models have been introduced (*Hacking et al., 1996*; *Taber, 1998b*; *Taber, 1998a*; *Zhou et al., 1999*; *Pries et al., 2005*; *Baumgarten and Hauser, 2013*; *Hu and Cai, 2013*; *Secomb et al., 2013*; *Akita et al., 2017*; *Katifori et al., 2010*; *Hu et al., 2012*). Yet, the mechanisms that govern vein adaptation and thereby network reorganization can only be conclusively determined experimentally.

Here, we investigate the vascular networks formed by the slime mold *Physarum polycephalum*. Since the organisms' body is reduced to approximately two dimensions (*Baumgarten and Hauser, 2013*; *Alim et al., 2013*; *Fricker et al., 2017*), it opens up the unique possibility to quantify vein dynamics and fluid flows simultaneously in the entire network. From the fluid flows, we then quantify shear rate, directly related to shear stress by the inverse of the fluid's dynamic viscosity. Flows in the veins arise from rhythmic contractions of vein walls due to actomyosin activity in the vein cortex. As the flows oscillatory component changes rapidly on 1 min to 2 min (*Stewart and Stewart, 1959*; *Isenberg and Wohlfarth-Bottermann, 1976*), average flows dominate long-term vein adaptation dynamics on 10 min and more. Our aim, here, is to employ *P. polycephalum* to quantify experimentally and rationalize *individual* and *global* vein reorganization dynamics.

Our quantitative data reveals that shear rate indeed feeds back on vein radii, notably with a time delay. Furthermore, the effect of shear rate is disparate: similar shear rate values may cause veins either to grow or to shrink. To reconcile these disparate dynamics, we derive a model of vein adaptation in networks based on Kirchhoff's laws. Our model reproduces experimental observations and predicts that shear rate is not the only driver of vein adaptation, but also network-integrating parameters take control: fluid pressure and relative vein resistance. Both parameters integrate the network's architecture since they derive from fluid volume conservation on the network scale expressed by Kirchhoff's laws. As veins shrink and grow, network architecture continuously changes. As a consequence, a vein's fate to remain or shrink, is not predetermined by the current static network architecture but rather changes in time. This dynamic perspective explains avalanches of shrinking and disappearing veins in connected clusters. The mechanistic insight gained by our model suggests that the rules of vein reorganization, particularly the role of network-integrating parameters like fluid pressure and relative vein resistance, might be critical to understanding vascular networks across different life forms.

## Individual vein dynamics have complex shear rate-radius relation
### Quantifying vein dynamics

We observe vein dynamics in *P. polycephalum* specimen using two complementary imaging techniques, either close-up observation of single veins or full network imaging (*Figure 1* and additional methods in Appendix 1). Close-up vein microscopy over long timescales (*Figure 1A.i*, see also *Video 1* and *Video 2*)

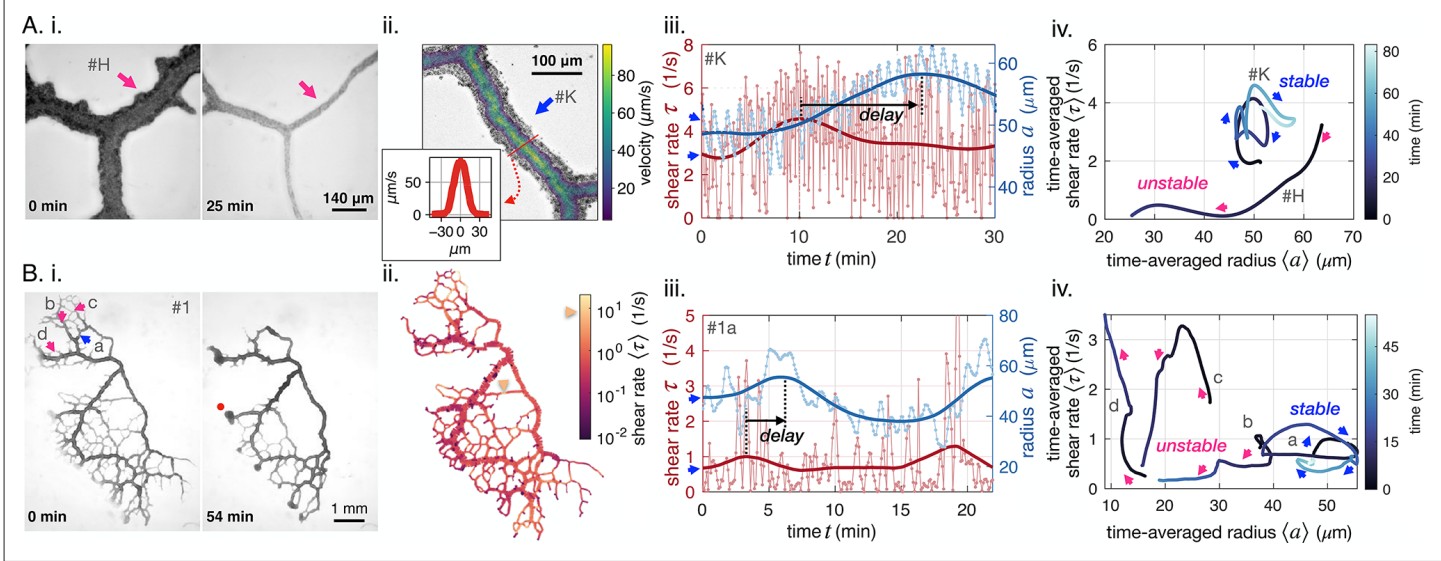

**Figure 1.** Diverse vein dynamics emerge during network reorganization. (**A**) Close-up and (**B**) full network analysis of vein radius dynamics and associated shear rate in *P. polycephalum*. (**i**) Bright-field images of reorganizing specimens allow us to record vein dynamics. (**ii**) Velocity measurements: (**A**) Velocity profiles along vein segments extracted with particle image velocimetry (inset: profile along vein cross-section) and (**B**) vein contractions driving internal flows over the entire network are integrated to calculate shear rate in veins (here shown at the initial observation time). The color scale indicates the magnitude of shear rate in each colored vein segment. For example, the yellow arrow points to a vein with a high calculated shear rate. (**iii**) Change in shear rate preceding changes in vein radius, both shown as a function of time (connected dots) and their time-averaged trends (full lines). (**A.iii**) shows the dynamics in the vein #K from (**A.ii**), (**B.iii**) shows the vein marked in blue in (**B.i**). (**iv**) The time-averaged shear rate versus the time-averaged radius displays circling dynamics for stable veins and diverse qualitative dynamics for unstable, vanishing veins. Blue color shades encode time. Trajectory arrow colors match arrow colors marking vein position in A.i (#H), A.ii (#K) and B.i, respectively. Veins marked in pink are shrinking, while stable veins are in blue.

allows us to directly measure radius dynamics $a(t)$ and velocity profiles $v(r, t)$ inside vein segments using particle image velocimetry (*Figure 1A.ii*), where $t$ is time and $r$ is the radial coordinate along the tube (all variable names are reported in *Appendix 1—table 1*). From velocity profiles, we extract the flow rate across a vein's cross-section $Q(t) = 2\pi \int v(r, t) r dr$. In full networks (*Figure 1B.i*, see also *Video 3*), radius dynamics $a(t)$ are measured for each vein segment and flow rates $Q(t)$ are subsequently calculated numerically integrating conservation of fluid volume via Kirchhoff laws, see Appendix 1.

Our imaging techniques resolve vein adaptation over a wide range of vein radii, $a = 5 - 70 \mu m$. Radii data show rhythmic peristaltic contractions, with a period of $T \simeq 1 - 2$ min (light blue in *Figure 1iii*). We calculate shear rate from fluid flows as $\tau = \frac{4}{\pi} \frac{|Q|}{a^3}$. Unlike shear stress, shear rate measurements do not require knowledge of the fluid's viscosity and are, therefore, more precise. Since both quantities are directly proportional, the conclusions we draw for shear rate apply to shear stress on the typical timescale of our experiments, where potential aging affects altering fluid viscosity can be neglected. We observe that shear rate $\tau$ oscillates with twice the contraction frequency (light red in *Figure 1iii*). In fact, since flows $Q$ reverse periodically, they oscillate around 0. In the shear rate $\tau$, oscillation periods are even doubled due to taking the absolute value of $Q$ in calculating $\tau$; see also *Appendix 1—figure 1*.

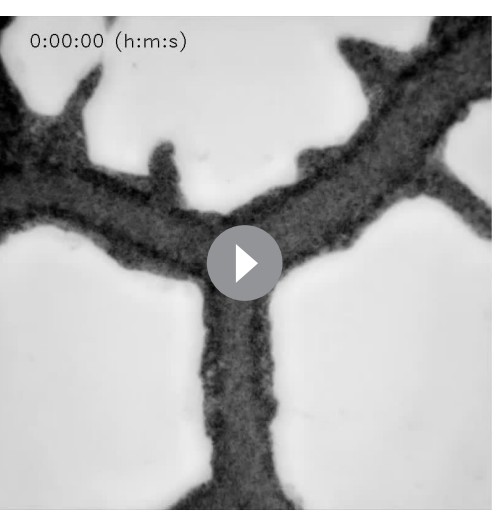

0:00:00 (h:m:s)

**Video 1.** Bright field stacked images of the close-up specimen including veins #H, #I and #J in *Figure 1A.i*. Frame sequence: 5 frames at 600 ms and 1 frame at 2.5 s. Scale 0.353 µm/pix.
https://elifesciences.org/articles/78100/figures#video1

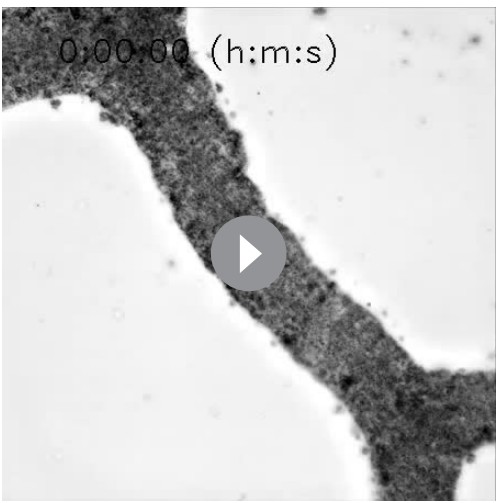

**Video 2.** Bright field stacked images of the close-up specimen including vein #K in *Figure 1A.ii*. Frame sequence: 5 frames at 600 ms and 1 frame at 5 s. Scale 0.25 μm/pix.

https://elifesciences.org/articles/78100/figures#video2

To access the long-time behavior of veins, we average out short timescales on the order of $T \simeq 1 - 2\,\mathrm{min}$ corresponding to the peristaltic contractions (*Isenberg and Wohlfarth-Bottermann, 1976*). We, thus, focus on the dynamics of the time-averaged radius $\langle a \rangle$ and shear rate $\langle \tau \rangle$ on longer timescales, from 10–60 min (full lines in *Figure 1iii*), corresponding to growth or disassembly of the vein wall, linked to *e.g* actin fiber rearrangements (*Salbreux et al., 2012*; *Fischer-Friedrich et al., 2016*).

## Diverse and reproducible vein dynamics

We relate time-averaged shear rate to time-averaged vein radius and find diverse, complex, yet reproducible trajectories (*Figure 1A, B.iv*, see also *Appendix 1—figure 4* and *Appendix 1—figure 5* for additional datasets). To illustrate this diversity, out of 200 randomly chosen veins in the full network of *Figure 1B*, we find 80 shrinking veins, 100 stable veins, and 20 are not classifiable.

In shrinking veins, the relation between shear rate and vein adaption is particularly ambiguous. As the radius of a vein shrinks, the shear rate either monotonically decreases (pink b in *Figure 1B.iv*), or, monotonically increases (pink d), or, increases at first and decreases again (pink c). For the specimen of *Figure 1B*, out of the 80 shrinking veins, monotonic decrease is observed for 25%, monotonic increase for 40%, and non-monotonic trajectories 15% of the time. The remaining 20% of vanishing veins are unclassifiable, as their recorded trajectories are too short to allow for any classification. Out of the 12 close-up veins investigated, 4 shrink and vanish, either with monotonic or non-monotonic dynamics (see also *Appendix 1—figure 2*).

In contrast, stable veins have a specific shear rate-radius relation: usually, stable veins perform looping trajectories in the shear rate-radius space (blue arrows in *Figure 1A, B.iv*). In the full network, these loops circle clockwise for 80% of 100 observed stable veins. Out of the 12 close-up veins investigated, 6 veins show stable clockwise feedback, 1 shows stable anticlockwise feedback, and 1 is not classifiable. Clockwise circling corresponds to an in/decrease in shear rate followed by an in/decrease in vein radius, thus, hinting at a shear rate feedback on local vein adaptation. This establishes a potential causality link between shear rate changes and vascular adaptation. In addition, the circular shape of stable vein trajectories suggests that there is a time delay between changes in shear rate and subsequent vein radius changes.

## Shear rate and resistance feedback alone can not account for the diversity of vein fates

We further test this potential causality link between shear rate and vein adaptation. Based on previous theoretical works (*Taber, 1998a*; *Hacking et al., 1996*; *Hu et al., 2012*; *Secomb et al., 2013*; *Pries et al., 1998*; *Pries et al., 2005*; *Hu and Cai, 2013*; *Tero et al., 2007*), we expect that the

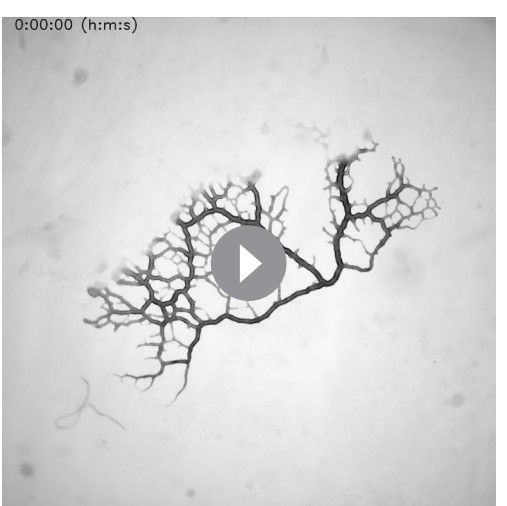

**Video 3.** Bright field stacked images of the full network specimen #1 in *Figure 1B.i*. Frame rate 6 s and scale 5.03 μm/pix.

https://elifesciences.org/articles/78100/figures#video3

magnitude of shear rate directly determines vein fate, that is lower shear rate results in a shrinking vein. Yet, this is not corroborated by our experimental measurements. First, despite displaying comparable shear rate and vein radii at the beginning of our data acquisition, some veins are stable (blue a in *Figure 1A, B.iv*), while others vanish (pink b). We, thus, map out shear rate throughout the entire network at the beginning of our observation, see *Figure 1B.ii*. We observe that dangling ends have low shear rate, due to flow arresting at the very end of the vein (dark purple terminal veins). Yet, some dangling ends will grow (i.e red dot in *Figure 1B.i*), in contradiction again with the assumption that 'low shear results in a shrinking vein'. Finally, small veins located in the middle of the organism show high shear rate, yet, will vanish (yellow arrow in *Figure 1B.ii*, other examples in *Appendix 1—figure 4C* and *Appendix 1—figure 5C*). Therefore, the hypothesis that veins with low shear rate should vanish, as they cannot sustain the mechanical effort (*Koller et al., 1993*; *Hoefer et al., 2013*), cannot be reconciled with our data.

Finally, also other purely geometrical vein characteristics such as vein resistance (*Baumgarten and Hauser, 2013*), $R = \frac{8\mu L}{\pi \langle a \rangle^4}$, where $\mu$ is the fluid viscosity and $L$ the vein length (*Happel and Brenner, 2012*), clearly do not determine vein fate either. In fact, geometrical vein characteristics are directly related to vein radius, thus in contradiction with our observation that veins with similar radius can experience different fates (stable blue a in *Figure 1iv* and vanishing pink b). Therefore, additional feedback parameters must play a role.

## Shear rate feedback on individual vein dynamics occurs with a time delay

The link between shear rate feedback and vein adaptation is clearly ambiguous in our data. To understand the feedback mechanism, we now turn to modeling and in-depth analysis.

### Vein radius adaptation in response to shear rate

Current theoretical models (*Hacking et al., 1996*; *Hu and Cai, 2013*; *Taber, 1998a*; *Ronellenfitsch and Katifori, 2016*) motivated by Murray's phenomenological rule of minimizing dissipation (*Murray, 1926*) suggest that vascular adaptation, $\frac{d\langle a \rangle}{dt}$, that is the change in time of the vein radius $\langle a \rangle$, is related to shear rate $\langle \tau \rangle$ via

$$\frac{d\langle a \rangle}{dt} = \frac{\langle a \rangle}{t_{\mathrm{adapt}}} \left( \frac{\tau_s^2(\langle \tau \rangle)}{\tau_0^2} - 1 \right). \tag{1}$$

Here, $\tau_s(\langle \tau \rangle)$ is the shear rate *sensed* by a vein wall and is directly related to fluid shear rate $\langle \tau \rangle$, in a way that we specify in the following paragraph. The parameter $t_{\mathrm{adapt}}$ is the adaptation time to grow or disassemble vein walls corresponding to fiber rearrangement (*Salbreux et al., 2012*; *Fischer-Friedrich et al., 2016*) and $\tau_0$ the vein's reference shear rate, corresponding to a steady state regime $\tau_s = \tau_0$ with constant shear rate – in agreement with Murray's law (see Appendix 2.1; *Murray, 1926*). $t_{\mathrm{adapt}}$ and $\tau_0$ are independent variables, constants over the timescale of a vein's adaptation, and could a priori vary from vein to vein, though existing models assume they do not (*Hacking et al., 1996*; *Hu and Cai, 2013*; *Taber, 1998a*; *Ronellenfitsch and Katifori, 2016*).

We here already incorporated two adaptations for our experimental system. First, we specifically indicate with $\frac{d\langle a \rangle}{dt}$ that we are interested in vascular adaptation, that is on long-time changes in the vein radius. In contrast, the short timescale variations $\frac{d(a - \langle a \rangle)}{dt}$ in *P. polycephalum* are driven by peristaltic contractions (*Isenberg and Wohlfarth-Bottermann, 1976*) and are not relevant for long-time adaptation. Second, we here, in contrast to all previous work, allow vein radii dynamics to potentially depend via a time delay on the shear rate, by describing radii dynamics as a function of a *sensed shear rate*, $\tau_s(\langle \tau \rangle)$, which itself depends on the average shear rate $\langle \tau \rangle$. We will specify this dependence in the section ''Model with a time delay quantitatively reproduces the data''.

Theoretical models differ in the precise functional dependence on shear rate on the right-hand side of *Equation 1*, but agree in all using a smooth function $f(\tau_s)$. We here employ a functional form with a quadratic scaling of the right-hand side on the shear rate $f(\tau_s) \propto \tau_s^2$ that we obtained via a bottom-up derivation from force balance on a vein wall segment in a companion work (*Marbach et al., 2023*). Within the force balance derivation, the cross-linked actin fiber cortex composing the

vein wall responds with a force in the normal direction compared to tangential shear and, hence, drives veins to dilate or shrink in response to shear (*Gardel et al., 2008*; *Janmey et al., 2007*; see Appendix 2.1). Experimental data measuring this anisotropic response of fibers in *Janmey et al., 2007*; *Vahabi et al., 2018*; *Kang et al., 2009* suggest a quadratic dependence of the change in fibers thickness on the applied shear. This quadratic dependence is also consistent with the top-down phenomenological result of *Hu et al., 2012*. That said, our upcoming results are robust against the specific choice of $f(\tau_s)$, as long as $f$ increases with $|\tau_s|$ and their exists a non-zero value of shear rate $\tau_0$ corresponding to Murray's steady-state, that is such that $f(\tau_0) = 0$.

Regarding the interpretation of the sensed shear rate $\tau_s$, it is apparent from our data that the link between shear rate and radius adaptation is not immediate but occurs with a time delay. *Figure 1iii* indeed shows lag times between peaks in time-averaged shear rate and radius dynamics, ranging from 1 min to 10 min. As a result, $\tau_s$ could correspond to a delayed shear rate compared to the actual one $\langle \tau \rangle$. We turn to confirm this assumption and analyze this time delay further.

## Statistical analysis of the time delay between shear rate and radius dynamics

We systematically investigate the time delay between shear rate $\langle \tau \rangle$ and vein adaptation $\frac{d\langle a \rangle}{dt}$. For each vein segment, we calculate the cross-correlation between averaged shear rate $\langle \tau \rangle(t - t_{\text{delay}})$ and vein adaptation $\frac{d\langle a \rangle}{dt}(t)$ as a function of the delay $t_{\text{delay}}$ (*Figure 2A*). Then, we record the value of $t_{\text{delay}}$ that corresponds to a maximum (*Figure 2B*). Time delays are recorded if the maximum is significant only, that is if the cross-correlation is high enough, and here we choose the threshold to be 0.5. Note, that slight changes in the threshold do not affect our results significantly. Both positive and negative time delays are recorded. Each full network data set contains more than 10,000 vein segments, which allows us to obtain statistically relevant data of $t_{\text{delay}}$ (*Figure 2C* and see also *Appendix 2—figure 2*). We present additional methods to extract the time delay also in close-up networks in Appendix 2. Note that $t_{\text{delay}}$ is different from $t_{\text{adapt}}$. Although both timescales are relevant to describe adaptation in our specimen: $t_{\text{delay}}$ represents the time to sense shear rate signals in vein walls; $t_{\text{adapt}}$ represents the time to grow or disassemble a vein wall.

Overall, we find 15 times more veins with positive time delays than with negative time delays for the specimen of *Figure 1B* (full time delay distribution in *Appendix 2—figure 2*). This clearly establishes a causality link between shear rate magnitude and radius adaptation. We also find that time delays of 1 to 3 min are quite common with an average of $t_{\text{delay}} \simeq 2\,\text{min}$ (*Figure 2C*). We repeat the analysis over different full network specimens (*Appendix 2—figure 2*) and close-up veins (*Appendix 2—figure 3*) and find similar results.

While unraveling the exact biophysical origin of the time delay is beyond the scope of this work, it is important to discuss potential mechanisms. First, the typical time delay measured $t_{\text{delay}} \simeq 2\,\text{min}$ appears close to the contraction period $T \simeq 1 - 2\,\text{min}$. This is not an artifact of the analysis (see benchmark test in Appendix 2). Rather, it hints that the cross-linked actomyosin and contractile cortex are key players in the delay. Measured data on the contractile response of cross-linked fibers (*Gardel et al., 2008*; *Janmey et al., 2007*; *Vahabi et al., 2018*) exhibits a time delay of about 1–30 s for in vitro gels. This time delay could accumulate in much longer time delays in vivo (*Armon et al., 2018*), as is the case in our sample, and potentially reach a time delay of about $2\,\text{min}$. Other mechanical delays could originate from the cross-linked actomyosin gel. For example, the turnover time for actin filaments in living cells ranges from 10 s to 30 s (*Fritzsche et al., 2013*; *Livne et al., 2014*; *Colombelli et al., 2009*), while the viscoelastic relaxation time is 100 s (*Joanny and Prost, 2009*), both timescales close to our measured time delay.

## Model with a time delay quantitatively reproduces the data

Having clearly established the existence of a positive time delay for shear rate feedback on vein adaptation, we must radically deviate from existing models (*Hacking et al., 1996*; *Taber, 1998b*; *Taber, 1998a*; *Pries et al., 2005*; *Secomb et al., 2013*) by incorporating the measured time delay $t_{\text{delay}}$ explicitly between the shear rate sensed by a vein wall $\tau_s$ and fluid shear rate $\langle \tau \rangle$. To this end, we use the phenomenological first-order equation

$$\frac{d\tau_s}{dt} = -\frac{1}{t_{\text{delay}}}(\tau_s - \langle \tau \rangle) \tag{2}$$

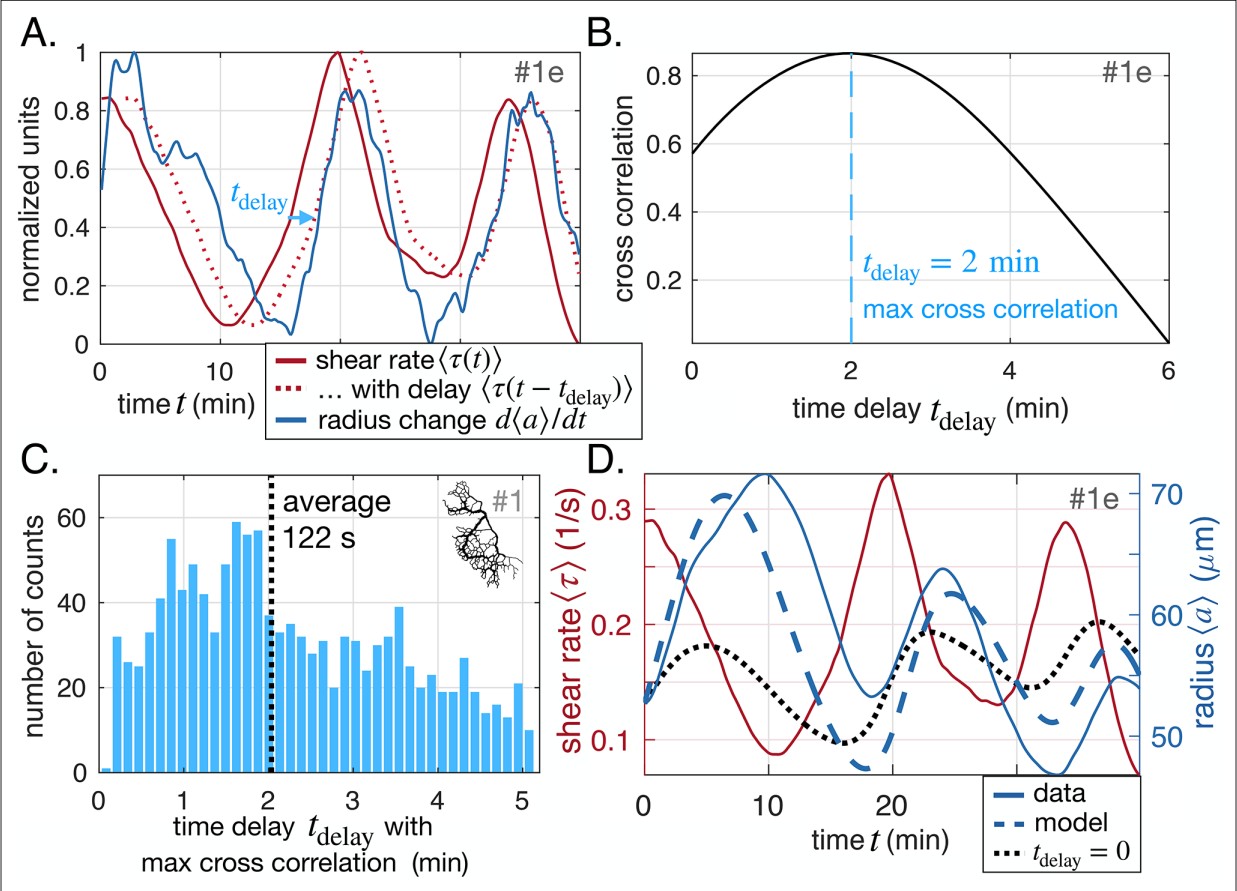

**Figure 2.** Shear rate induces vein adaptation with a time delay. (**A**) Principle of the cross-correlation in time between the delayed time-averaged shear rate $\langle \tau(t - t_{\text{delay}}) \rangle$ and the time-averaged radius change $\frac{d\langle a \rangle}{dt}(t)$. The plot shows the delayed curve with the best score. (**B**) Resulting cross-correlation for various values of $t_{\text{delay}}$ and maximum extracted. (**C**) Statistical analysis over all veins of specimen #1 of maximal cross-correlation with $t_{\text{delay}}$, zoomed in on positive time delays, since they outnumber negative time delays by a factor of 15, see **Appendix 2—figure 2** for full distribution. (**D**) Fit result of the model **Equation 1** and **Equation 2**, here, with $t_{\text{adapt}} = 37 \pm 2$ min and $\tau_0 = 1.1 \pm 0.2$ s$^{-1}$ (heavy dashed blue). The vein investigated is the same as in (**A–B**) and hence we took $t_{\text{delay}} = 2$ min. The relative fitting error is $\epsilon_{\text{err}} = 0.07$. The fit result of model **Equation 1** taking $\tau_s = \langle \tau \rangle$, i.e. with $t_{\text{delay}} = 0$ is also shown (heavy dotted black), and has error $\epsilon_{\text{err}} = 0.11$.

At steady state, we recover a constant shear rate $\langle \tau \rangle = \tau_s = \tau_0$, corresponding to Murray's law (Appendix 2) (**Murray, 1926**).

We further verify that our model with the adaptation rule **Equation 1** and the time delay shear rate sensing **Equation 2** quantitatively accounts for the observed dynamics with physiologically relevant parameters. We fit our 12 close-up data sets, as well as 15 randomly chosen veins of the full network in **Figure 1B**. We take shear rate data $\langle \tau \rangle(t)$ as input and fit model constants $t_{\text{adapt}}$ and $\tau_0$ to reproduce radius data $\langle a \rangle(t)$. Note, that $t_{\text{adapt}}$ and $\tau_0$ are independent variables that vary from vein to vein, and over long timescales and between specimen (**Swaminathan et al., 1997**; **Puchkov, 2013**; **Fessel et al., 2017**; **Lewis et al., 2015**; **Marbach et al., 2023**). To test the robustness of model fits, we employ different strategies to set the time delay $t_{\text{delay}}$ before fitting. The time delay is either set to the same average value for all veins, or to the best cross-correlation value for a specific vein, or fitted with a different value for each vein, with no significant change in the resulting goodness of fit and fit parameter values.

Overall, we find a remarkable agreement between fit and data (see example in **Figure 2D** and Appendix 2 for more results). We find a small relative error on fitted results $\epsilon_{\text{err}} = \int dt \frac{|\langle a \rangle - \langle a \rangle^{\text{fit}}|}{\langle a \rangle} \simeq 0.001 - 0.17$. This suggests that the minimal ingredients of this model are sufficient to reproduce experimental data. Fits without the time delay yield systematically worse results, with larger fitting errors $\epsilon_{\text{err}}$ (see **Figure 2D**, dotted black line and **Appendix 2—table 3**).

In all samples, fitting parameters resulted in physically reasonable values. We found $t_{\text{adapt}} \simeq 10 - 100$ min corresponding to long timescale adaptation of vein radii. Note again, the physical difference between the time to adapt vein radius $t_{\text{adapt}}$ and the time delay to sense shear rate $t_{\text{delay}}$ also translates to orders of magnitude differences with $t_{\text{delay}} \simeq 2$ min and $t_{\text{adapt}} \simeq 10 - 100$ min. This 10–100 min is indeed the timescale over which we observe significant adaptation. Reorganization of biological matter occurs on similar timescales in other comparable systems, from 15 min for individual cells to several days for blood vasculature (*Livne et al., 2014*; *Landau et al., 2018*).

When examining fit results of the target shear rate $\tau_0$ it is a priori hard to estimate which values to expect since $\tau_0$ is only reached at steady state. Yet, in our continuously evolving specimen, we never reach steady state and, hence, can not measure $\tau_0$. However, we can compare $\tau_0$ to shear rate values measured in our specimen and find that they are consistently of the same order of magnitude. Finally, we find that our model yields better results if we fit the data over intermediate time frames (15 min to 40 min), exceeding results of fitting over longer time frames (40 min to 100 min). This is in line with our theoretical expectation (*Marbach et al., 2023*) that $t_{\text{adapt}}$ and $\tau_0$ change over long timescales, since they depend on physical parameters that also change over long timescales, in particular in response to network architecture changes. Since veins typically vanish over 15 min to 40 min and, hence, significant network changes occur over exactly that timescale, $t_{\text{adapt}}$ and $\tau_0$ are no longer constant for time frames $\gtrsim 40$ min.

While we have focused so far on timescales of individual vein adaptation, we now aim to understand how their individual disparate fates arise. We will show that the origin of different fates resides in the evolution of the rest of the network.

## Relative resistance and pressure determine vein fate within a network

### Stable and unstable vein dynamics are predicted within the same model

To capture the impact of the entire network on the dynamics of a single vein modeled by *Equations 1; 2*, we must specify the flow-driven shear rate $\langle \tau \rangle$. Since $\langle \tau \rangle = \frac{4|Q|}{\pi \langle a \rangle^3}$, it is sufficient to specify the flow rate $Q$ in a vein. $Q$ is coupled to the flows throughout the network by conservation of fluid volume through Kirchhoff's laws, and is, therefore, an indirect measure of network architecture.

We, here, consider the most common vein topology of a vein connected at both ends to the remaining network, more specialized topologies follow in ''Specific vein fates''. The network is then represented by a vein of equivalent resistance $R_{\text{net}}$ parallel to the single vein of $R = \frac{8\mu L}{\pi \langle a \rangle^4}$ considered within an equivalent flow circuit, see *Figure 3A*. $R_{\text{net}}$ is the equivalent resistance corresponding to all the resistances making up the rest of the network, obtained with Kirchhoff's laws (see examples in Appendix 3). $R_{\text{net}}$ is therefore integrating the network's architecture. Such a reduction of a flow network to a simple equivalent flow circuit is always possible due to Norton's theorem (*Morris, 1978*).

The time-averaged net flow generated by the vein contractions is $Q_{\text{in}} = \left\langle \left| L \frac{d(\pi a^2)}{dt} \right| \right\rangle \simeq \frac{8\pi L \epsilon \langle a \rangle^2(t)}{T}$ where $\epsilon$ is the relative contraction amplitude. The absolute values in this definition are used to measure the *net* flow. $Q_{\text{in}}$ thus measures the mass exchanges between the network and the vein. As mass is conserved, this results in an inflow of $Q_{\text{net}} = -Q_{\text{in}}$, into the rest of the network. Within the vein, a total flow rate $Q$ circulates – see *Figure 3A.ii*. The flow rate $Q$ through the vein follows from Kirchhoff's second law: $QR = -(Q + Q_{\text{net}})R_{\text{net}}$. We, thus, obtain that the time-averaged shear rate in the vein is

$$\langle \tau \rangle \simeq \frac{4|Q|}{\pi \langle a \rangle^3}(t) = \frac{4Q_{\text{in}}(\langle a \rangle)}{\pi \langle a \rangle^3} \frac{1}{1 + R(\langle a \rangle)/R_{\text{net}}}. \tag{3}$$

The coupled dynamics of $\{\langle \tau \rangle, \tau_s, \langle a \rangle\}$ are now fully specified through *Equations 1–3*. To simplify our analysis, we now explore the reduced system $\{\tau_s, \langle a \rangle\}$ by replacing $\langle \tau \rangle$ in *Equation 2* by its expression in *Equation 3*. Using standard tools of dynamical systems theory, see Appendix 3.3, we now characterize the typical trajectories predicted within the model.

Our dynamic system $\{\tau_s, \langle a \rangle\}$ reproduces the key features of the trajectories observed experimentally. We find two stable fixed points at $(0, 0)$ and $(\tau_0, \langle a \rangle_{\text{stable}}(R/R_{\text{net}}, \tau_0))$, and one unstable fixed point at $(\tau_0, \langle a \rangle_{\text{unstable}}(R/R_{\text{net}}, \tau_0))$ (see *Figure 3B*). The stable fixed point with finite radius, $(\tau_0, \langle a \rangle_{\text{stable}})$

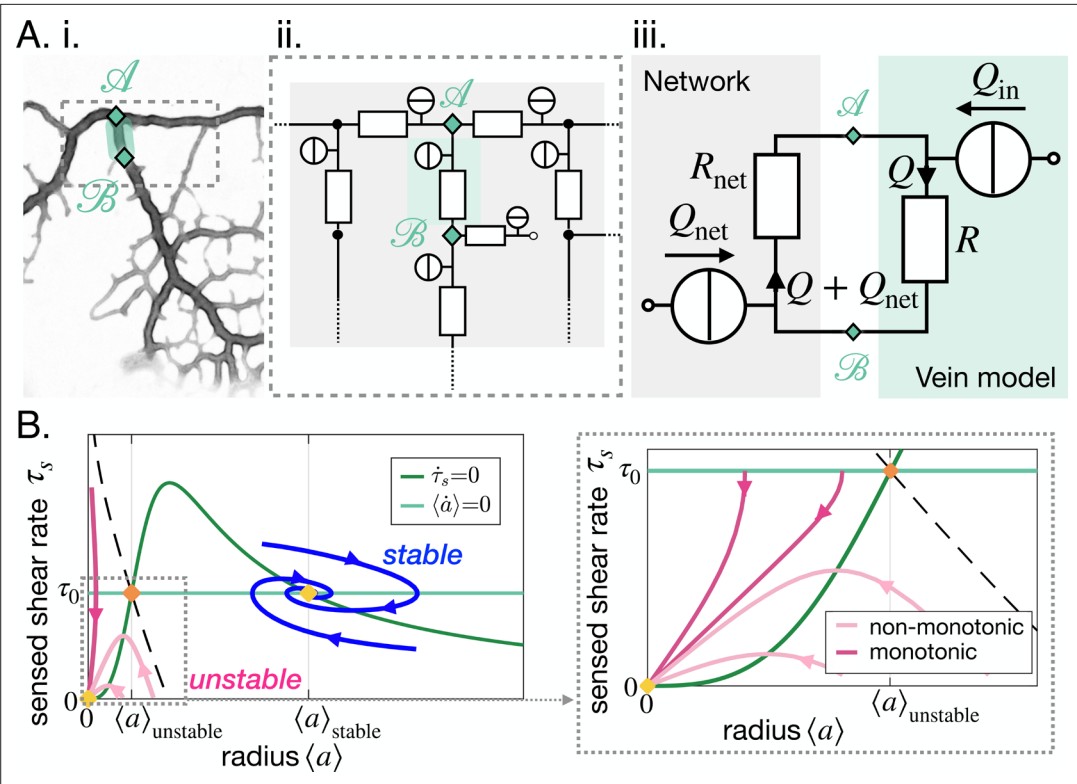

**Figure 3.** Stable and unstable vein dynamics are predicted within the same model. (**A**) Translation of (**i**) a bright field image of specimen into (**ii**) vein networks; each vein is modeled as a flow circuit link. (**iii**) Reduction of (**ii**) via Northon's theorem into an equivalent and simplified vein flow circuit consisting of a flow source $Q_{in}$ (due to vein's pumping) and a resistor $R$ (viscous friction). The rest of the network is modeled by an equivalent circuit with flow source $Q_{net} = -Q_{in}$ and resistor $R_{net}$. $Q$ flows through the vein. (**B**) (Left) Time-averaged sensed shear rate $\tau_s$ versus radius from (1)-(3) with fixed points and typical trajectories. The green lines correspond to stationary solutions for $\tau_s$ or $\langle a \rangle$. The blue lines correspond to stable trajectories and the pink lines to unstable ones. (Right) Zoom of the phase space corresponding to shrinking veins, including monotonic and non-monotonic trajectories.

corresponds to Murray's steady state. The set of fixed points was also found in a related theoretical study that investigates a phenomenological model resembling *Equation 1*, yet without any time delay, and examining the stability of a vein, or resistance, connected to a pressure source and another resistance (*Hacking et al., 1996*). This suggests that the presence of the three fixed points is universal. Furthermore, we find similar dynamical trajectories in the $\{\tau_s, \langle a \rangle\}$ as those observed experimentally. Trajectories spiral in the clockwise direction near the stable fixed point $(\tau_0, \langle a \rangle_{stable})$ (blue in *Figure 3B*) and veins shrink with monotonic (dark pink in *Figure 3B*) or with non-monotonic shear rate decrease (light pink *Figure 3B*). The dynamics of $\langle \tau \rangle$ are then closely related to that of $\tau_s$.

## Relative resistance and pressure control vein fate

Analysis of the vein network model as a dynamic system, *Equations 1–3*, clearly highlights that different vein fates may occur depending on the value of the relative resistance $R/R_{net}$ and on the value of the target shear rate $\tau_0$ for that specific vein. We will, therefore, now investigate their values throughout the network more carefully.

Before proceeding, we must specify the meaning of the target shear rate $\tau_0$. The force balance derivation in *Marbach et al., 2023* finds that the shear rate reference $\tau_0$ is related to the local fluid pressure $P$, as $\tau_0 \sim \tau_{active} - \langle P - P_0 \rangle / \mu$ (see short derivation in Appendix 2). Here, $P - P_0$ characterizes the pressure imbalance between the fluid pressure inside the vein, $P$, and the pressure outside, $P_0$, namely the atmospheric pressure. We recall that $\mu$ is the fluid viscosity. Finally, $\tau_{active} = \sigma_{active}/\mu$ is a shear rate related to the active stress $\sigma_{active}$ generated by the actomyosin cortex (*Radszuweit et al., 2013*; *Alonso et al., 2017*). The active stress sustains the contractile activity of the vein, and is,

therefore, an indirect measure of the metabolic or energetic consumption in the vein. The local pressure $P$ results from solving Kirchhoff's law throughout the network. It is, therefore, indirectly integrating the entire network's morphology. Hence, not only $R/R_{net}$ but also $\tau_0$ is a flow-based parameter, integrating network architecture.

In our experimental full network samples, we can calculate both the relative resistance $R/R_{net}$ and the local pressure $P$, and its short time-averaged counterpart $\langle P \rangle$, up to an additive constant (see *Figure 4*). We find that pressure maps of $\langle P \rangle$ are mostly uniform, except towards dangling ends where relevant differences are observed (*Figure 4A*). Hence, particularly in dangling ends, veins with similar shear rate $\tau$ may suffer different fates, as described through *Equation 1*. This is a radical shift compared to previous theoretical works which consider that $\tau_0$ is a constant throughout the network (*Taber, 1998a*; *Hacking et al., 1996*; *Hu et al., 2012*; *Secomb et al., 2013*; *Pries et al., 1998*; *Pries et al., 2005*; *Hu and Cai, 2013*; *Tero et al., 2007*).

The relative resistance $R/R_{net}$ varies over orders of magnitude (*Figure 4B*), with values that are not correlated with vein size (see *Appendix 1—figure 6*). Rather, $R/R_{net}$ indicates how a vein is localized within the network compared to large veins that have lower flow resistance and that serve as highways for transport. For example, a small vein immediately connected to a highway will show a large value of $R/R_{net}$. In this case among all possible flow paths that connect the vein's endpoints, there exists a flow path that consists only of highways, and therefore we expect $R \gg R_{net}$ (see red arrow in *Figure 4B*). In contrast, a similarly small vein yet localized in between other small veins, further away from highways, will show a smaller value of $R/R_{net}$. In this latter case, all flow paths have to pass through small nearby veins and, hence, have high resistance $R_{net} \gg R$ (see blue arrow in *Figure 4B*). $R/R_{net}$, therefore, reflects the relative cost to transport fluid through an individual vein rather than through the rest of the network.

The relative resistance $R/R_{net}$ is, thus, a natural candidate to account for individual vein adaptation: it measures the energy dissipated by flowing fluid through an individual vein, $Q^2 R/2$, compared to rerouting this flow through the rest of the network, $Q^2 R_{net}/2$. Hence, we may expect that when in

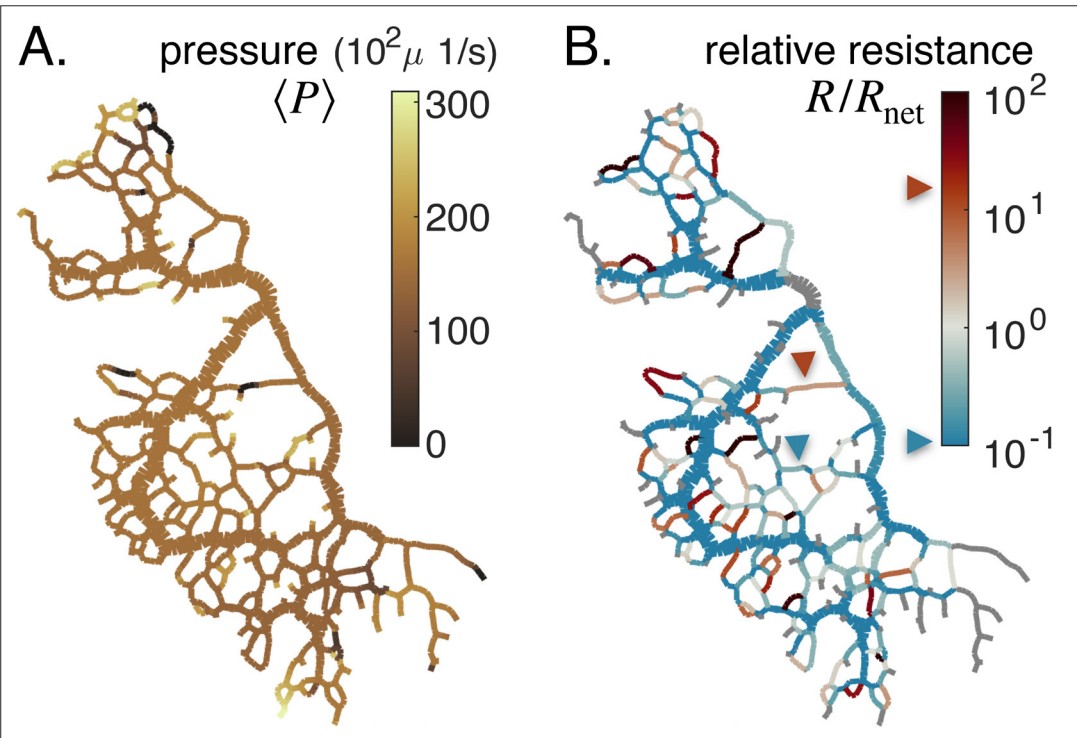

**Figure 4.** Feedback parameters integrate the network's architecture and provide information on vein relative location. Full network maps of the same specimen as in *Figure 1B*, at the beginning of the observation, of (**A**) the average fluid pressure in a vein $\langle P \rangle$ and (**B**) of the relative resistance $R/R_{net}$. The fluid pressure $\langle P \rangle$ is defined up to an additive constant. Grey veins in (**B**) correspond to bottleneck veins or dangling ends for which $R_{net}$ can not be defined. The color scales indicate the magnitude of each variable in each colored vein. For example, in (**B**), the red arrow indicates a vein with large relative resistance $R/R_{net}$.

a given vein $R > R_{net}$, it is energetically more favorable to flow fluid through the rest of the network and hence to shrink the vein. Reciprocally, if $R < R_{net}$, we expect that the vein is stable. Analyzing our equations gives further support to this intuitive rule. When $R \gg R_{net}$, from *Equation 3*, we may expect $\langle \tau \rangle$ to be relatively small, in particular, small relative to the vein's specific steady state $\tau_0$ and hence via *Equation 1* the vein would likely shrink. Reciprocally, if $R \ll R_{net}$, we may expect $\langle \tau \rangle$ to be relatively large compared to its specific $\tau_0$, and hence the vein is stable. Yet, since $R/R_{net}$ is nondimensional, it can provide more systematic insight than $\langle \tau \rangle$, since $\tau_0$ is not known a priori. Notice that the red arrow in *Figure 4B* presents a shrinking vein that indeed verifies $R > R_{net}$. However, according to shear rate measures (see yellow arrow in *Figure 1B.ii*), the shear rate is large in that vein, preconditioning the vein to grow, according to previous works (*Taber, 1998a*; *Hacking et al., 1996*; *Hu et al., 2012*; *Secomb et al., 2013*; *Pries et al., 1998*; *Pries et al., 2005*; *Hu and Cai, 2013*). We can therefore show why occasionally, veins at high shear rate shrink, and veins at low shear rate grow by highlighting that $R/R_{net}$, beyond shear rate, is crucial to predict vein fate.

Our aim is now to investigate, in more detail, how these novel feedback parameters integrating network architecture, the relative resistance $R/R_{net}$ and the local pressure $P$ via the target shear rate, control vein dynamics on the basis of three key network topologies of a vein.

## Specific vein fates: Dangling ends, parallel veins, and loops
### Dangling ends are unstable: Disappearing or growing

As observed in our data, dangling ends are typical examples of veins that can start with very similar shear rate and radius and yet suffer radically different fates (*Figure 1B.i, ii*, *Figure 5A*). Dangling ends either vanish or grow but never show stably oscillating trajectories.

Topologically, and unlike the middle vein considered in *Figure 3A*, dangling ends are only connected to the rest of the network by a single node. Therefore, the relative resistance $R_{net}$ cannot be calculated in a dangling end and cannot play a role. The shear rate in a dangling end is simply $\langle \tau \rangle = \frac{4\langle |Q_{in}| \rangle}{\pi \langle a \rangle^3} \simeq \frac{32L\epsilon}{\langle a \rangle T}$. Using this expression instead of *Equation 3* and analyzing the dynamical system with *Equations 1; 2*, we find that dangling veins can only shrink or grow (see Appendix 4). Furthermore, $\tau_0$ determines the threshold for growth over shrinkage. Since $\tau_0 \sim \tau_{active} - \langle P - P_0 \rangle/\mu$ a large $\langle P \rangle$ decreases $\tau_0$. Hence, the model predicts that a larger pressure at a dangling end facilitates growth.

We observe for the example of *Figure 5A* that large values of $\langle P \rangle$ indeed appear to favor growth, and small values prompt veins to vanish. This agrees with physical intuition: when a vein is connected to a large input pressure, one expects the vein to open up. Notice, however, that here the mechanism is subtle. The shear rate itself is not large. Rather, the shear rate threshold to grow is lowered by the high local pressure. Local pressure is thus connected to dangling end fate: it is a prime example of the importance of *integrating network architecture*.

### Competition between parallel veins decided by relative resistance

Parallel veins are another example in which initially very similar and spatially close veins may suffer opposite fates; see *Figure 5B*. Often, both parallel veins will eventually vanish, yet what determines which vanishes first?

To investigate this situation we can simply extend the circuit model of *Figure 3A* with another parallel resistance, corresponding to the parallel vein (Appendix 4). We then have two veins with respective resistance say $R_1$ and $R_2$. We can analyze the stability of this circuit with similar tools as above. We find that if one vein's relative resistance is larger than the other one's, say for example $R_1/R_{net,1} > R_2/R_{net,2}$, then vein 1 vanishes in favor of the other vein 2 as previously predicted in simpler scenarios for steady states (*Hacking et al., 1996*). Exploring $R/R_{net}$ in our full network (*Figure 5B*), we find that a vein with a large relative resistance $R/R_{net} > 1$ will vanish. In contrast, a nearby, nearly parallel vein with $R/R_{net} \simeq 1$ will remain stable.

The relative resistance $R/R_{net}$ is thus a robust predictor for locally competing veins. Although it is connected to shear rate, as highlighted through *Equation 3*, there are clear advantages to the investigation of $R/R_{net}$ over the shear rate itself: $R/R_{net}$ is straightforward to compute from global network architecture as it does not require to resolve flows, and it is non-dimensional.

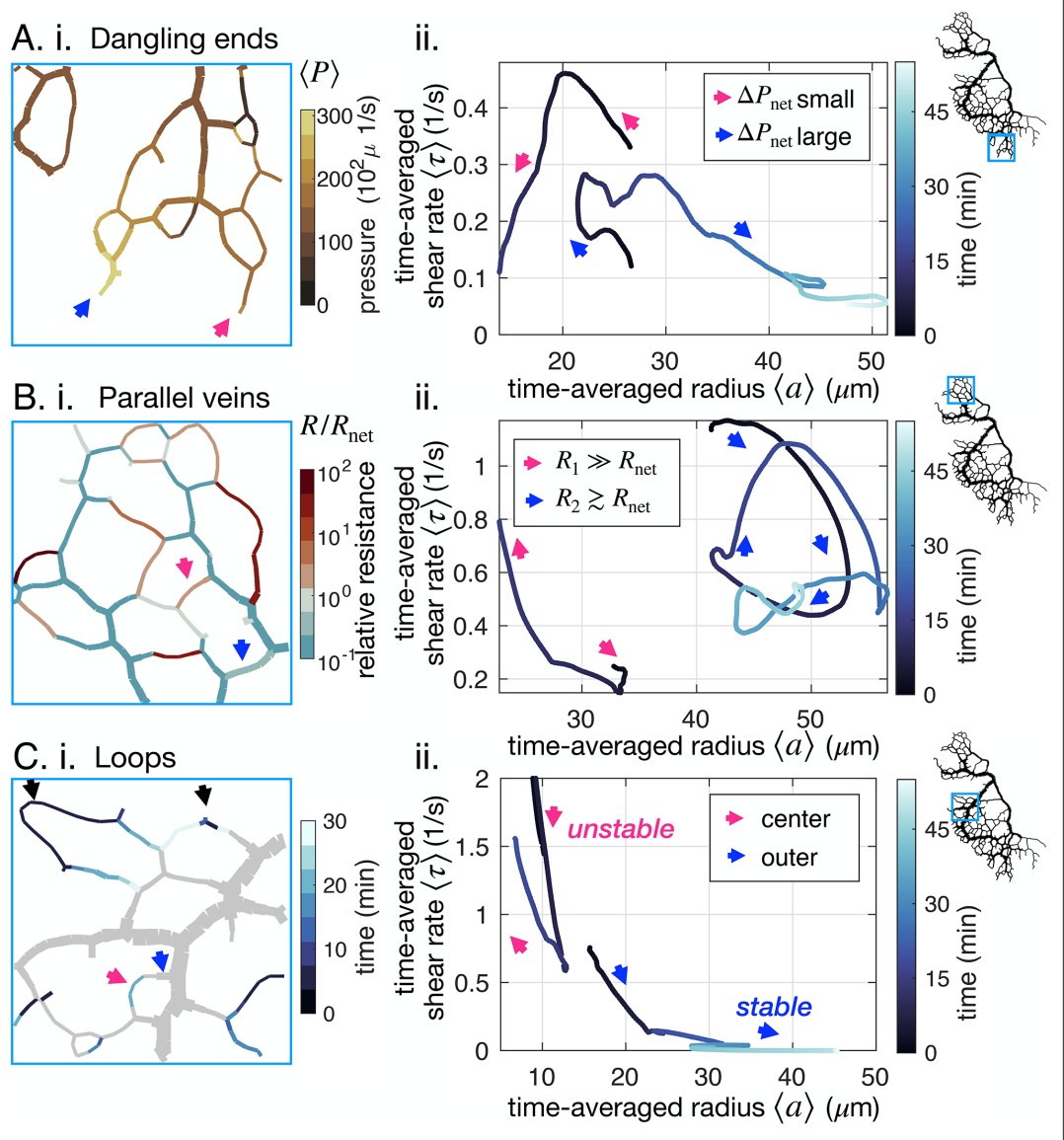

**Figure 5.** Network architecture controls vein fate as exemplified in three cases. (**A–C**) (**i**) determining factors mapped out from experimental data for the specimen of *Figure 1B* and (ii) typical trajectories from data. All pink (respectively blue) arrows indicate shrinking (respectively stable or growing) veins. (**A**) (ii) Dangling ends either vanish or grow indefinitely, coherently with (**i**) the relative local pressure $\langle P \rangle$. Arrows point to veins initially similar in size (~23 μm). (**B**) Parallel veins are unstable: one vanishes in favor of the other one remaining (ii), coherently with (**i**) its relative resistance, $R/R_\text{net}$, being higher. (**C**) Loops first shrink in the center of the loop (ii) – that is from the point furthest away from the nodes connecting it to the rest of the network – as evidenced by focusing on (**i**) the time of vein segment vanishing. Black arrows point to other loops also vanishing from the center. For all graphs on the left, the color scales indicate the magnitude of each variable in each colored vein.

## Loops shrink first in the middle

Finally, loopy structures *i.e.* a long vein connected at both ends to the remaining network, are often observed in *P. polycephalum*. Surprisingly, we experimentally observe loops to start shrinking in their very middle (*Figure 5C*, *Appendix 1—figure 4F* and *Appendix 1—figure 5F*) despite the almost homogeneous vein diameter and shear rate along the entire loop. This is all the more surprising as quantities such as $\langle P \rangle$ and $R/R_\text{net}$ are also similar along the loop.

This phenomenon again resides in the network architecture, and we can rationalize it with an equivalent flow circuit (see Appendix 4). When a vein segment in the loop shrinks, mass has to be

redistributed to the rest of the network. This increases shear rate in the outer segments, preventing the disappearance of the outer segments of the loop. Once the center segment has disappeared, both outer segments follow the dynamics of dangling ends. Their fate is again determined by network architecture, through the local pressure $\langle P \rangle$ in particular.

Importantly, we find that as soon as a vein disappears, the network's architecture changes: flows must redistribute, and vein connections are updated. Hence, an initially stable vein may become unstable. Vein fates, thus, dramatically evolve over time.

## Single vanishing vein triggers an avalanche of vanishing events among neighboring veins

After focusing on individual vein dynamics, we now address global network reorganization. Observing a disappearing network region over time, we find that veins vanish sequentially in time (**Figure 6A, B**). Inspired by the importance of relative resistance for parallel veins, we here map out relative resistance $R/R_{net}$ at subsequent time points in an entire region (**Figure 6A**). At the initial stage (**Figure 6A**, 2 min), the majority of veins are predicted to be stable with a relative resistance $R/R_{net} < 1$. As expected, the few veins with high relative resistance (red arrows in **Figure 6A**, 2 min) indeed vanish first (black crosses in **Figure 6A**, 5 min).

As a consequence of veins vanishing, the local architecture is altered, and the relative resistance, through $R_{net}$, changes drastically. Veins that were stable before are now predicted to be unstable. This avalanche-like pattern, in which individual vanishing veins cause neighboring veins to become unstable, repeats itself until the entire region disappears in less than 15 min (**Appendix 1—figure 4F** and **Appendix 1—figure 5F** show similar avalanches in other specimens). Note that a vanishing vein may rarely also stabilize a previously unstable vein (**Figure 6A**, 16 min, blue arrow).

The fundamental origin of these avalanches of vanishing veins can be narrowed down again to network architecture. We explore a model network region, inspired by a region in an actual specimen (**Figure 6C**). We simplify the investigation by considering the region is made of a few veins of similar resistance $r$ connected to the rest of a network, represented by an overall equivalent resistance $R_{rest}$. $R_{rest}$ represents the rest of the network relative to the region, distinct from $R_{net}$, which is relative to a single vein. We precondition all veins to be stable, assuming that for each vein its relative resistance $R/R_{net} \lesssim 1$. Since in our model network for each vein, we approximately have $R/R_{net} \sim r/R_{rest}$ this prescribes the initial values of $r/R_{rest} \lesssim 1$.

We now perturb a vein slightly, for example with a smaller radius, and therefore with a slightly higher resistance, say $2r$ (purple in **Figure 6C**). The perturbed vein's relative resistance thus may become greater than 1, making the vein unstable. As the vein vanishes, two network nodes are removed, and individual veins previously connected through the node now become a single *longer* vein. A longer vein has a higher hydraulic resistance. Hence, the 'new' longer vein also becomes unstable (blue in **Figure 6C**). Once it vanishes, in turn, another neighboring vein becomes longer and unstable (green in **Figure 6C**). Reciprocally, vein growth and parallel vein disappearance can – more rarely – decrease $R/R_{net}$, and in turn, stabilize a growing vein, as highlighted by the blue arrow in **Figure 6A** at 16 min.

In our simple mechanistic model, the series of events follows an avalanche principle similar to that observed in our experiments: a vanishing vein disturbs local architecture. This modifies the relative resistance of nearby veins and hence their stability. The avalanche of disappearing veins eventually results in the removal of entire network regions.

## Discussion

We here report highly resolved data of spontaneous network reorganization in *P. polycephalum* in which both individual vein dynamics and fluid flows pervading veins are quantified simultaneously. We observe disparate vein dynamics originating from shear-driven feedback on vein size. Strikingly, shear-driven feedback occurs with a time delay ranging from 1 min to 3 min. Our vein network model challenges previous concepts showing that vein fate is not only determined through shear rate magnitude but also through parameters that integrate network architecture via fluid flow. In particular, dangling end fate is connected to local fluid pressure $\langle P \rangle$, with larger pressures stabilizing dangling ends. Inner network vein fate is tightly determined by the vein's resistance relative to the resistance to fluid flow

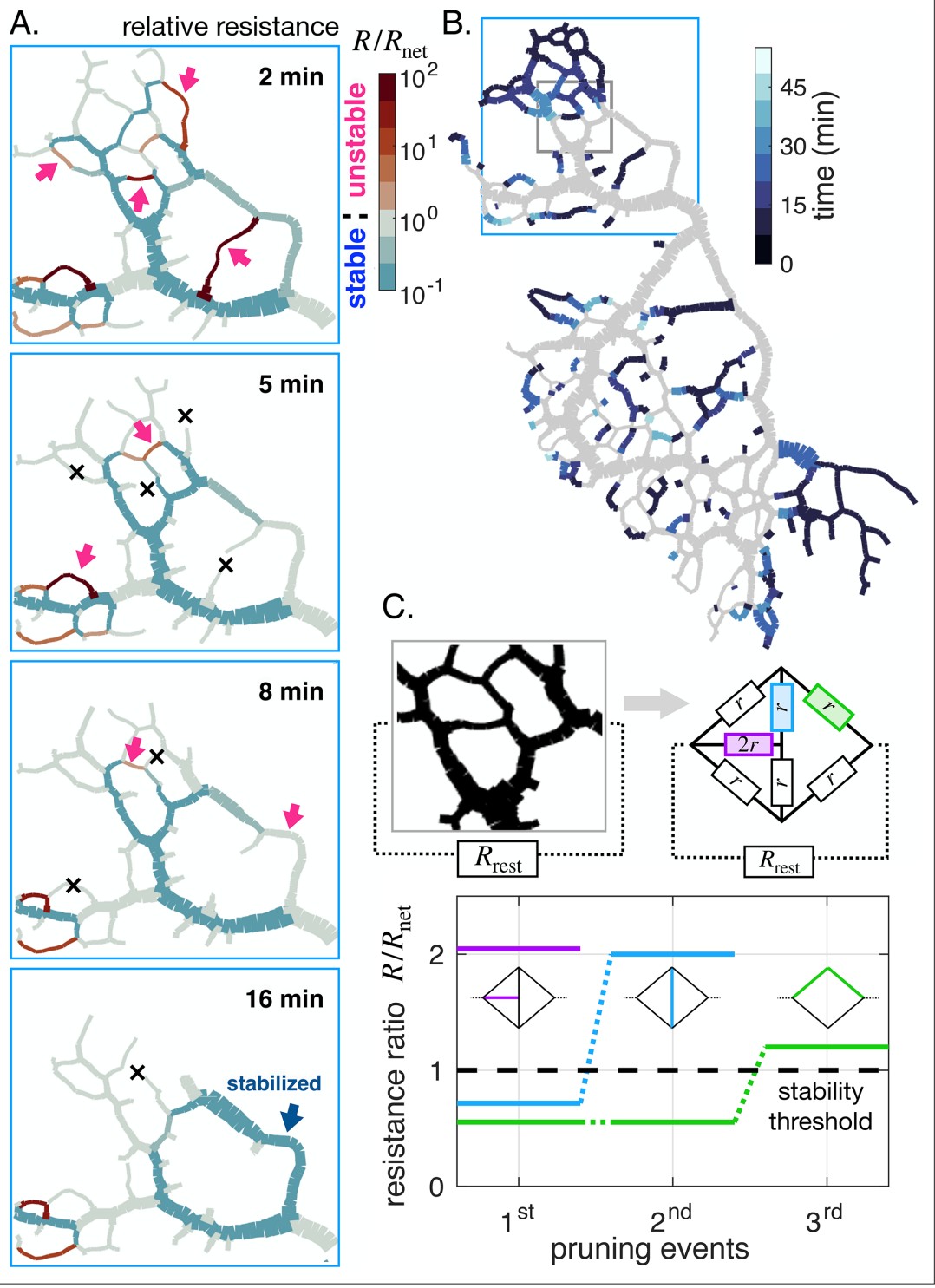

**Figure 6.** Avalanche of sequentially vanishing veins. (**A**) Time series of network reorganization. Each vein is colored according to the ratio between the resistance of an individual vein $R$ and the rest of the network $R_{net}$ in each vein. Red arrows highlight vanishing veins in the experiment; black crosses indicate veins that disappeared within the previous time frame. Veins for which the relative resistance cannot be calculated, such as dangling ends, are plotted with $R/R_{net} = 1$. (**B**) Map indicating vanishing vein events, with veins colored according to their disappearance time reported in the color scale. Gray veins will remain throughout the experiment. (**C**) Dynamics of the relative resistance of the three color-coded veins within a minimal network, inspired by the highlighted gray region of the network in (**B**). Vein resistances are chosen as $R = r$ except for a perturbed vein for which $R = 2r$.

*Figure 6 continued on next page*

*Figure 6 continued*

$R_{\text{rest}}$ represents the rest of the network relative to the region. In this model, a vein vanishes if its individual relative resistance $R/R_{\text{net}} > 1$. The disappearance of veins sequentially increases the relative resistance of neighboring veins, making them unstable. Here $r/R_{\text{rest}} = 0.1$, yet similar behavior was obtained consistently over a wide range of $r$ values.

---

through the rest of the network, $R/R_{\text{net}}$. When $R/R_{\text{net}} > 1$ (reciprocally $R/R_{\text{net}} < 1$), this preconditions the vein to shrink (respectively to grow or be stable). While $R/R_{\text{net}}$ is directly related to shear, it can be easily computed from network morphology, *without* needing to resolve flows. Both relative resistance $R/R_{\text{net}}$ and local pressure $\langle P \rangle$ are based on fluid flow physics and are indirect measures of the entire network architecture. Yet, network architecture strongly depends on time. As unstable veins vanish, the relative architecture of changes, inducing avalanches of vanishing veins, resulting in significant spontaneous reorganization.

While our experimental investigation is specific to *P. polycephalum*, we expect that the two key concepts unraveled here, time delay and network architecture governing vein fate through relative resistance and fluid pressure, may very well be at play in other vascular networks. First, the ubiquity of delayed shear rate feedback, beyond the contractile response of actomyosin, suggests that a diversity of vein dynamics (circling, non-monotonic) may also occur in other vascular networks. In fact, also the turnover time for actin filaments in living cells ranges from 10 s to 30 s, close to our measured time delay (*Fritzsche et al., 2013*; *Livne et al., 2014*; *Colombelli et al., 2009*). Other pathways, such as chemical pathways for sheared endothelial cells in blood vasculature, are processed with a time delay of a few minutes (*Lu and Kassab, 2011*; *Godbole et al., 2009*; *Fernandes et al., 2018*), while reorganization occurs on longer timescales ranging from 15 min for individual cells to several days for blood vasculature (*Livne et al., 2014*; *Landau et al., 2018*).

Second, network architecture feedback, through relative resistance and pressure, is connected to the laminar flows pervading the network. Thus, our perspective could be extended to other networks where laminar flows are an essential building block, in essence, to the diversity of networks where Murray's law holds at steady state (*West et al., 1997*; *Kassab, 2006*; *McCulloh et al., 2003*; *Akita et al., 2017*; *Fricker et al., 2017*). Particularly, our insight suggests simple parameters to map out, such as the purely geometrical relative resistance. Likely these parameters, which integrate network architecture, may explain discrepancies between shear rate and network reorganization in other vascular networks (*Chen et al., 2012*; *Baumgarten and Hauser, 2013*; *Rosenfeld et al., 2016*; *Chang and Roper, 2019*; *Sugden et al., 2017*).

Notably, imaging of biological flow network as a whole is, as of now, a rare feature of our experimental system that enabled us to unravel the importance of the network architecture for vein fate. Yet, we are hopeful that future theoretical work may allow for vein fate prediction with relative resistances determined only with partial information of a network's architecture, with sufficient accuracy. At the same time, novel experimental techniques now open up the way for in toto imaging of vascular systems and quantitative assessment of dynamics (*Daetwyler et al., 2019*).

The fact that pervading flows and network architecture are so intermingled originates in the simple physical principle that flows are governed by Kirchhoff's laws at nodes, and hence 'autonomously' sense the entirety of the network's architecture. Yet, Kirchhoff's laws are not limited to flow networks, but also govern electrical (*Dillavou et al., 2022*), mechanical (*Hexner et al., 2018*; *Goodrich et al., 2015*; *Berthier et al., 2019b*; *Berthier et al., 2019a*), thermal (*Chen et al., 2015*) and resistor-based neural networks (*Erokhin et al., 2010*; *Li et al., 2018*). Having the physics of Kirchhoff-driven self-organization at hand may thus pave the way for autonomous artificial designs with specific material (*Hexner et al., 2018*; *Goodrich et al., 2015*) or learning properties (*Dillavou et al., 2022*; *Erokhin et al., 2010*; *Li et al., 2018*).

## Acknowledgements

The authors are indebted to Charles Puelz, Emilie Verneuil, and Agnese Codutti for enlightening discussions. SM was supported in part by the MRSEC Program of the National Science Foundation under Award Number DMR-1420073. This work was supported by the Max Planck Society and has received funding from the European Research Council (ERC) under the European Union's Horizon 2020 research and innovation programme (grant agreement No. 947630, FlowMem).

## Additional information

### Funding

| Funder | Grant reference number | Author |
| --- | --- | --- |
| National Science Foundation | MRSEC Program DMR-1420073 | Sophie Marbach |
| Max Planck Society | | Karen Alim |
| Horizon 2020 - Research and Innovation Framework Programme | Grant agreement No. 947630 | Karen Alim |

The funders had no role in study design, data collection and interpretation, or the decision to submit the work for publication. Open access funding provided by Max Planck Society.

### Author contributions

Sophie Marbach, Data curation, Formal analysis, Investigation, Writing – original draft, Writing – review and editing; Noah Ziethen, Data curation, Formal analysis, Investigation, Writing – original draft, Experiments; Leonie Bastin, Data curation, Visualization, Data Acquisition, Experiments; Felix K Bäuerle, Software; Karen Alim, Conceptualization, Supervision, Writing – original draft, Writing – review and editing, Data Acquisition, Experiments

### Author ORCIDs

Sophie Marbach ⬡ http://orcid.org/0000-0002-2427-2065
Noah Ziethen ⬡ http://orcid.org/0000-0002-5936-3450
Karen Alim ⬡ http://orcid.org/0000-0002-2527-5831

### Decision letter and Author response

Decision letter https://doi.org/10.7554/eLife.78100.sa1
Author response https://doi.org/10.7554/eLife.78100.sa2

## Additional files

### Supplementary files

• MDAR checklist

### Data availability

Original microscopic images of all the specimens used for this study are available as movies in MP4 format (Videos 1–3 and Appendix 1—videos 1–8). All data used to generate the figures and the custom written matlab codes are available on the open access data repository platform mediaTUM at https://doi.org/10.14459/2023mp1705720.

The following dataset was generated:

| Author(s) | Year | Dataset title | Dataset URL | Database and Identifier |
| --- | --- | --- | --- | --- |
| Marbach S, Ziethen N, Bastin L, Bäuerle F, Alim K | 2023 | Data underlying the publication: Vein fate determined by flow-based but time-delayed integration of network architecture | https://doi.org/10.14459/2023mp1705720 | mediaTUM, 10.14459/2023mp1705720 |

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

# Appendix 1

## Preparation, imaging, and general data analysis

Microscopic images of all the specimens used for this study are available as movies in MP4 format. Numerical data analysis available at https://doi.org/10.14459/2023mp1705720.

## Preparation and imaging of *P. polycephalum*

*P. polycephalum* (Carolina Biological Supplies) networks were prepared from microplasmodia cultured in liquid suspension in culture medium (*Li et al., 2018*; *Fessel et al., 2012*). For the full network experimental setup, as in *Figure 1B* of the main text (see also *Video 2*, *Appendix 1—Video 7*, and *Appendix 1—Video 8*) microplasmodia were pipetted onto a 1.5% (w/v) nutrient free agar plate. A network developed overnight in the absence of light. The fully grown network was trimmed in order to obtain a well-quantifiable network. The entire network was observed after 1 h with a Zeiss Axio Zoom V.16 microscope and a 1 x/0.25 objective, connected to a Hamamatsu ORCA-Flash 4.0 camera. The organism was imaged for about an hour with a frame rate of 10 fpm.

In the close-up setup, as in *Figure 1A* of the main text (see also *Video 1*, *Appendix 1—Video 1*, *Appendix 1—Video 2*, *Appendix 1—Video 3*, *Appendix 1—Video 4*, *Appendix 1—Video 5* and *Appendix 1—Video 6*) the microplasmodia were placed onto a 1.5% agar plate and covered with an additional 1 mm thick layer of agar. Consequently, the network developed between the two agar layers to a macroscopic network which was then imaged using the same microscope setup as before with a 2.3 x/0.57 objective and higher magnification. The high magnification allowed us to observe the flow inside the veins for about one hour. Typical flow velocities range up to 1 mms$^{-1}$ (*Bykov et al., 2009*). The flow velocity changes on much longer timescales of 50 s to 60 s. To resolve flow velocity over time efficiently 5 frames at a high rate (typically 60 ms, detailed frame rates are specified for each Video) were imaged separated by a long exposure frame of about 2 s. As different objectives were required for the two setups, they could not be combined for simultaneous observation. Typically the longer exposure frame appears as a bright flash in the Videos. The 12 close-up data sets are indexed #A–L consistently in the main text and Appendix.

## Image analysis

For both experimental setups, image analysis was performed using a custom-developed MATLAB (The MathWorks) code. This procedure extracts the entire network information of the observed organism (*Bäuerle et al., 2017*): single images were binarized to identify the network's structure,

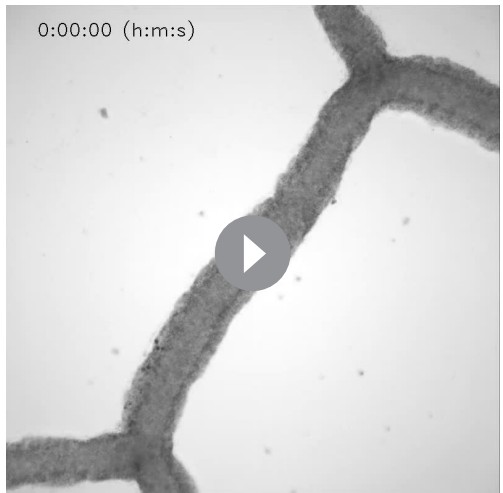

**Appendix 1—video 1.** Bright field stacked images of the close-up specimen including vein #A. Frame sequence: 5 frames at 60 ms and 1 frame at 5 s. Scale 0.574 µm/pix.

https://elifesciences.org/articles/78100/figures#video1

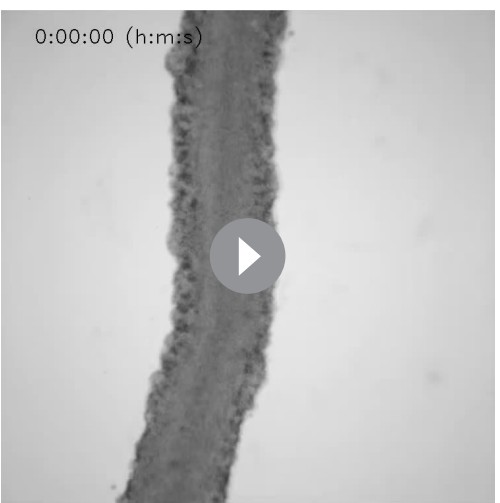

**Appendix 1—video 2.** Bright field stacked images of the close-up specimen including vein #B. Frame sequence: 5 frames at 60 ms and 1 frame at 2.5 s. Scale 0.48 µm/pix.

https://elifesciences.org/articles/78100/figures#video2

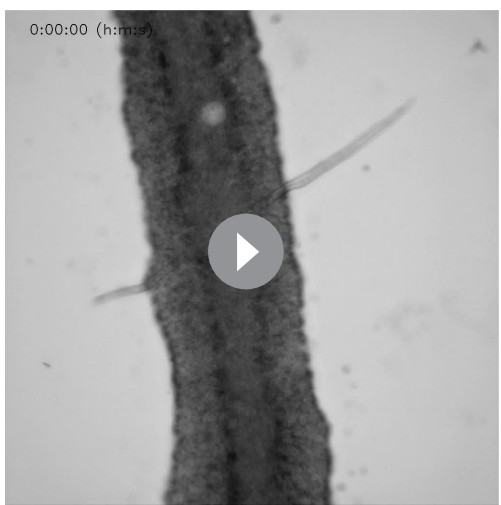

**Appendix 1—video 3.** Bright field stacked images of the close-up specimen including vein #C. Frame sequence: 5 frames at 60 ms and 1 frame at 2.5 s. Scale 0.408 µm/pix.

https://elifesciences.org/articles/78100/figures#video3

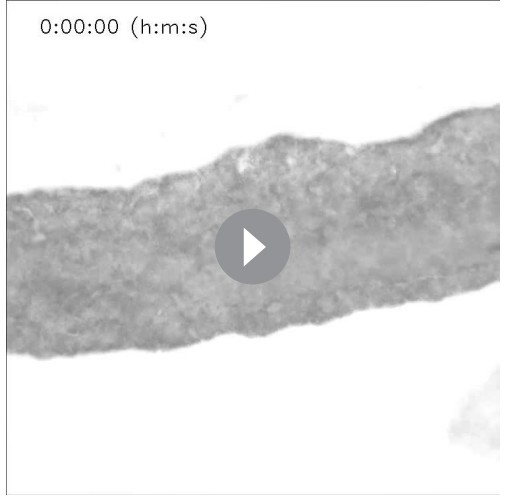

**Appendix 1—video 4.** Bright field stacked images of the close-up specimen including vein #D. Frame sequence: 5 frames at 600 ms and 1 frame at 2.5 s. Scale 0.25 µm/pix.

https://elifesciences.org/articles/78100/figures#video4

using pixel intensity as well as pixel variance information, extracted from an interval of images around the processed image. As the cytoplasm inside the organism moves over time, the variance gives accurate information on which parts of the image belong to the living organism and which parts are biological remnants. The two features were combined and binarized using a threshold. The binarized images were skeletonized and the vein radius and the corresponding intensity of transmitted light were measured along the skeleton. The two quantities were correlated according to Beer-Lambert's law and the intensity values were further used as a measure for vein radius, as intensity provides higher resolution. For the imaging with high magnification, in addition to the network information, the flow field was measured using a particle image velocimetry (PIV) algorithm inspired by *Thielicke and Stamhuis, 2014b*; *Thielicke and Buma, 2014a*; *Thielicke and Stamhuis, 2014c*, see *Figure 1A. ii* of the main paper. The particles necessary for the velocity measurements are naturally contained within the cytoplasm of *P. polycephalum*.

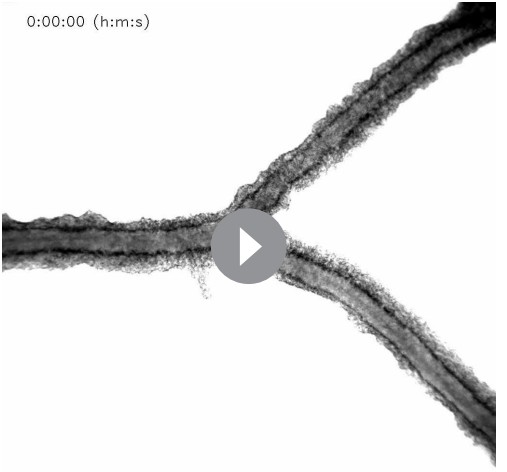

**Appendix 1—video 5.** Bright field stacked images of the close-up specimen including vein #E, #F and #G. Frame sequence: 5 frames at 60 ms and 1 frame at 2.5 s. Scale 1.06 µm/pix.

https://elifesciences.org/articles/78100/figures#video5

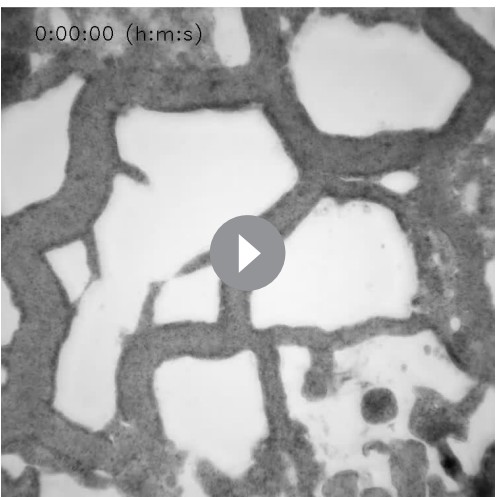

**Appendix 1—video 6.** Bright field stacked images of the close-up specimen including vein #L. Frame sequence: 5 frames at 600 ms and 1 frame at 2.5 s. Scale 0.513 µm/pix.

https://elifesciences.org/articles/78100/figures#video6

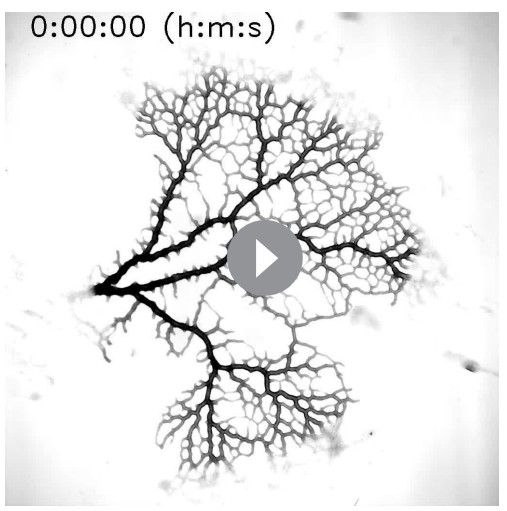

0:00:00 (h:m:s)

**Appendix 1—video 7.** Bright field stacked images of the full network specimen #2. Frame rate 6s and scale 5.36 μm/pix.

https://elifesciences.org/articles/78100/figures#video7

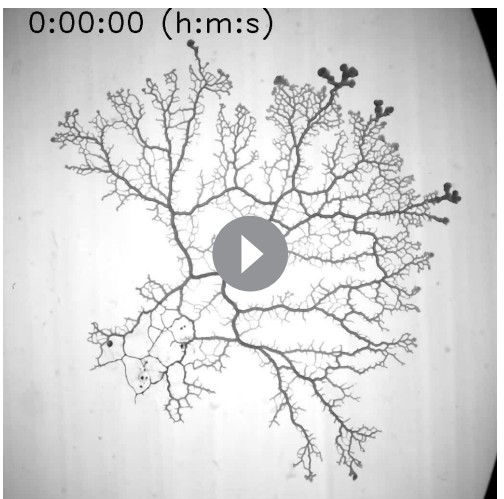

0:00:00 (h:m:s)

**Appendix 1—video 8.** Bright field stacked images of the full network specimen #3. Frame rate 6 s and scale 12.26 μm/pix.

https://elifesciences.org/articles/78100/figures#video8

## Flow calculation from vein contractions

Building on the previous image analysis, we used a custom-developed MATLAB (The MathWorks) code to calculate flows within veins for the full networks, based on conservation of mass. The algorithm follows a two stage process.

First, the network structure obtained from the images was analyzed to construct a *dynamic* network structure. This structure consists in discrete segments that are connected to each other at node points. At every time point, the structure can evolve according to the detected vein radii: if a radius is lower than a certain threshold value, the corresponding segment vanishes from the structure. Segments which are isolated due to vanishing segments are also removed. We carefully checked by eye that the threshold levels determining when a segment vanished agreed with bright-field observations. Note that we do not account for entirely new segments in the dynamic structure. As no substantial growth occurs in our data, this is a good approximation.

Second, flows and pressure in each segment were calculated building on *Alim et al., 2013*. We formalize this step briefly. Let $n$ and $p$ be two indices to describe node $n$ and node $p$ connected by a segment say $i$. In each segment, there is an unknown inflow from neighboring segments $Q_{0,np}$. There is also added flow arising due to periodic contractions $Q_{\mathrm{in},np} = 2\pi a_i L_i \frac{\partial a_i}{\partial t}$ where $a_i$ denotes the radius of segment $i$ and $L_i$ is the length of the vein. Note that all flows are given directed from node $n$ to node $p$. As a result the flow arriving from segment $i$ at node $p$ is simply $Q_{0,np} + Q_{\mathrm{in},np}$. According to Kirchhoff laws, at each node in the network, at each time point, the total incoming flux from each segment has to be zero

$$\sum_p Q_{0,np} + Q_{\mathrm{in},np} = 0. \tag{A1.1}$$

This can be rewritten

$$\sum_p Q_{\mathrm{in},np} = -\sum_p Q_{0,np} = -Q_p. \tag{A1.2}$$

where $Q_p$ are the new unknowns. Since Poiseuille law holds, the $Q_{0,np}$ are given by

$$P_n - P_p = -Q_{0,np}R_{np}, \tag{A1.3}$$

where $P_n$ is the local pressure at node $n$ (respectively $P_p$ at node $p$) and $R_{np}$ is the hydraulic resistance of a vein such that $R_{np} = \pi a_i^4/8\mu L_i$. Hence

$$Q_p = \sum_p \frac{P_n - P_p}{R_{np}}, \tag{A1.4}$$

which is a linear equation of the form $\bar{Q} = \bar{\bar{G}}\bar{P}$ where $\bar{P}$ is the vector of pressure at each node in the network, similarly $\bar{Q}$ is the vector of unknown inflows at each node, and $\bar{\bar{G}}$ is a matrix of inverse resistances taking into account the architecture of the network. We can invert this equation to obtain the values of pressure at network nodes. Then we calculate the inflow from neighboring veins through **Equation A1.3**. Finally, we obtain pressure in segment $i$ as $P_i = (P_n + P_p)/2$.

Compared to **Alim et al., 2013**, we introduced two major additions. On the one hand, the actual live contractions $a_i(t)$ are used, as detected from sequential images. To ensure that Kirchhoff's laws are solved with a good numerical accuracy, the radius traces $a_i(t)$ were (1) adjusted at each time so that overall cytoplasmic mass is conserved (mass calculated from image analysis varied by less than 10% over the analysis time) and (2) overdiscretized in time by adding 2 linearly interpolated values between each frame. Hence the simulation time step $\Delta t = 2$ s is 3 times smaller than the acquisition time, and favors numerical convergence of all time dependent processes. Note that the results were found to be independent of the simulation time step $\Delta t$ when decreasing it by a factor 2. On the other hand, a segment (or several) that vanishes creates (just before disappearing) an added inflow of $-\pi a_i^2 L_i/\Delta t$, where $a_i$ the segment's radius just before disappearing. This corresponds to radius retraction as observed in the data.

## Data analysis – time averages

For all data, we extract short time averages by using a custom-developed MATLAB (The MathWorks) routine. To determine the short time averages of the oscillating shear rate and vein radius, we used a moving average with a window size of $t_{\text{ave}} \simeq 2 - 3T$ ($T \simeq 120$ s). The $i^{\text{th}}$ element of the smoothed signal is given by $\tilde{x}_i = \frac{1}{N}\sum_j^N x_{i-\frac{N}{2}+j}$, where $N$ is the window size. At the boundary where the averaging window and the signal do not overlap completely, a reflected signal was used as compensation. This can be done because the averaging window is relatively small and the average varies slowly in time. The determined trend (for the close-up data sets) was then smoothed with a Gaussian kernel to reduce artefacts of the moving average filter.

In experimental data of the shear rate, we observe that raw shear rate appear to oscillate at rather high frequency (see *e.g.* **Figure 1iii**). Here we briefly rationalize this behavior. First a zoom in time of the data in **Figure 1A.iii**, see **Appendix 1—figure 1**, shows that in fact the frequency at which raw shear rate oscillates is double that of the frequency of oscillations of the vein radius. We explain this frequency doubling based on a minimal example. Consider a minimal example with a contraction pattern $a(t) \simeq \langle a\rangle(t)(1 + \epsilon\cos(2\pi t/T))$, where the average radius slowly evolves in time as $\langle a\rangle = L\cos(2\pi t/t_{\text{adapt}})$. The flow in the vein is $Q = L\frac{d(\pi a^2)}{dt}$ and therefore the shear rate at lowest order in $\epsilon$ is

$$\tau(t) = \frac{4}{\pi}\frac{|Q|}{a^3} \simeq \frac{4}{\pi}\frac{4\pi L}{T\langle a(t)\rangle}\epsilon|\sin(2\pi t/T)|.$$

The resulting shear rate contains the absolute value of a periodic quantity of period $T$, hence, is periodic with half the period $T/2$. We plot the minimal example curves in **Appendix 1—figure 1B**.

We further check whether our algorithm to extract the shear rate trend is correct even on these high frequency raw data. Averaging the raw shear rate obtained in the above minimal model over one contraction period yields

$$\langle\tau(t)\rangle = \frac{1}{T}\int_0^T \tau dt = \frac{2}{T}\frac{4}{\pi}\frac{4\pi L}{T\langle a\rangle}\epsilon\int_0^{T/2}\sin(2\pi t/T) = \frac{32L}{\pi T\langle a(t)\rangle}\epsilon$$

which is exactly the amplitude of the raw $\tau$ data up to a constant numerical prefactor. Hence, our averaging is well suited to extract reliable trends of the shear rate. In **Appendix 1—figure 1B**, we present the results from our averaging algorithm (full thick red line) and the theoretically calculated trend (yellow dashed) and obtain excellent agreement. Our time-averaging algorithm is therefore well-suited to the investigation of even these high frequency data.

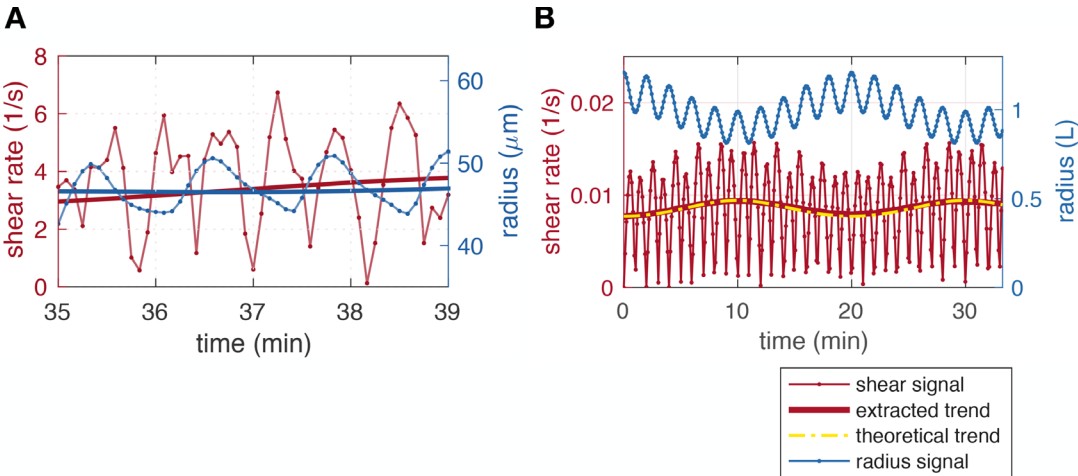

**Appendix 1—figure 1.** Extracting average shear rates from shear rate data. (**A**) Time zoom of a close-up data set (that of *Figure 1A.iii*, #K) showing the doubling of the frequency of the shear rate compared to the radius. (**B**) Minimal model example with short timescale and long timescale radius oscillations, resulting in shear rate with a doubled frequency. Here, the contraction period $T = 2$ min and $t_{adapt} = 20$ min.

## Data analysis – Additional shear rate - radius data
To add to the data presented in *Figure 1iv* presenting the time-averaged dynamics of radius adaptation and shear rate, we show in *Appendix 1—figure 2* (resp. *Appendix 1—figure 3*) additional dynamics for the close-up datasets (respectively the full network #1 of *Figure 1B*).

## Data analysis – Additional data on full networks
### Additional data on different full network specimen
In what follows we present additional data on full networks. In particular, we investigate two other full networks besides specimen #1 (of *Figure 1B*), which we call #2 and #3. These two additional networks show significant spontaneous reorganization over time and we show snapshots of their initial and final networks in *Appendix 1—figure 4A, B* and *Appendix 1—figure 5A, B*.

We also present additional data to demonstrate the existence of similar ambiguity in shear rate - radius response in other full networks. We show with yellow arrows additional places where shear rate is initially high yet the vein will disappear in *Appendix 1—figure 4C* and *Appendix 1—figure 5C*. Red dots in *Appendix 1—figure 4B* and *Appendix 1—figure 5B* also show veins where shear rate is initially low however these veins will grow in time.

We present pressure data in *Appendix 1—figure 4D* and *Appendix 1—figure 5D*. We find that pressure doesn't vary much throughout the network. A global pressure wave is observed corresponding to a stable direction of the peristaltic contractions. We identify in these maps nearby veins and find that the ones with larger pressure remain (blue stable) while those with lower pressure vanish (pink unstable).

We present relative resistance data $R/R_{net}$ in *Appendix 1—figure 4E* and *Appendix 1—figure 5E*. We find a number of veins with $R/R_{net} > 1$, indicated by pink arrows, that indeed vanish in time.

To finish with the analysis of additional networks, we present a map of the time of disappearance of veins in the full specimen in *Appendix 1—figure 4F* and *Appendix 1—figure 5F*. We find that loops consistently vanish by their center, as highlighted via black arrows.

### Additional data on full network specimen #1
In *Appendix 1—figure 6* we present additional data on Specimen #1 that is the main example under scrutiny in the main text. We provide in particular maps of quantities that are not shown in the main text, such as the connected resistance $R_{net}$ (C) and $Q_{in} = \left\langle \left| \pi \frac{da^2}{dt} \right| \right\rangle$ (F). We find that $Q_{in}$ typically evolves like the vein radius: showing larger values (light blue) for larger veins and reciprocally smaller values (dark blue) for smaller veins. $R_{net}$ in contrast evolves quite dramatically from vein to vein, according to how the vein is close or not to major highways.

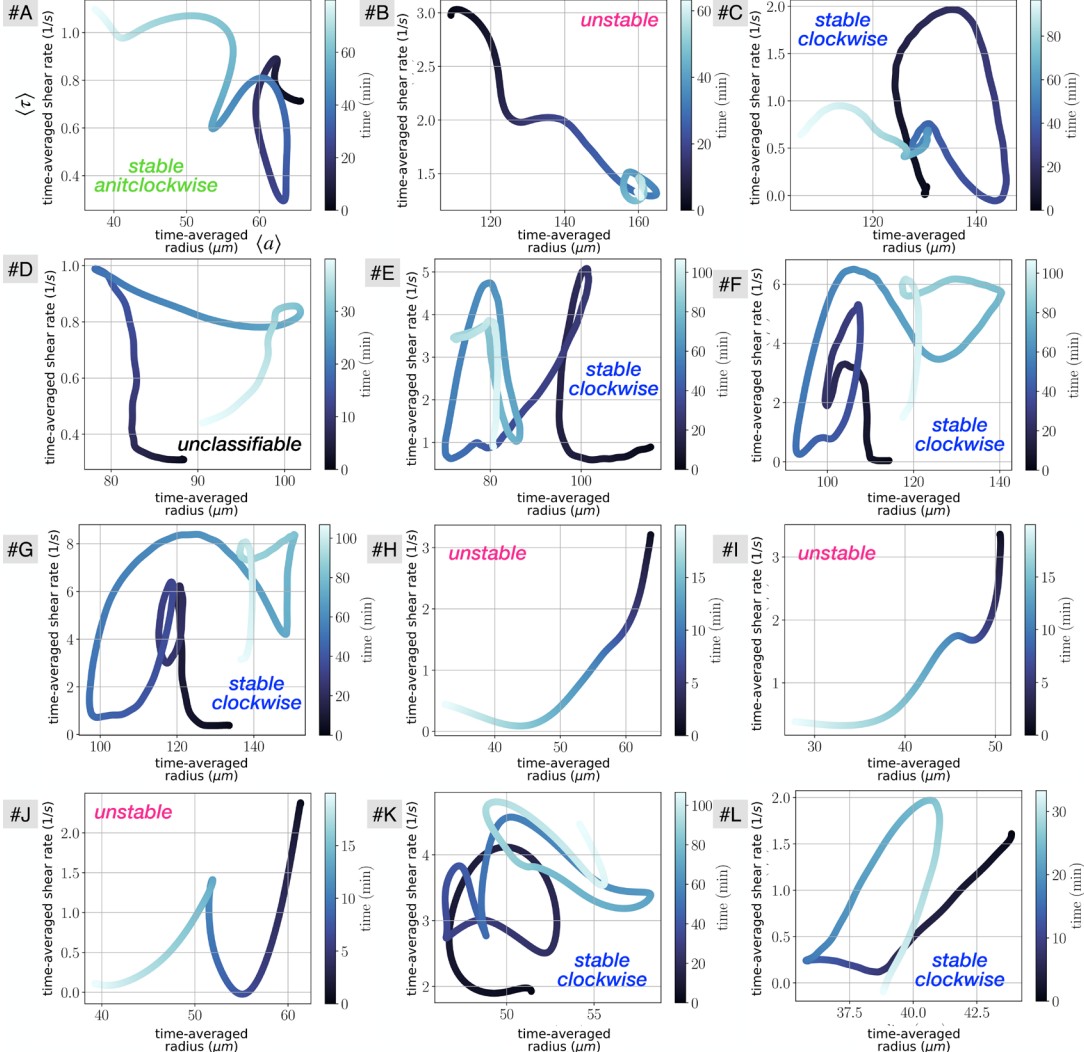

**Appendix 1—figure 2.** Vascular adaptation dynamics for all close up experiments, using time-averaged shear rates $\langle\tau\rangle$ and time-averaged radius $\langle a\rangle$. The letters indicate the data set names, and are used consistently throughout the manuscript. Typical classification of vein dynamics is indicated for each plot.

We also provide cross-correlation data between specific quantities and initial vein radius $\langle a\rangle$ at the beginning of the experiment, $R_{net}$, $R/R_{net}$, $Q_{in}$ and $\langle P\rangle$ (A,B,D,E). We find that the only quantity that is significantly correlated with $\langle a\rangle$ is $Q_{in}$, coherently since we expect $Q_{in} \propto \langle a^2\rangle$.

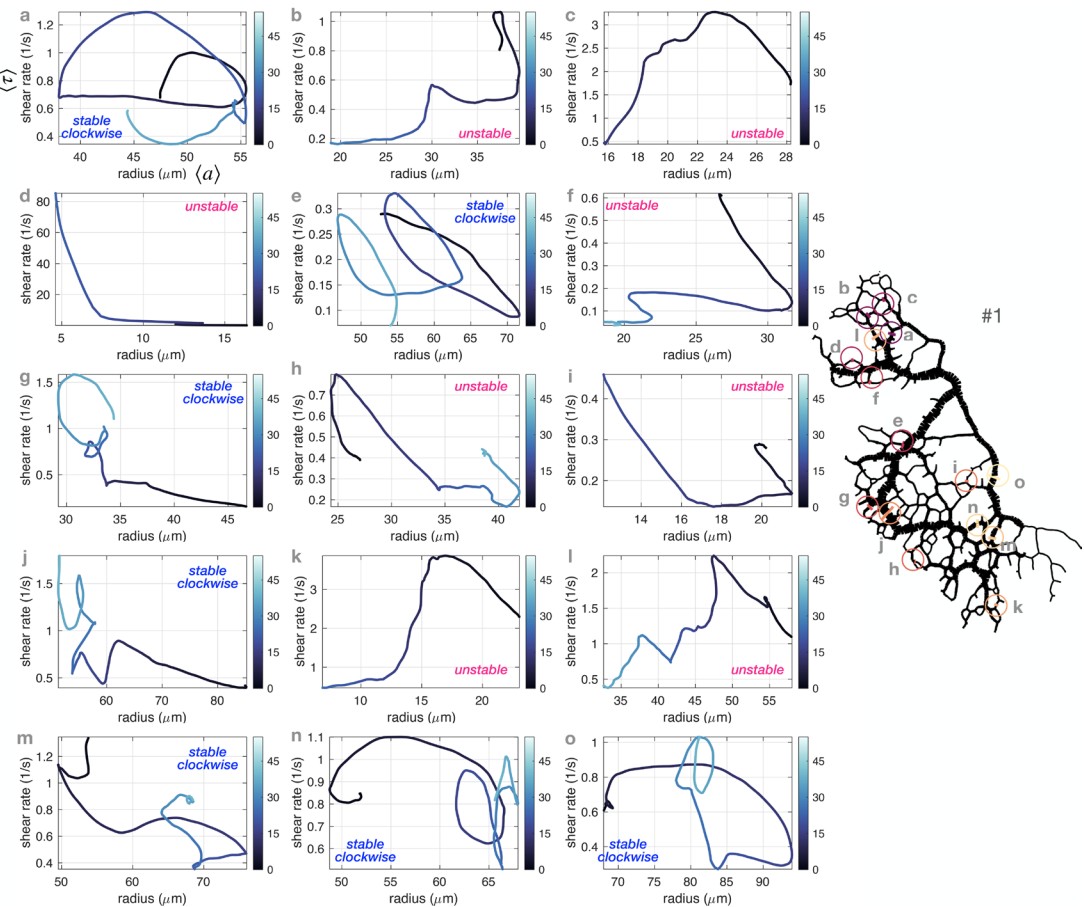

**Appendix 1—figure 3.** Vascular adaptation dynamics for a few veins in the full network #1 using time-averaged shear rates ⟨τ⟩ and time-averaged radius ⟨a⟩. The veins shown are chosen randomly but distributed throughout the network. The network sketch on the right hand side shows circles indicating at their center the location of each vein, with consistent labels. Typical classification of vein dynamics is indicated for each plot.

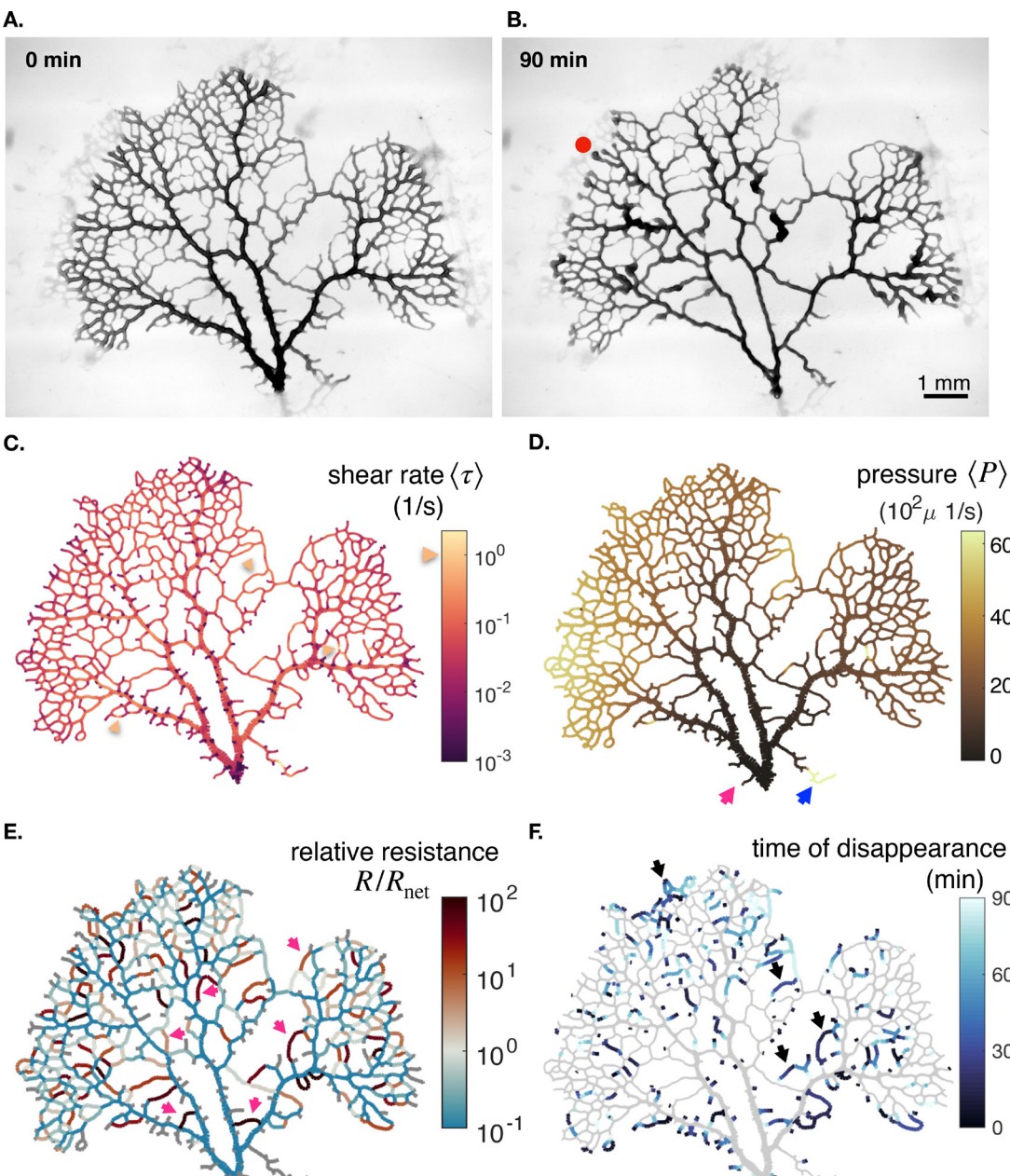

**Appendix 1—figure 4.** Additional data on the full network specimen #2. (**A - B**) Bright field images of a specimen with long time dynamics of vanishing veins, specimen 2. (**c**) Mean shear rate $\langle \tau \rangle$, (**D**) pressure $\langle P \rangle$ and (**E**) relative resistance $R/R_{\text{net}}$ at the initial stage. (**F**) Time of vein disappearance for the entire experiment. See text for more information on the arrows and what to take out from the color maps.

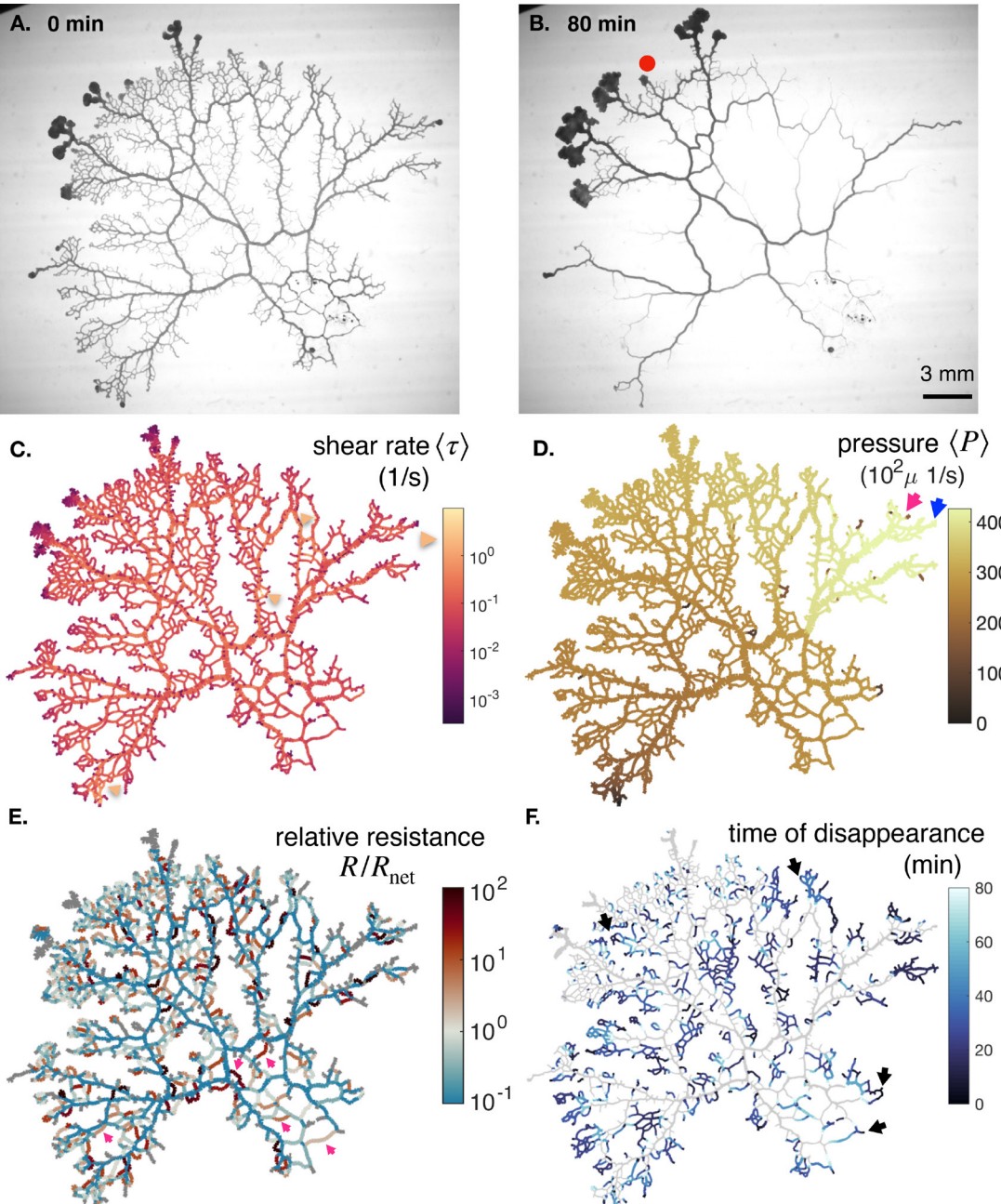

**Appendix 1—figure 5.** Additional data on the full network specimen #3. (**A–B**) Bright field images of a specimen with long time dynamics of vanishing veins, specimen . (**c**) Mean shear rate $\langle \tau \rangle$, (**D**) pressure $\langle P \rangle$ and (**E**) relative resistance $R/R_{\mathrm{net}}$ at the initial stage. (**F**) Time of vein disappearance for the entire experiment. See text for more information on the arrows and what to take out from the color maps.

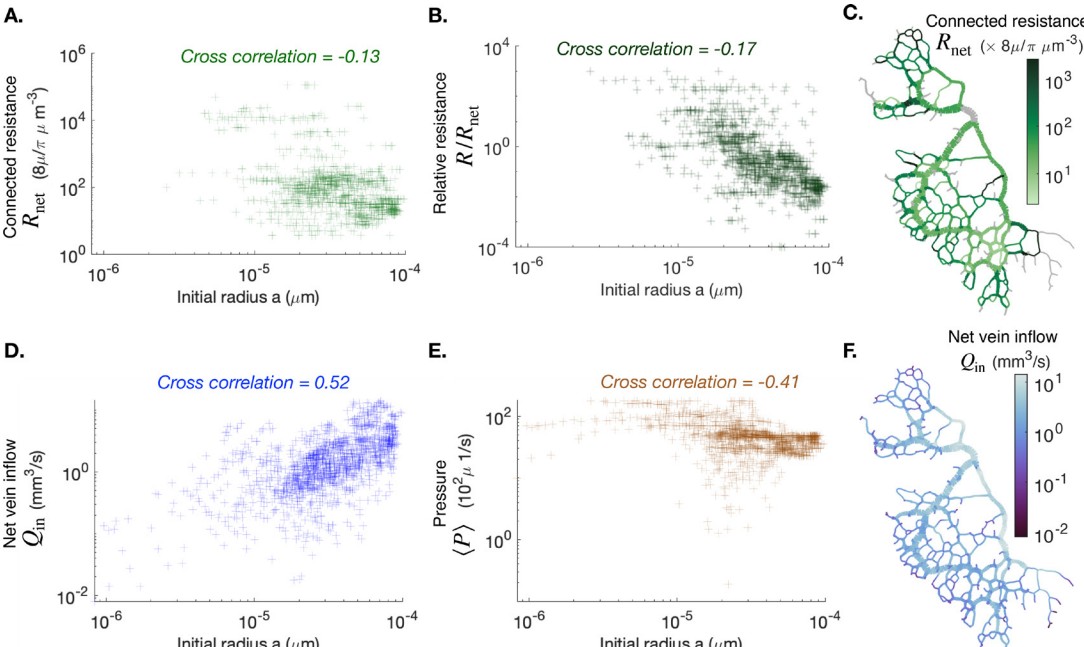

**Appendix 1—figure 6.** Cross correlation between average vein radius and different flow-based parameters (**A**) the connected resistance $R_{net}$, (**B**) the relative resistance $R/R_{net}$, (**C**) the vein outflow $Q$ and (**E**) the local pressure $\langle P \rangle$. We also present maps of the connected resistance $R_{net}$ in (**C**) and of the vein outflow $Q$ in (**F**). The color scales indicate the magnitude of each variable in each colored vein. All cross-correlations and maps are done at the initial observation time for the full network specimen #1.

**Appendix 1—table 1.** List of commonly used variables in our work in alphabetical order and significance.

Short length scale variations correspond to variables that can vary strongly from one vein to a neighboring vein, while long length scale variations vary smoothly throughout the network. Variables have short timescale variations when they have significant variations over timescales much smaller than the peristaltic contractions $T \simeq 1 - 2$ min; and long timescale variations if they vary over longer timescales corresponding to vascular adaptation and rearrangement.

| Variable | Significance | Length scale variations | Timescale variations |
|---|---|---|---|
| $a$ | Radius of a vein | Short | Short and long |
| $\epsilon$ | Relative contraction amplitude | – | – |
| $L$ | Length of a vein | – | – |
| $\mu$ | Inner fluid viscosity | Long | Long |
| $\omega = 2\pi/T$ | Contraction frequency | – | – |
| $P$ | Inner fluid pressure | Long | Short and long |
| $P_0$ | Atmospheric pressure | – | – |
| $Q$ | Fluid flow pervading a vein | Short | Short and long |
| $Q_{in}$ | Fluid flow generated by peristaltic contractions in a vein | Short | Short and long |
| $Q_{net}$ | Fluid flow generated by the rest of the network | Short | Short and long |

*Appendix 1—table 1 Continued on next page*

*Appendix 1—table 1 Continued*

| Variable | Significance | Length scale variations | Timescale variations |
|---|---|---|---|
| $R$ | Resistance of a vein, $R = 8\mu L/\pi a^4$ | Short | Short and long |
| $R_{\text{net}}$ | Resistance of the rest of the network attached to a vein at both vein ends | Short | Short and long |
| $\tau$ | Shear rate on a vein's inner wall | Short | Short and long |
| $\tau_s$ | Sensed shear rate for adaptation | Short | Long |
| $\tau_0$ | Steady state shear rate, $\tau_0 \sim \tau_{\text{active}} - \langle P - P_0 \rangle/\mu$ | Long | Long |
| $\tau_{\text{active}}$ | Active contribution to $\tau_0$, due to energetic consumption from the actomyosin cortex to maintain contractions | Long | Long |
| $t$ | Time | – | – |
| $t_{\text{adapt}} \simeq 10 - 100 \text{ min}$ | Long-time adaptation timescale | Long | Long |
| $t_{\text{delay}} \simeq 2 \text{ min}$ | Delay between adaptation and shear rate | Long | Long |
| $T \simeq 1 - 2 \text{ min}$ | Peristaltic contraction period | – | – |
| $\langle X \rangle$ | Long-time average of $X$ | – | Long |

## Appendix 2

### Adaptation model, time delay, and fitting procedure

Vascular adaptation from force balance

We briefly summarize here the derivation of our vascular adaptation model from force balance and provide more details in our accompanying publication (*Marbach et al., 2023*). We consider the force balance equation on a small vein wall segment of radius $a$, length $L$, thickness $e$. As the wall motion is typically slow and occurring over microscopic scales we neglect inertial contributions and write

$$0 = 2\pi aL\left((P - P_0) + \sigma_{\text{circum}} + \sigma_{\text{active}} + \sigma_r(\mu\tau_s)\right) - \gamma L\frac{da}{dt}, \tag{A2.1}$$

where $P - P_0$ is the hydrodynamic pressure difference between interior and exterior, $\sigma_{\text{circum}}$ is the circumferential stress (or elastic tension), $\sigma_{\text{active}}$ corresponds to active stresses from the actomyosin cortex (*Radszuweit et al., 2013*; *Alonso et al., 2017*), and $\gamma L\frac{da}{dt}$ is the friction force reflecting the long timescale for fiber rearrangement (*Salbreux et al., 2012*; *Fischer-Friedrich et al., 2016*). Note that since the shear rate $\tau$ acts longitudinally on the walls, it does not contribute to the force balance on the radial direction. Yet, the vein walls consist of a material with an anisotropic response to shear, namely cross-linked fibers (the actomyosin gel). Hence, when sheared, a radial stress $\sigma_r(\mu\tau_s)$ builds up as a result of longitudinal shear rate sensing (with a time delay) (*Gardel et al., 2008*; *Janmey et al., 2007*; *Vahabi et al., 2018*; *Lu and Kassab, 2011*; *Godbole et al., 2009*; *Fernandes et al., 2018*).

The general force balance (2.1) significantly simplifies when we average over the short timescales of vein contractions (1–2 min) (*Isenberg and Wohlfarth-Bottermann, 1976*), typically corresponding to elastic deformations, to focus on the longer timescales of 10–60 min corresponding to vein wall assembly or disassembly inherited from *e.g.* actin fiber rearrangements (*Salbreux et al., 2012*; *Fischer-Friedrich et al., 2016*).

On these longer timescales, significant morphological vein adaptation of $\langle a \rangle$ occurs. $\langle \sigma_{\text{active}} \rangle$ is a constant as it is expected to vary only on short timescales in line with the periodic contractions. Note also that it is a negative stress, that tends to shrink a vein – this reflects the impact of metabolic cost, here induced by vein wall activity. $\langle \sigma_{\text{circum}} \rangle \simeq 0$ over short timescales, as such forces are intrinsically elastic forces and hence do not pertain long time features. Finally, our numerical calculations of pressures within observed networks show that $\langle P - P_0 \rangle$ depends smoothly on the location within the network, but barely varies in time (*Alim, 2018*; *Figure 4A*). We obtain a time-independent, yet position-specific constant $\tau_{\text{target}} = -\frac{1}{\mu}\langle P - P_0 \rangle + \tau_{\text{active}}$, where we wrote $\tau_{\text{active}} = \langle \sigma_{\text{active}} \rangle/\mu$.

Furthermore, we assume a phenomenological functional form for the radial stresses, as $\sigma_r(\mu\tau_s) \simeq \mu\frac{\tau_s^2}{\tau_c}$, in line with observations of sheared cross-linked actin fibers (*Gardel et al., 2008*; *Janmey et al., 2007*) where $\tau_c$ is a positive constant. Importantly, this radial stress, acts in the positive direction, *i.e.* dilates vessels. This functional form is also consistent with measured data on fibrin gels (*Vahabi et al., 2018*; *Kang et al., 2009*) and models of anisotropic response based on nonlinear elastic theory (*Vahabi et al., 2018*).

Finally, to simplify the expressions we now introduce $\tau_0 = \sqrt{\tau_c \tau_{\text{target}}}$ and

$$t_{\text{adapt}} = \frac{\gamma}{2\pi\mu\tau_c} \tag{A2.2}$$

a characteristic adaptation timescale for vascular rearrangement. This allows us to recover the vascular adaptation rule *Equation 1*. While the two parameters $\tau_0$ and $t_{\text{adapt}}$ may appear to be coupled at the scale of the network, there is actually no reason for $\tau_c$ or for $\gamma$ to be constant throughout the network. In fact they may very well depend on the age of the vein, the absolute thickness of the actomyosin gel, *etc*. Again, we refer the reader to more details on the derivation in our accompanying manuscript (*Marbach et al., 2023*).

Agreement with Murray's law Our model is consistent with Murray's steady state assumption. In fact, the (non-trivial) steady state of our model *Equations 1; 2* corresponds to a constant average shear in the vein $\langle \tau \rangle = \tau_0$. This corresponds exactly to Murray's result of minimum work.

In fact, Murray stipulates that the energy dissipation $\mathcal{E}$ of a single vein (of radius $a$ and length $L$) is given by flow dissipation associated with the vein's resistance and energy expense to sustain the vein

$$\mathcal{E} = \frac{1}{2}Q^2R + \pi bLa^2 = \frac{4\mu LQ^2}{\pi a^4} + \pi bLa^2. \tag{A2.3}$$

where $R = 8\mu L/\pi a^4$ is the vein resistance assuming Poiseuille flow in the vein, $b$ is a local metabolic constant per unit volume, $Q$ the flow rate and $\mu$ viscosity. The principle of minimum energy expense suggests to search for the minimum of $\mathcal{E}$ with respect to the vein radius $a$ which gives the relation $a^6_{\text{optimal}} = \frac{8Q^2\eta}{b\pi^2}$. The shear rate $\tau$ can be expressed as $\tau = \frac{4Q}{\pi a^3}$ and hence the optimal (or steady state) shear rate is independent of radius and flow rate $\tau_{\text{optimal}} = \sqrt{b/\mu}$. This is consistent with our steady state where shear rate is constant $\langle \tau \rangle = \tau_0$. The constant $\tau_0$ can thus also be interpreted as being related to the typical local energy expense to sustain the vein $\sqrt{b/\mu}$ (which corresponds very closely to our $\tau_{\text{active}}$ characterizing metabolic expense to sustain the contractile activity). Note that we bring further insight compared with Murray's derivation, as our adaptation dynamics (2.1) originates from force balance on the vein wall, and hints that $\tau_0$ (or the metabolic cost) also depends on local pressure $\langle P \rangle$.

## Extracting the time delay from data analysis

In this section we discuss our procedure to extract the time delay from data.

First, we verify that the time delay we extract is independent of the averaging technique. To do so, we investigate the time delays obtained from the cross-correlation of $da/dt$ and $\tau$ instead of their averaged counterparts $d\langle a \rangle/dt$ and $\langle \tau \rangle$. We obtain a distribution of best time delays, over the nearly 10000 vein segments of the full network, and we retain maxima regardless of the value of the cross-correlation. We present the results in **Appendix 2—figure 1A**. The average time delay is 52 s, which is comparable in orders of magnitude to the average time delay of 122 s for the same network data but where radii and shear rate trends were extracted (**Figure 2C**). Note, that the correlation however is much less clear without extracting trends and in average the correlation score is 0.25 with only 5% of veins achieving a score $gt_{0.2}$ compared to 0.66 average score with trends with 15% of veins achieving a score $gt_{0.5}$. Note that the average correlation is quite low because in general the data are

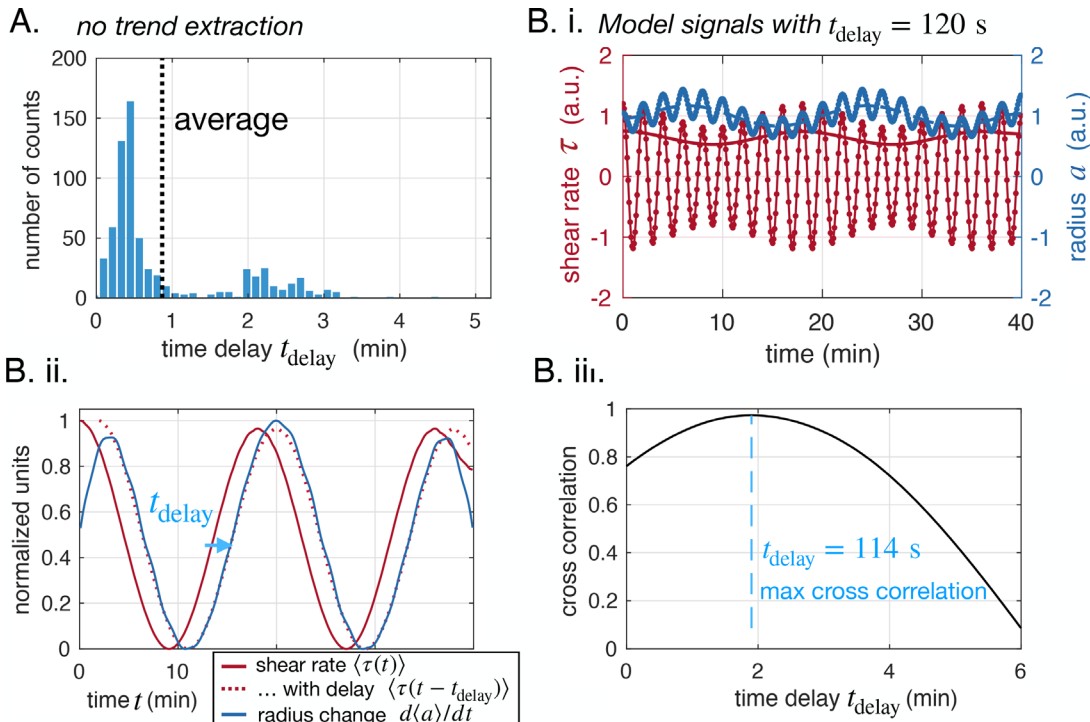

**Appendix 2—figure 1.** Time delay extracted is independent of averaging technique or oscillation frequency. (**A**) Time delays measured without extracting trends from data (same plot as **Figure 2C** but without extracting trends). (**B**) Extracting the time delay for model data with (**i**) model data and extracted trends, (ii) extracted trends and best time delay obtained shown on trends and (iii) cross correlation between $da/dt(t)$ and $\tau(t - t_{\text{delay}})$ with respect to the searched time delay $t_{\text{delay}}$, and maximum value shown.

not perfectly periodic and smooth. Hence, we decide to keep the analysis on the data trends, that appears to be much more precise.

Second, we check that even if the time delay between adaptation and shear rate is close to the peristaltic contraction frequency ($T \simeq 1 - 2$ min), we are still able to extract it with our method reliably. To do so, we consider model data $a(t)/a_0 = (1 + 0.2 \sin[\omega_s(t - t_{\text{delay}})])(1 + 0.2 \cos[\omega(t - t_{\text{delay}})])$ and $\tau(t)/\tau_0 = (1 + 0.2 \cos[\omega_s(t)]) \cos[\omega(t)]$. We impose a contraction period $T = 2\pi/\omega = 120$ s and the long time adaptation period $2\pi/\omega_s = 20$ min, and a delay similar to the beating period $t_{\text{delay}} = T = 120$ s. Using our methodology to extract the time delay, we find $t_{\text{delay}} = 114\ s$, which is equal to the set time delay of 120 s within the error bar of 6 s corresponding to the time step where data was sampled. We conclude that the time delay we obtain is independent of the value of the contraction frequency.

Finally, some trajectories appear to oscillate on long timescales say with period $T_{\text{osc}}$. Hence, it may not be obvious by cross-correlation for these specific trajectories to determine whether the delay is $t_{\text{delay}}$ or $-T_{\text{osc}} + t_{\text{delay}}$, or another combination. $T_{\text{osc}}$ characterizes rarely observed long cycles in the long time adaptation dynamics, for example see *Figure 2D*, and typically $T_{\text{osc}} = 20$ min. In contrast, the apparent phase lag between $\langle \tau \rangle$ and $\langle a \rangle$ is usually of the order of a few minutes in the samples where the delay can be inferred unambiguously ($t_{\text{delay}} \sim 1 - 5$min). We may thus expect that the time delay is indeed $t_{\text{delay}} \sim 1 - 5$min and not $-T_{\text{osc}} + t_{\text{delay}}$ which would be much longer. We impose this condition by adding a cutoff on the time delay at 5 min. Changes to the time delay cutoff, for example setting the cutoff to 10 min, does not affect the results significantly. In fact, strictly oscillatory signals are very rare. For example *Figure 1B.iii* clearly shows a lag time (between 7–15 min) that allows one to resolve the causality relation unambiguously.

## Time delays in close-up and full networks – additional data

We now present time delay analysis in all our specimens.

In *Appendix 2—figure 2* we present time delay data in full networks. Time delays (both positive and negative) were retained for veins for which the maximum cross-correlation was higher than 0.5. Time delays may only be extracted with sufficient accuracy for stable veins, which do not represent the majority of veins in the network. Hence approximately $15 - 25\%$ of observed veins reach a significant cross-correlation and allow us to record a value of the time delay. To avoid biasing the statistical search with either positive or negative time delays, we allow the algorithm to record both positive and negative time delays for a single vein if these maxima are significant. The phenomenon of negative time delays is quite infrequent. For example in specimen #1, out of the observed veins that yield a time delay, we find 94% with a positive time delay, 4% with a negative time delay, and 2% with both a positive and negative time delay. For specimen #2 we find 96% positive, 3% negative and 1% positive and negative, and for specimen #3 respectively 87%, 12% and 1%. Hence, positive time delays are much more likely. The average time delay is consistently $t_{\text{delay}} \simeq 2$ min.

We also investigate the time delay on close-up data sets (see *Appendix 2—figure 3*), and only on stable close-up data sets as they will allow us to extract the time delay more reliably. Notice that cross correlations are usually quite smooth as the correlation continuously increases until significant shear rate and radius changes are aligned. The correlation maximum corresponds to a strongly correlated configuration ($gt_{0.7}$). Vein #E finds a best time delay that is quite large ($t_{\text{delay}} \sim 12$ min), potentially due to the fact that we are exploring a very long time sequence for this particular vein and that the cross correlation algorithm picks up a large change unrelated to the actual short delay. Notice, however, that a time delay of 2–5 min potentially corresponding to the cross-correlation shoulder also seems suited here. The variability in the time delay extracted on close-up data sets show the need for statistical analysis of the time delay, which we perform on full networks.

## Fitting of the model to data

Fitting of the model *Equations 1; 2* to the data was performed using a non-linear least squares algorithm included in the SciPy optimize package (*Virtanen et al., 2020*), or a linear least squares algorithm, according to whether two or three model parameters had to be fitted. The relative fitting error is defined as

$$\epsilon_{\text{err}} = \frac{1}{N_t} \sum_{t=1}^{N_t} \left( |\langle a \rangle_t^{\text{data}} - \langle a \rangle_t^{\text{fit}}| / \langle a \rangle_t^{\text{data}} \right), \tag{A2.4}$$

where $N_t$ is the number of data points.

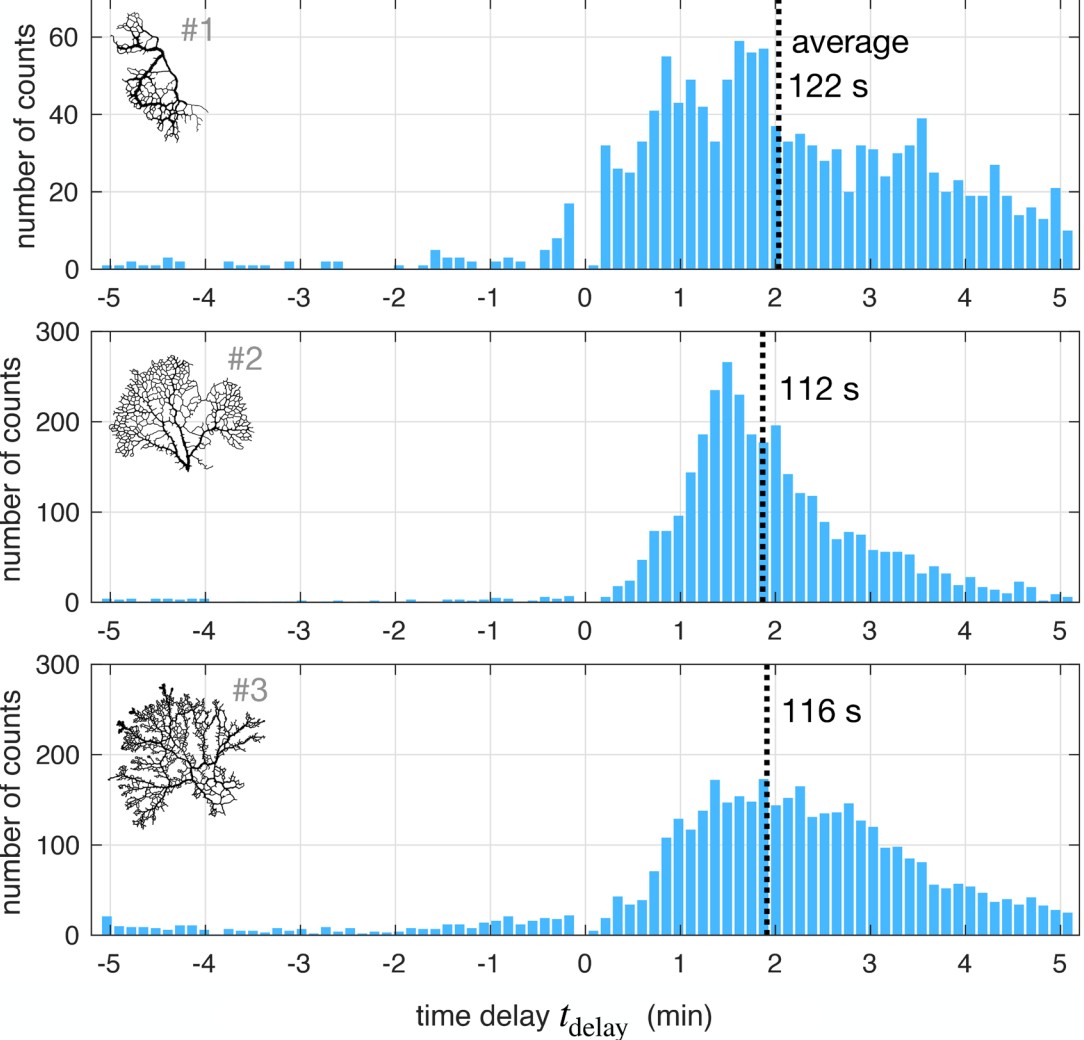

**Appendix 2—figure 2.** Time delay statistics from full networks. Distribution of best time delays for all veins in the network (#1, with about 10,000 vein segments and #2 and #3, both of which have about 30,000 vein segments). (insets). Network maps – not to scale.

First, we fit close-up data sets, for all three parameters $t_{adapt}$, $t_{delay}$ and $\tau_0$. As stressed in the main text the model parameters are not expected to be constant over long times (on which loopy trajectories are typically observable). To find suitable time frames where model parameters where approximately constant and loopy trajectories observable, we systematically varied the time windows of the data used for fitting. To find the optimal time windows for fitting including fitting the time delay $t_{delay}$, we chose close-up data sets forming loopy trajectories (#G, #E, #F and #K), as loops are a characteristic feature ensuing from the time delayed dynamics. The distribution of time delays fitted for different time windows was found to range from 1 min to 10 min (see **Appendix 2—figure 4**), which is within the range obtained via the cross-correlation algorithm described above in Appendix 2.3. Fitted trajectories reproduce the main features observed experimentally in detail. The corresponding fitted parameters are reported in **Appendix 2—table 1**.

Second, we fit all close-up data sets now only including two model parameters, $\tau_0$ and $t_{adapt}$. We fixed the time delay to a constant value $t_{delay} = 120\,\text{s}$. We fit different time intervals in the data sets and find very good agreement between data and fits – see **Appendix 2—figure 5**. We report the corresponding fitted parameters in **Appendix 2—table 2**.

Finally, we fit a random sample of 15 veins from the full network specimen #1. We include two model parameters, $\tau_0$ and $t_{adapt}$. We fixed the time delay to the value obtained by cross correlation. We fit only over one rather larger time interval of about 40 min and find reasonable agreement

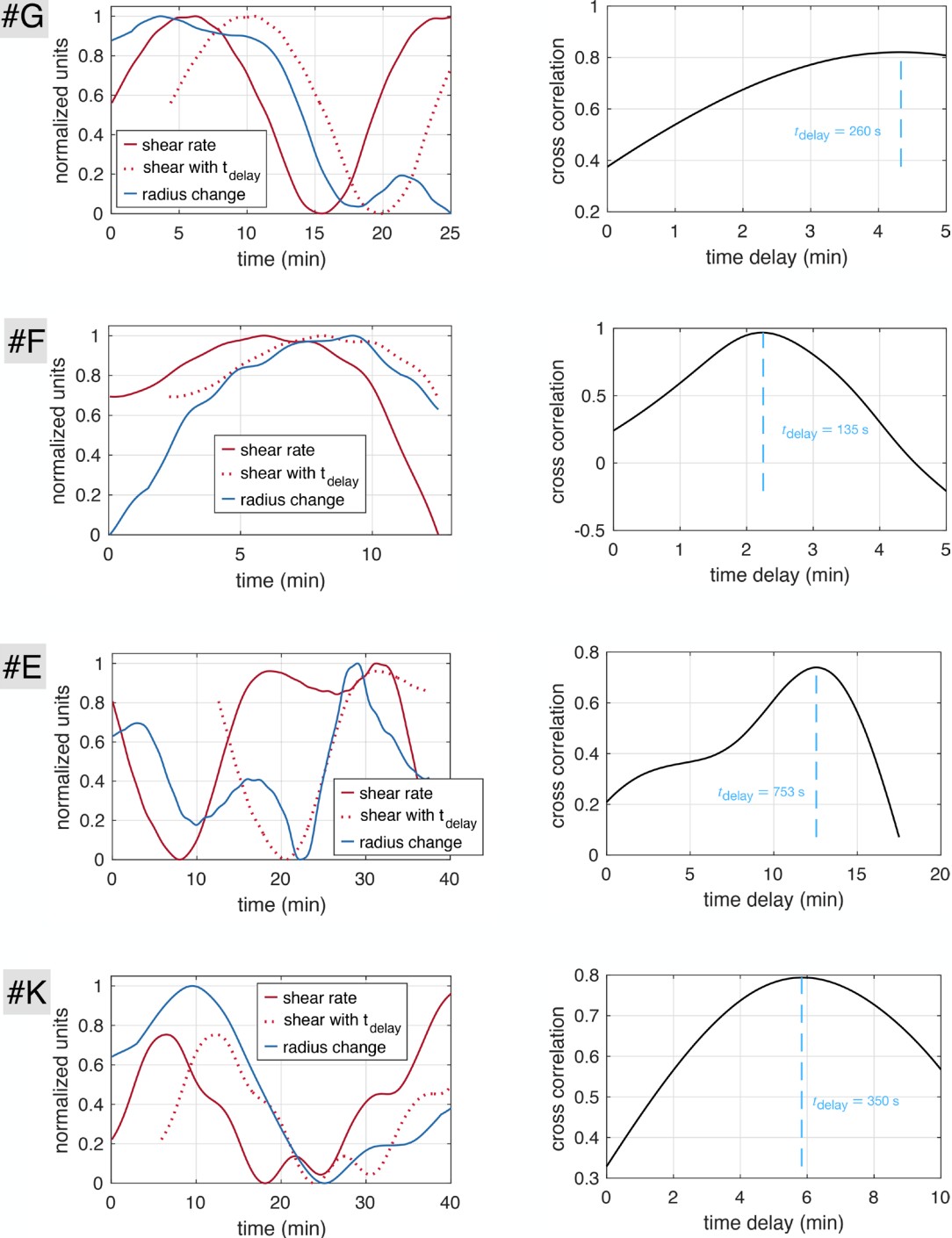

**Appendix 2—figure 3.** Time delays obtained with cross-correlation method on stable close-up data sets. (Left hand side) Time-averaged shear rate $\langle \tau \rangle$ (red) and radius change $(d\langle a \rangle/dt)$ with time for each vein (#E, #F, #G, #K), as well as time delayed shear right producing the best cross correlation $\langle \tau(t - t_{\text{delay}}) \rangle$. (Right hand side) Cross-correlation with varying time delay and optimal time delay obtained at the correlation maximum.

between data and fit (see *Appendix 2—figure 6*). The corresponding fitted parameters are reported in *Appendix 2—table 3*. In addition for the 15 veins from the full network we also fit the model with no time delay $t_{\text{delay}} = 0$. For these fits, we set $\tau_s = \langle \tau \rangle$ instead of *Equation 2*. We show the fitted results in black dotted lines in *Appendix 2—figure 6* and report here only the corresponding fitting error $\epsilon_{\text{err}}$ in *Appendix 2—table 3*. We find a systematic higher fitting error for fits without time delay over those with time delay.

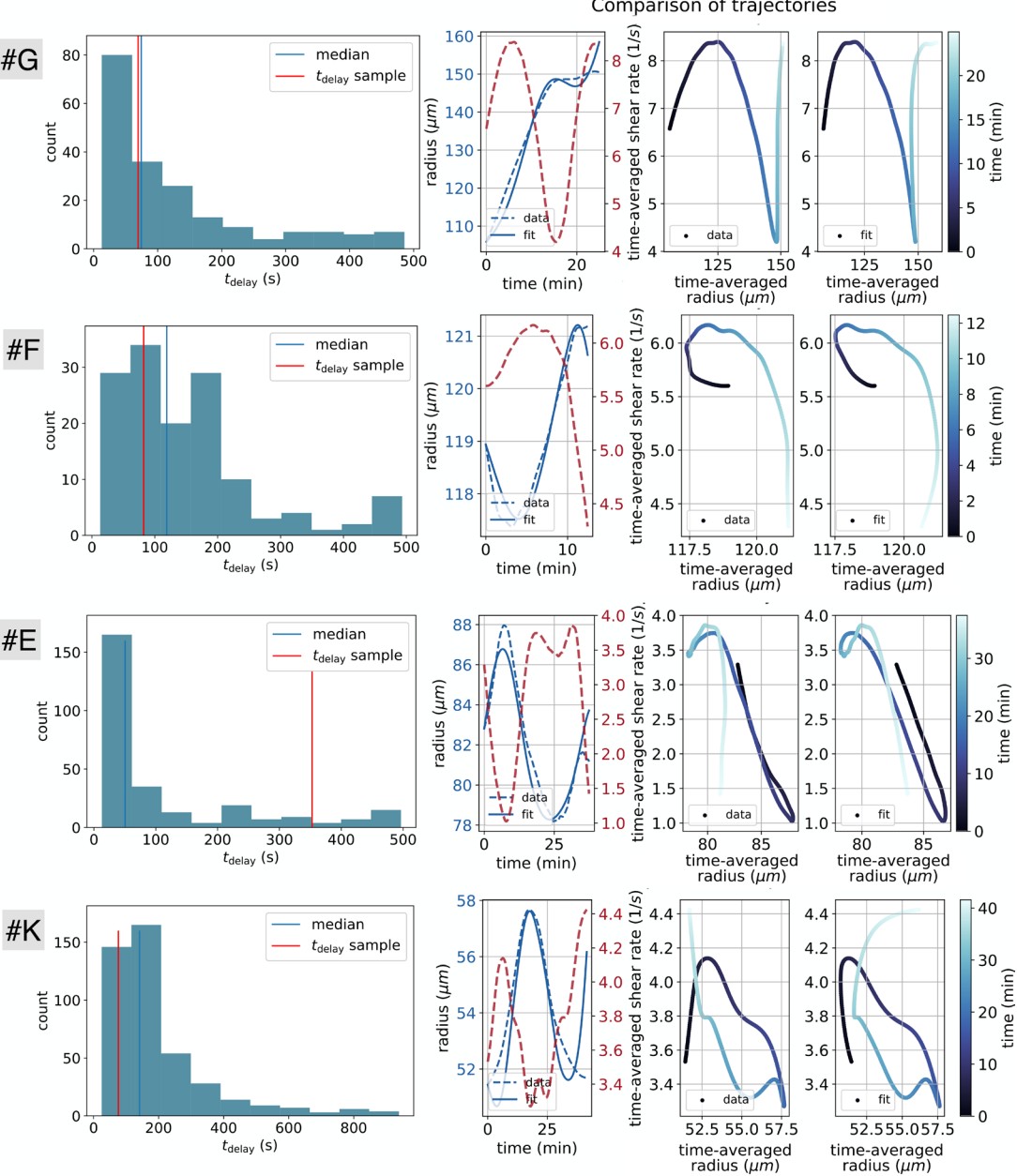

**Appendix 2—figure 4.** Evaluation of all three model parameters $t_{\text{adapt}}$, $t_{\text{delay}}$ and $\tau_0$ from suitable data sets (#E, #F, #G, #K). In the left column the distribution of the obtained time delays using *Equation 1* and *Equation 2* over a distribution of time windows is depicted. To obtain time windows of approximately constant model parameters we performed a fit for every possible time window with the constraints of a reasonable range of fitting parameters and time windows greater than 10 min. The right columns (three graphs) depict a sample of the results of a fitted trajectory, with a given time delay $t_{\text{delay}}$ highlighted on the left hand side graphs as '$t_{\text{delay}}$ sample'. Among the three graphs on the right, the first, shows the fitted radius data as a function of time and the two next, show the data and the fitted result trajectories in the phase space with shear rate and radius.

**Appendix 2—table 1.** Summary of all fitting parameters of sample trajectories depicted in *Appendix 2—figure 4*, right-hand side.

These fits include fitting of $t_\text{delay}$. The fits are done over a range in time $t \in [t_\text{min}, t_\text{max}]$.

| Experiment | $t_{delay}$ in s | $t_{adapt}$ in s | $\tau_0$ in s$^{-1}$ | $t_\text{min} - t_\text{max}$ in min | error $\epsilon_\text{err}$ |
|---|---|---|---|---|---|
| data set G* | 69 ± 3 | 903 ± 190 | 5.4 ± 0.04 | 25–46 | 1.6 % |
| data set F* | 82 ± 3 | 335 ± 8 | 5.7 ± 0.03 | 90–102 | 0.17 % |
| data set E* | 352 ± 7 | 798 ± 3 | 3.07 ± 0.02 | 67–104 | 0.6 % |
| data set K* | 76 ± 4 | 85 ± 4 | 3.63 ± 0.01 | 50–90 | 2.7 % |

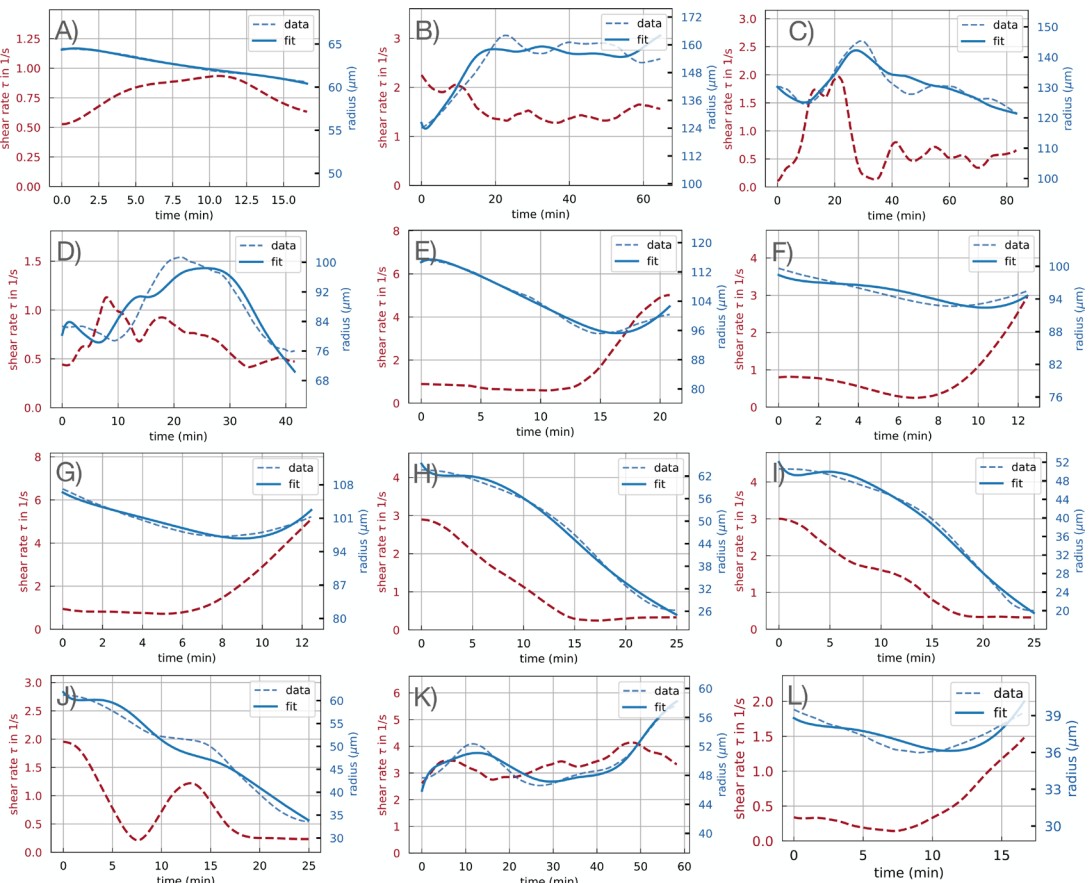

**Appendix 2—figure 5.** Fit results graphical representation of model *Equation 1* and *Equation 2* using a fixed time delay of $t_\text{delay} = 120\,s$ for all 12 close-up data sets on a given time window for each data set. The obtained fit parameters are reported in *Appendix 2—table 2*. All shear rate and radius data presented is time-averaged.

**Appendix 2—table 2.** Summary of all fitting parameters with a fixed time delay of $t_\text{delay} = 120\,\text{s}$ for the 12 close-up data sets.

Note that when a data set name is repeated, it corresponds to fitting results over different time ranges $t \in [t_\text{min}, t_\text{max}]$ for the same data set.

| Experiment | $t_{adapt}$ in s | $\tau_\text{loc}$ in s$^{-1}$ | $t_\text{min} - t_\text{max}$ in min | error $\epsilon_\text{err}$ |
|---|---|---|---|---|
| data set A | 2682 ± 831 | 1.48 ± 0.17 | 0–17 | 0.13 % |
| data set A | 220 ± 10 | 0.44 ± 0.00 | 17–33 | 0.22 % |
| data set A | 246 ± 960 | 1.21 ± 0.59 | 67–83 | 0.68 % |

*Appendix 2—table 2 Continued on next page*

*Appendix 2—table 2 Continued*

| Experiment | $t_{adapt}$ in s | $\tau_{\text{loc}}$ in s$^{-1}$ | $t_{\text{min}} - t_{\text{max}}$ in min | error $\epsilon_{\text{err}}$ |
|---|---|---|---|---|
| data set B | 1195 ± 20 | 1.43 ± 0.00 | 10–75 | 1.4 % |
| data set C | 5948 ± 122 | 0.97 ± 0.00 | 0–83 | 1.9 % |
| data set D | 1531 ± 69 | 0.58 ± 0.01 | 0–33 | 2.0 % |
| data set D | 377 ± 6 | 0.79 ± 0.00 | 8–50 | 3.2 % |
| data set E | 1312 ± 37 | 1.46 ± 0.01 | 0–21 | 0.93 % |
| data set E | 1894 ± 49 | 3.05 ± 0.02 | 58–83 | 0.66 % |
| data set F | 4323 ± 319 | 3.30 ± 0.06 | 0–42 | 1.9 % |
| data set F | 4323 ± 319 | 3.30 ± 0.06 | 0–42 | 1.9 % |
| data set F | 910 ± 24 | 5.08 ± 0.03 | 83–100 | 0.27 % |
| data set G | 1153 ± 51 | 6.07 ± 0.04 | 12–29 | 0.38 % |
| data set G | 1808 ± 322 | 1.00 ± 0.09 | 42–54 | 0.61 % |
| data set G | 1233 ± 54 | 6.68 ± 0.07 | 83–100 | 0.31 % |
| data set H | 355 ± 8 | 2.35 ± 0.02 | 0–25 | 1.4 % |
| data set I | 274 ± 5 | 2.27 ± 0.01 | 0–25 | 1.6 % |
| data set J | 601 ± 132 | 1.41 ± 0.09 | 0–25 | 3.1 % |
| data set K | 278 ± 6 | 3.16 ± 0.00 | 8–67 | 2.0 % |
| data set K | 85 ± 2 | 3.70 ± 0.00 | 45–75 | 0.6 % |
| data set K | 862 ± 119 | 3.40 ± 0.13 | 67–108 | 1.5 % |
| data set K | 123 ± 2 | 4.18 ± 0.00 | 108–133 | 0.53 % |
| data set L | 2553 ± 248 | 0.27 ± 0.01 | 6–23 | 1.3 % |

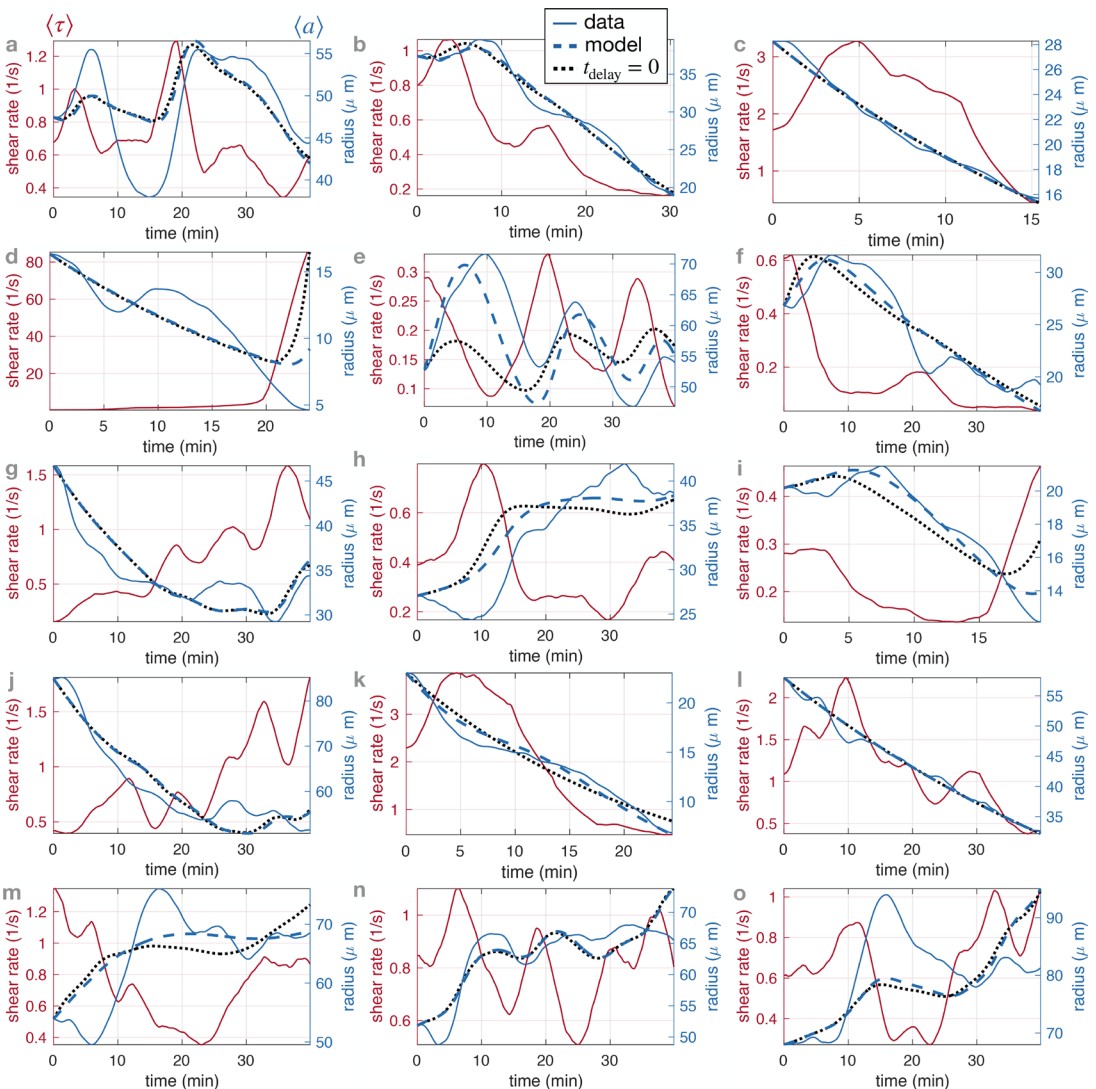

**Appendix 2—figure 6.** Fit results of model *Equation 1* and *Equation 2* for 15 randomly selected veins on specimen #1. The corresponding fitted parameters are reported in *Appendix 2—table 3*. The vein positions correspond to those indicated in *Appendix 1—figure 3*.

**Appendix 2—table 3.** Summary of fitted parameters for a 15 randomly selected veins in the full network *Appendix 2—figure 6*.

$t_{\text{delay}}$ was established with cross-correlation, while $t_{\text{adapt}}$ and $\tau_0$ were obtained through linear least-squares fitting. Error bars correspond to the 95% confidence interval and N.R. corresponds to non relevant points for which the 95% confidence interval yielded error bars as big as the parameters themselves and were, hence, deemed non-relevant.

| Vein index in full network | $t_{delay}$ in s | $t_{adapt}$ in s | $\tau_0$ in s$^{-1}$ | error $\epsilon_{\text{err}}$ | error $\epsilon_{\text{err}}$ with $t_{delay} = 0$ |
|---|---|---|---|---|---|
| a | 24 | 1800 ± 250 | 0.76 ± 0.02 | 8.1% | 7.9% |
| b | 78 | 1520 ± 40 | 0.89 ± 0.02 | 2.6% | 2.9% |
| c | 360 | 1500 ± 150 | N.R. | 1.4% | 1.4% |
| d | 120 | 1800 ± 100 | N.R. | 17% | 23% |

*Appendix 2—table 3 Continued on next page*

*Appendix 2—table 3 Continued*

| Vein index in full network | $t_{delay}$ in s | $t_{adapt}$ in s | $\tau_0$ in s$^{-1}$ | error $\epsilon_{err}$ | error $\epsilon_{err}$ with $t_{delay} = 0$ |
|---|---|---|---|---|---|
| e | 120 | 900 ± 70 | 0.2 ± 0.05 | 7.4% | 11% |
| f | 186 | 2750 ± 100 | 0.38 ± 0.05 | 5.2% | 6.3% |
| g | 42 | 2450 ± 50 | 0.94 ± 0.03 | 5.0% | 5.1% |
| h | 246 | N.R. | 0.26 ± 0.03 | 8% | 11% |
| i | 186 | 870 ± 40 | 0.26 ± 0.01 | 3.3% | 6.9% |
| j | 54 | 2200 ± 100 | 1.1 ± 0.2 | 5.2% | 5.0% |
| k | 288 | 730 ± 50 | 4.0 ± 1.0 | 4.7% | 6.7% |
| l | 108 | 4050 ± 200 | N.R. | 1.1% | 1.2% |
| m | 360 | 10000 ± 3000 | 0.62 ± 0.1 | 6.1% | 8.0% |
| n | 18 | 2050 ± 200 | 0.73 ± 0.02 | 3.7% | 3.8% |
| o | 54 | 7300 ± 3000 | 0.48 ± 0.12 | 6.5% | 6.8% |

## Appendix 3

## Generic ciruit, stability analysis, and model parameters estimation

### Equivalent resistances

Equivalent resistances ($R_{\text{net}}$) in our full network structures are calculated using an algorithm based on Kirchhoff's laws (**Han, 2020**), from the values of $R$ for each vein segments directly evaluated from the data-based network architecture. The algorithm was tested to yield correct results on simple geometries where analytic expressions may be found.

We briefly explain the principle of the algorithm and how to interpret the results of $R/R_{\text{net}}$ on the basis of a few examples in **Appendix 3—figure 1**.

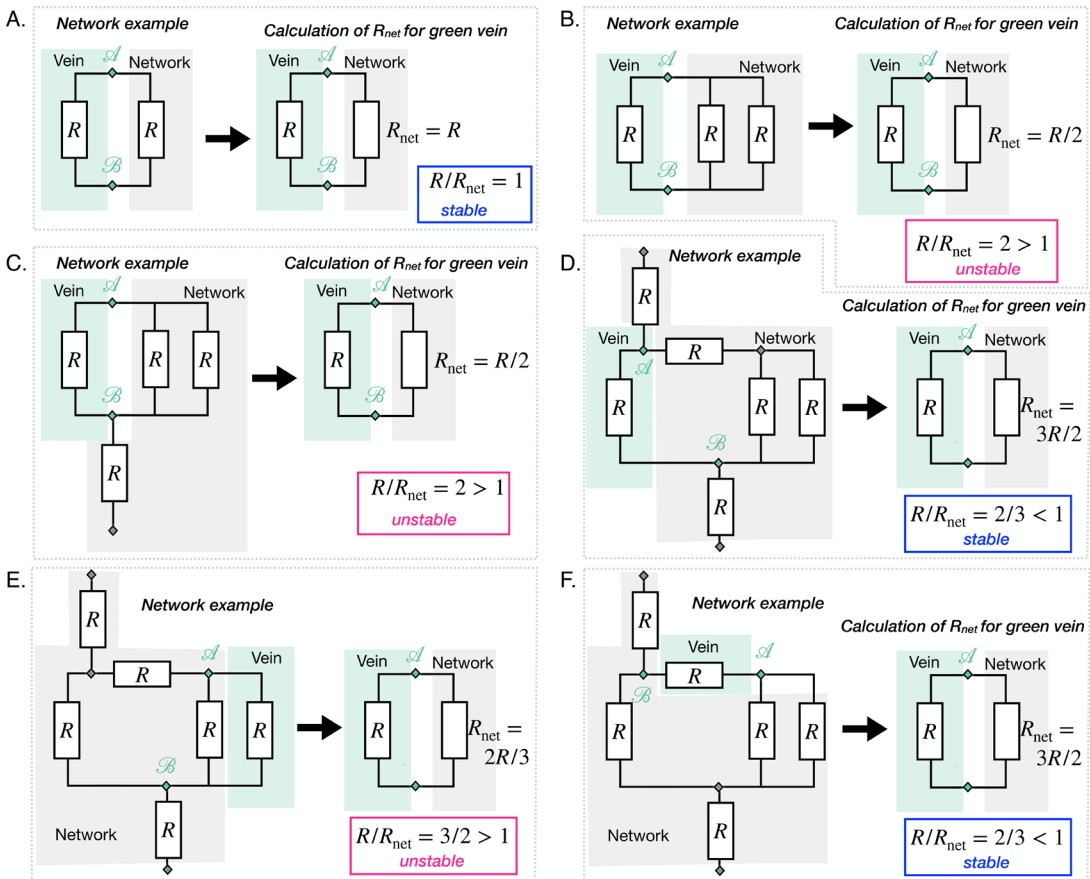

**Appendix 3—figure 1.** Principle of the calculation of $R_{\text{net}}$ on the basis of a few examples. The networks are different in all cases except networks D-E-F are the same. D-E-F differ in which vein is under scrutiny. For each case, equivalent resistances $R_{\text{net}}$ of the rest of the network relative to the vein under scrutiny are calculated via Kirchhoff's laws. The resulting $R_{\text{net}}$ is compared to $R$. When $R > R_{\text{net}}$ (respectively $R < R_{\text{net}}$) the vein is unstable, in pink (respectively unstable, in blue).

In **Appendix 3—figure 1(A)** a simple network consisting of two veins in series is considered. Considering one of these veins as the vein under scrutiny gives simply that the resistance in the rest of the network is $R_{\text{net}} = R$ since it consists only of one vein. Then $R = R_{\text{net}}$ and the vein is a priori stable.

Adding yet another vein in parallel in **Appendix 3—figure 1(B)** modifies the rest of the network. Now it consists in two parallel veins of resistance $R$ and hence $R_{\text{net}} = R/2$ (two resistances in parallel). As a result $R > R_{\text{net}}$ and the vein under scrutiny is a priori unstable.

Adding a dangling end to the network **Appendix 3—figure 1(C)** does not modify the resistance of the network attached to the vein. Hence, the vein under scrutiny is still unstable.

We make a slightly more complex network in **Appendix 3—figure 1(D)** adding another dangling end and another resistance in series. Again the dangling end does not contribute to the calculation

of the equivalent resistance however the vein in series does. We have one vein in series of resistance $R$ with two veins in parallel of resistance $R$. The equivalent resistance is $R_{\text{net}} = R + R/2 = 3R/2 > R$ and the vein is a priori stable now.

Since this network is slightly more complex we can investigate the fate of other veins in that same network, which we do in **Appendix 3—figure 1E, F**. We find that these other veins are unstable or stable. This shows that even in a simple network, the relative resistance is a key measure to discriminate between different veins.

## Generic flow network equivalent circuit

We focus on the generic flow network equivalent circuit as given in **Figure 2A.iii** of the main paper and derive the circuit laws as given in the text – see also **Appendix 4—figure 1A** that recapitulates notations.

Because of Kirchhoff's laws we easily find that $Q_{\text{in}} = -Q_{\text{net}}$. Then we look for the value of the flow rate flowing through the vein of interest $Q$. We see that

$$U = RQ = -R_{\text{net}}(Q + Q_{\text{net}}) = -R_{\text{net}}(Q - Q_{\text{in}}), \tag{A3.1}$$

leading to

$$Q = Q_{\text{in}} \frac{R_{\text{net}}}{R + R_{\text{net}}}. \tag{A3.2}$$

We can then write shear rate in the vein as

$$\tau = \frac{4|Q|}{\pi a^3} = \frac{4|Q_{\text{in}}|}{\pi a^3} \frac{R_{\text{net}}}{R + R_{\text{net}}}. \tag{A3.3}$$

Writing $R = 8\mu L/\pi a^4$, $Q = -Q_{\text{net}}$ and averaging over short timescales, we obtain the shear rate at time $t$

$$\langle \tau \rangle(t) \simeq \frac{4\langle |Q| \rangle}{\pi \langle a \rangle^3} = \frac{4Q_{\text{in}}(\langle a \rangle)}{\pi \langle a \rangle^3} \frac{1}{1 + R(\langle a \rangle)/R_{\text{net}}}. \tag{A3.4}$$

which is exactly **Equation 2** of the main paper.

## Analysis of the feedback system between shear rate and vein radius

In the main text, we have established a set of coupled equations describing the adaptation of veins in a dynamic network. The specific form of these dynamic equations depends on the position of the considered vein within the network. In this section we discuss the stability of a vein fully connected to the network (generic flow network equivalent circuit). Note that other cases (dangling ends, loops, parallel veins) can be easily discussed with similar methodologies.

To simplify the discussion of the fixed points of the dynamical system $(\langle \tau \rangle, \tau_s, \langle a \rangle)$, it is equivalent to study the fixed points of $(\tau_s, \langle a \rangle)$, taking into account **Equation 3** in **Equation 2**. The dynamic system of equations is then given by:

$$\frac{d\langle a \rangle}{dt} = \frac{\langle a \rangle}{t_{\text{adapt}}} \left( \frac{\tau_s^2}{\tau_0^2} - 1 \right), \tag{A3.5}$$

$$\frac{d\tau_s}{dt} = -\frac{1}{t_{\text{delay}}} (\tau_s - \langle \tau \rangle(\langle a \rangle)). \tag{A3.6}$$

where $\langle \tau \rangle$ is a function of the tube diameter $\langle a \rangle$:

$$\langle \tau \rangle(\langle a \rangle) = \frac{4Q_{\text{net}}(\langle a \rangle)}{\pi \langle a \rangle^3} \frac{1}{1 + R(\langle a \rangle)/R_{\text{net}}} \simeq \frac{32L\epsilon}{T} \frac{\langle a \rangle^3}{\langle a \rangle^4 + 8L\mu/(\pi R_{\text{net}})}, \tag{A3.7}$$

since $Q_{\text{net}} = 8\pi L\epsilon\langle a \rangle^2/T$ where $\epsilon$ is the characteristic contraction percentage of the vein (dimensionless). Plotting the nullclines of **Equations A3.5 and A3.6** in the $(\tau_s, \langle a \rangle)$ space, we observe one, two or no intersections of the nullclines, which correspond to fixed points of the system, depending on the

physical parameters. In particular, there is a fixed point corresponding to a vanishing vein in . In the following, we will investigate the conditions for the existence and the stability of these fixed points.

## Existence of the fixed points

The dynamical system has more than one fixed point if the nullclines intersect. As depicted in **Figure 2B**, this is the case if $\max_{\langle a \rangle}(\langle \tau \rangle (\langle a \rangle)) \geq \tau_0$.

The maximum of $\tau(\langle a \rangle)$ is determined by:

$$\frac{\partial \langle \tau \rangle}{\partial \langle a \rangle} = \frac{32 L \epsilon}{T} \frac{3 \langle a \rangle^2 (\langle a \rangle^4 + 8 \mu L/(\pi R_{\text{net}})) - 4 \langle a \rangle^6}{(\langle a \rangle^4 + 8 L \mu/(\pi R_{\text{net}}))^2} = 0, \Leftrightarrow \langle a \rangle = \left( \frac{21 \mu L}{R_{\text{net}} \pi} \right)^{(1/4)}. \tag{A3.8}$$

Inserting this in $\langle \tau \rangle$, we get the condition

$$\frac{32 \pi \epsilon R_{\text{net}}}{29 \mu} \left( \frac{21 \mu L}{R_{\text{net}} \pi} \right)^{(3/4)} \geq \tau_0, \tag{A3.9}$$

where equality corresponds to one additional fixed point and strict inequality corresponds to two additional fixed points.

## Linear stability of the feedback-system

The dynamical system defined in **Equations A3.5 and A3.6** has up to three fixed points. To analyze the stability of those fixed points we use linear stability analysis (**Strogatz, 1994**; **Argyris et al., 2017**). The first fixed point is at $(\tau_s = 0, \langle a \rangle = 0)$, the other two are defined by $(\tau_s = \tau_0, \langle a \rangle = r_{0,\pm})$, where $r_{0,\pm}$ are the real positive solutions of the equation $\tau_0 = \langle \tau \rangle (r_0)$. To analyze the stability of those fixed points, we calculate the Jacobi matrices $J$ at each location:

$$J(\tau_s, \langle a \rangle) = \begin{bmatrix} \frac{-1 + \frac{\tau_s^2}{\tau_0^2}}{t_{\text{adapt}}} & \frac{2 \langle a \rangle \tau_s}{t_{\text{adapt}} \tau_0^2} \\ -\frac{\frac{128 L \epsilon \langle a \rangle^6}{T \left( \frac{8 L \mu}{R_{\text{net}} \pi} + \langle a \rangle^4 \right)^2} - \frac{96 L \epsilon \langle a \rangle^2}{T \left( \frac{8 L \mu}{R_{\text{net}} \pi} + \langle a \rangle^4 \right)}}{t_{\text{delay}}} & -\frac{1}{t_{\text{delay}}} \end{bmatrix}; \quad J(0,0) = \begin{bmatrix} -\frac{1}{t_{\text{adapt}}} & 0 \\ 0 & -\frac{1}{t_{\text{delay}}} \end{bmatrix}. \tag{A3.10}$$

For $(0,0)$ the eigenvalues can be read off from the Jacobi matrix as $\lambda_{0,1} = -\frac{1}{t_{\text{adapt}}}$ and $\lambda_{0,2} = -\frac{1}{t_{\text{delay}}}$. Consequently, the fixed point is stable, as all model parameters are positive. The two other fixed points, as mentioned above depend on the root $\langle a \rangle_0$ of $\tau_0 = \langle \tau \rangle (\langle a \rangle_0)$.

The stability of those fixed points is therefore conditional on the value of these roots. To gain insight on the stability of the fixed points we look at the two extreme cases of either small or large tube radii (as specified below). We will then extend our insight to intermediate tube radii.

- $\langle a \rangle \to 0$: In the case of a small tube radius, we can expand **Equation A3.7** in orders of $\langle a \rangle$. Expanding up to the first non-trivial order gives:

$$\langle \tau \rangle (\langle a \rangle) \simeq \frac{4 R_{\text{net}} \epsilon \pi \langle a \rangle^3}{T \mu}. \tag{A3.11}$$

There are thus two fixed points at $(\tau_{s,1} = \tau_0, \langle a \rangle_1 = \frac{1}{2} \sqrt[3]{\frac{2 T \mu \tau_0}{R_{\text{net}} \epsilon \pi}})$ and at $(\tau_{s,2} = 0, \langle a \rangle_2 = 0)$. The resulting Jacobian at $(\tau_s, \langle a \rangle)$ is

$$J(\tau_s, \langle a \rangle) = \begin{bmatrix} -\frac{1}{t_{\text{adapt}}} + \frac{\tau_s^2}{t_{\text{adapt}} \tau_0^2} & \frac{2 \langle a \rangle \tau_s}{t_{\text{adapt}} \tau_0^2} \\ \frac{12 R_{\text{net}} \epsilon \pi \langle a \rangle^2}{T \mu t_{\text{delay}}} & -\frac{1}{t_{\text{delay}}} \end{bmatrix} \tag{A3.12}$$

The eigenvalues of $J$ at $(\tau_{s,1} = \tau_0, \langle a \rangle_1 = \frac{1}{2} \sqrt[3]{\frac{2 T \mu \tau_0}{R_{\text{net}} \epsilon \pi}})$ are given by $\lambda_{1,+} = \frac{-\sqrt{t_{\text{adapt}}} + \sqrt{t_{\text{adapt}} + 24 t_{\text{delay}}}}{2 \sqrt{t_{\text{adapt}} t_{\text{delay}}}}$, $\lambda_{1,-} = -\frac{\sqrt{t_{\text{adapt}}} + \sqrt{t_{\text{adapt}} + 24 t_{\text{delay}}}}{2 \sqrt{t_{\text{adapt}} t_{\text{delay}}}}$. As all model parameters are positive it is easy to see that $\lambda_{1,+} > 0$ and $\lambda_{1,-} < 0$. Consequently, the fixed point is a saddle point. For the second fixed point $(0,0)$ we recover the same eigenvalues as in the general case, $\lambda_{0,1} = -\frac{1}{t_{\text{adapt}}}$ and $\lambda_{0,2} = -\frac{1}{t_{\text{delay}}}$, which are both negative and indicate a stable fixed point.

- $\langle a \rangle \to \infty$: In the case of a large tube radius, the shear rate simplifies to $\langle \tau \rangle (\langle a \rangle) = \frac{32L\epsilon}{T} \frac{1}{\langle a \rangle}$ and we find only one fixed point at $(\tau_s = \tau_0, \langle a \rangle = \frac{32L\epsilon}{T\tau_0})$. The Jacobian at this fixed point is

$$J(f_3) = \begin{bmatrix} 0 & \frac{64L\epsilon}{Tt_{\mathrm{adapt}}\tau_0^2} \\ -\frac{T\tau_0^2}{32L\epsilon t_{\mathrm{delay}}} & -\frac{1}{t_{\mathrm{delay}}} \end{bmatrix}, \tag{A3.13}$$

with the eigenvalues

$$\lambda_{2,\pm} = \frac{-\sqrt{t_{\mathrm{adapt}}} \pm \sqrt{t_{\mathrm{adapt}} - 8t_{\mathrm{delay}}}}{2\sqrt{t_{\mathrm{adapt}}t_{\mathrm{delay}}}} \tag{A3.14}$$

We now have to differentiate two cases. The first one is $t_{\mathrm{adapt}} > 8t_{\mathrm{delay}}$. Then $\lambda_{2,\pm} < 0$ and the fixed point is stable. For the case $t_{\mathrm{adapt}} < 8t_{\mathrm{delay}}$, we have $\Re(\lambda_{2,\pm}) < 0$, $\Im(\lambda_{2,\pm}) \neq 0$ and the fixed point is a stable spiral, which introduces an additional rotation to the system's trajectories. To investigate the direction of the rotation of this hypothetical spiral, one can look at the sign of *Equation A3.5* for positive displacements $\delta$ along the shear rate axis. We find that

$$\frac{d\langle a \rangle}{dt}\Big|_{(\tau_s=\tau_0+\delta, \langle a \rangle=\langle a \rangle_1)} = \frac{\langle a \rangle_1}{t_{\mathrm{adapt}}} \frac{\tau_0\delta + \delta^2}{\tau_0^2} > 0 \tag{A3.15}$$

and therefore the spiral rotates in the clockwise direction.

In summary, our stability analysis has shown that the system has up to three fixed points. Two of them are stable and separated by a saddle point. The qualitative stability in the limiting case is also valid for intermediate tube radii, as the stability of a fixed point only changes when two fixed points collide, which is only the case at the bifurcation point (when $\langle a \rangle_0$ corresponds to the maximum of $\tau(\langle a \rangle)$) (*Strogatz, 1994*).

## Appendix 4

## Equivalent vein flow circuit models for other network topologies of a vein

### Dangling ends

We investigate dangling ends as shown in the network depicted in **Appendix 4—figure 1B**. Here we consider that a dangling end is connected to the rest of the network at a node where pressure is $\langle P \rangle$. Since the vein is a dangling end, the only flow flowing through the vein is that generated by peristaltic contractions $Q = Q_{\text{in}}$. Then the shear rate through the dangling vein is simply

$$|\tau_{\text{vein}}(\langle a \rangle)| = \frac{4}{\pi \langle a \rangle^3}|Q| = \frac{32 L \epsilon}{\langle a \rangle T}, \tag{A4.1}$$

again since $Q \simeq 8\pi L \epsilon \langle a \rangle^2 / T$ where $\epsilon$ is the relative contraction amplitude of the vein. We see that this shear rate is the same as in the limiting case $\langle a \rangle \to \infty$ for the generic circuit. Consequently, this expression does not give rise to any stable non-zero fixed point. Hence, the vein either vanishes or grows indefinitely – see **Appendix 4—figure 1B**. The crossover between the two regimes occurs for a critical radius

$$a_c = \frac{32 L \epsilon}{T \tau_0} \simeq \frac{32 L \mu \epsilon}{T(\mu \tau - \langle P - P_0 \rangle)} \tag{A4.2}$$

For $\langle a \rangle \geq a_c$, the vein grows, otherwise it vanishes. In other words, when $\langle P \rangle$ is large, the vein is likely to grow, whereas it vanishes when $\langle P \rangle$ is small.

### Parallel veins

We investigate parallel veins as shown in the network depicted in **Appendix 4—figure 1D.i**. The flow rate $Q$ splits up into two currents in the two vein branches such that, according to Kirchhoff's laws

$$U = Q_1 R_1 = Q_2 R_2 = -Q R_{\text{net}} \tag{A4.3}$$

where $Q_1$ (resp. $Q_2$) is the flow pervading vein 1 (resp. 2). Since incoming flow rates have to sum up to zero at nodes, we also have

$$Q + Q_{\text{in},1} + Q_{\text{in},2} = Q_1 + Q_2, \tag{A4.4}$$

where $Q_{\text{in},i}$ is the net flow generated by each vein indexed by $i$ over long times. The shear rates inside each of the veins are

$$\tau_i = \frac{4}{\pi a_i^3} Q_i = \frac{4}{\pi a_i^3} \frac{Q R_{\text{net}}}{R_i}. \tag{A4.5}$$

After standard calculation steps, we obtain

$$\tau_i = \frac{4}{\pi a_i^3} \frac{1}{R_i} \frac{Q_1 + Q_2}{\frac{1}{R_1} + \frac{1}{R_2} + \frac{1}{R_{\text{net}}}}. \tag{A4.6}$$

Now we remark that we can define for each of the parallel veins the resistance of the rest of the network from the viewpoint of each vein. In fact, for $R_1$, the rest of the network is comprised of $R_{\text{net}}$ and $R_2$ in parallel. Hence from the single vein perspective of $R_1$, we may define the equivalent resistance of the rest of the network $R_{\text{net},1}$, such that $\frac{1}{R_{\text{net},1}} = \frac{1}{R_{\text{net}}} + \frac{1}{R_2}$. Similarly for $R_2$. As a result we see that

$$\tau_i = \frac{4}{\pi a_i^3} \frac{Q_{\text{in},1} + Q_{\text{in},2}}{1 + \frac{R_i}{R_{\text{net},i}}}. \tag{A4.7}$$

We thus coherently find that the relative resistance $R_1 / R_{\text{net},1}$ will determine the magnitude of $\tau_i$ and hence its potential stability.

In terms of the respective vein radii, we have

$$\tau_{1/2}(a_1, a_2) = \frac{32 a_{1/2} R_{\text{net}} \epsilon L}{T} \frac{a_1^2 + a_2^2}{\left(8 \eta L + R_{\text{net}} \pi \left(a_1^4 + a_2^4\right)\right)} = 4 a_{1/2} R_{\text{net}} \frac{Q_1 + Q_2}{\left(8 \eta L + R_{\text{net}} \pi \left(a_1^4 + a_2^4\right)\right)}. \tag{A4.8}$$

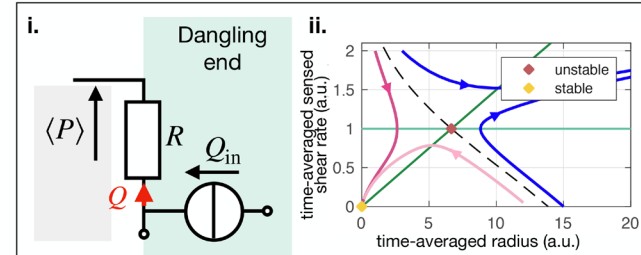

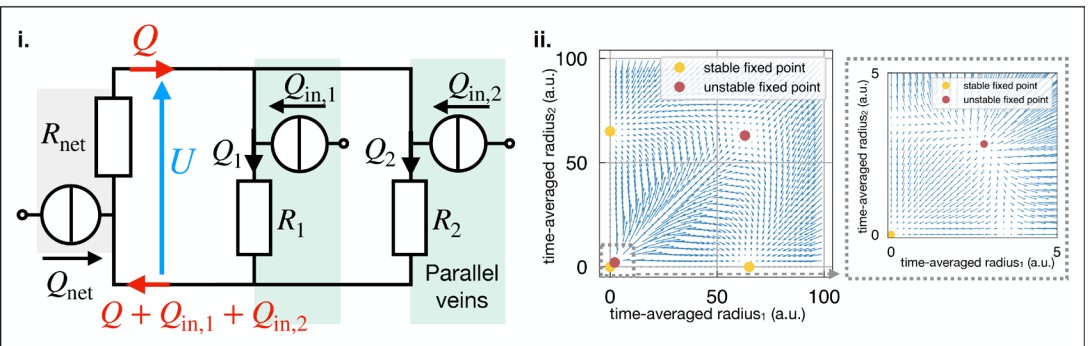

**Appendix 4—figure 1.** All circuits discussed in the text. (**A**) General circuit for a vein connected to the rest of the network. (**B**) Dangling ends; (**i**) Electric circuit and notations and (**ii**) stability diagram in the shear radius space. (**C**) Loops; (**i**) Electric circuit and notations (**ii**) Stability diagram in the shear radius space for vein 2 and (**iii**) for veins 1 and 3. (**D**) Parallel Veins; (**i**) Electric circuit and notations (**ii**) Stability diagram in the radius - radius space.

This expression allows us to draw a stability diagram in the $a_1$, $a_2$ space, see **Appendix 4—figure 1D** We find that there are three stable fixed points $(0,0)$, $(0, a_c)$, $(a_c, 0)$ and $(a_c, a_c)$ is an unstable fixed point. Note that this diagram is very similar to the one obtained by *Hacking et al., 1996*. As a consequence of $(a_c, a_c)$ being unstable, one vein always shrinks in favor of the other.

We check that the instability of the parallel veins is consistent with the predictions that we could make with the resistance ratio. According to the stability diagram, one vein say of index 1 shrinks in favor of the vein with index 2 if and only if $R_1 > R_2$. In that case, we also have $\frac{R_1}{R_{\text{net},1}} > \frac{R_2}{R_{\text{net},2}}$. In fact we have the series of inequalities

$$
\begin{aligned}
\frac{R_1}{R_{\text{net},1}} &> \frac{R_2}{R_{\text{net},2}}, \\
\frac{R_1(R_2+R_{\text{net}})}{R_{\text{net}}R_2} &> \frac{R_2(R_1+R_{\text{net}})}{R_1 R_{\text{net}}}, \\
R_1^2 R_{\text{net}}(R_2 + R_{\text{net}}) &> R_2(R_1 + R_{\text{net})}R_{\text{net}}R_2, \\
R_1^2(R_2 + R_{\text{net}}) &> R_2^2(R_1 + R_{\text{net}}), \\
R_1^2 R_2 + R_1^2 R_{\text{net}} &> R_2^2 R_1 + R_2^2 R_{\text{net}}, \\
R_1(R_1 R_2 + R_1 R_{\text{net}}) &> R_2(R_1 R_2 + R_2 R_{\text{net}}),
\end{aligned}
\tag{A4.9}
$$

which is indeed the case since $R_1 > R_2$. Since $\frac{R_1}{R_{\text{net},1}} > \frac{R_2}{R_{\text{net},2}}$, we can read off directly that vein 1 is more likely to shrink than vein 2, since the energetic gain to shrink vein 1 is bigger than that to shrink vein 2.

## Loops

We investigate loops as shown in the network depicted in *Appendix 4—figure 1C*. Kirchhoff laws impose $Q_{\text{net}} = -3Q_{\text{in}}$ and

$$Q = -Q_{\text{in}} \frac{6R}{3R + R_{\text{net}}} \tag{A4.10}$$

such that the shear rate through each of the veins writes as

$$
\begin{cases}
|\tau_1| &= \dfrac{4}{\pi \langle a \rangle^3} Q \dfrac{|3R - R_{\text{net}}|}{3R + R_{\text{net}}}, \\[2mm]
|\tau_2| &= \dfrac{4}{\pi \langle a \rangle^3} Q_{\text{in}} \dfrac{2R_{\text{net}}}{3R + R_{\text{net}}}, \\[2mm]
|\tau_3| &= \dfrac{4}{\pi \langle a \rangle^3} Q_{\text{in}} \dfrac{|3R + 3R_{\text{net}}|}{3R + R_{\text{net}}}.
\end{cases}
\tag{A4.11}
$$

Since $R = \frac{8\mu L}{\pi \langle a \rangle^4}$, we find that

$$
\begin{cases}
|\tau_1| &= \dfrac{4}{\pi \langle a \rangle^3} Q \dfrac{|24\eta L - R_{\text{net}}\pi \langle a \rangle^4|}{24\eta L + R_{\text{net}}\pi \langle a \rangle^4}, \\[2mm]
|\tau_2| &= 4Q_{\text{in}} \langle a \rangle \dfrac{2R_{\text{net}}}{24\eta L + R_{\text{net}}\pi \langle a \rangle^4}, \\[2mm]
|\tau_3| &= \dfrac{4}{\pi \langle a \rangle^3} Q_{\text{in}} \dfrac{|24\eta L + 3R_{\text{net}}\pi \langle a \rangle^4|}{24\eta L + R_{\text{net}}\pi \langle a \rangle^4}.
\end{cases}
\tag{A4.12}
$$

From these equations, we see that vein 2 behaves just like the generic vein (of the generic circuit). If $\langle a \rangle$ is small, most probably that vein will disappear – see *Appendix 4—figure 1C*. In general veins 1 and 3 only have one stable fixed point, and essentially have a bounded size – see *Appendix 4—figure 1C* (as long as 2 has not vanished yet).

