## [Editor Report]

This fundamental work elucidates the physical forces that shape rearrangement of vascular networks using the model system slime mold. The authors provide compelling theoretical and experimental evidence to demonstrate how the fluid flow locally deforms the veins and ultimately dictates a global remodelling of network architecture. This profound and experimentally validated theory will be of great interest for many readers working on dynamically rearranging networks, which are ubiquitous in living systems.

---

## [Decision Letter]

**Decision letter after peer review:**

Thank you for submitting your article "Network architecture determines vein fate during spontaneous reorganization, with a time delay" for consideration by *eLife*. Your article has been reviewed by 2 peer reviewers, and the evaluation has been overseen by a Reviewing Editor and Aleksandra Walczak as the Senior Editor. The reviewers have opted to remain anonymous.

Essential revisions:

(1) Please clarify the distinction between shear rate and shear stress.

(2) Please edit the manuscript for clarity throughout.

(3) Please clarify the relationship between the adaptation time scale and target shear rate in the model fit.

(4) Please clarify whether the direction of the time delay was assumed or derived.

(5) Please better connect the results to the literature. What is novel and what was previously recognised should emerge more clearly.

(6) Explain more clearly the feedback mechanism based on the actin networks' response to shear stress, for example by considering the suggestion made by referee 2 to move Figure A.1 to the main text and more clearly explain the essence and the novelty of that mechanism there so that the reader doesn't need to fully parse Appendix A.

(7) Please provide a clear description of how Δ P_net (Figure 3C and elsewhere) is calculated from data. Is it just the product of R/R_net (Figure 3A) and Q_net (Figure 3B)? Or?

(8) Please explain how R/Rnet is calculated – the text in the Methods is quite vague and refers to ref. 70 but the citation appears incomplete.

*Reviewer #1 (Recommendations for the authors):*

I will admit that I found that paper hard going for a variety of reasons documented below. I think the authors may be writing an important paper but the writing is sufficiently vague in places that I can't unravel (a) was some of this in fact already recognized several decades ago (reference 27), even if not fully appreciated by the community? (b) how much of the model is really a "force balance" in contrast to a plausible model based on approximate ideas? (c) the time delay, while motivated by the data, is apparently not understood, even in terms of the origin of the magnitude of the time scale, which is fine since the idea seems new but again the authors seem to obfuscate some of the understanding and interpretation. Finally, in some places, the authors seem to confuse shear stress and shear rate, and in other places seem to imply that pressure drops do not change significantly but shear rates do change significantly, yet they also state that pressure drop is proportional to shear rate. Perhaps the authors will write in response that I have misunderstood but then I can only say that their writing is unclear. Also, several figure captions seem to lack important information, e.g., what do colors mean, and in other cases, the choice of the color scale, combined with my inability to appreciate shades of darkness means I can't understand what the authors are trying to convey in a figure.

Other remarks:

(1) The title is much too general. The opening sentence of the abstract ("Vascular networks continuously reorganize their morphology by growing new or shrinking existing veins to optimize function.") similarly sounds too general.

(2) The abstract seems not so clearly written – "dependent on shear rate" is not inconsistent with being dependent on shear rate with a time delay. Indeed, the authors even write on p. 1: "shear rate indeed feeds back on vein radii, interestingly with a time delay" (though shear rate values may cause veins either to grow or to shrink).

(3) p. 2: it looks like the authors are using Δ t and t_delay to denote the same time delay.

(4) The authors write that the time delay "consistent with measured data on the contractile response of active fibers" – but this is so vague that I have no idea what they are talking about and whether or I not should think that there is a relation.

(5) In many places the text with long paragraphs would benefit from paragraph breaks.

(6) There is something wrong with equation (1) and the ensuing discussion: "taus is shear stress sensed by the vessel wall, tau_loc is a shear rate reference" – but since these appear as a ratio in equation (1), and are compared to unity, they must have the same dimensions, which they do not. A shear stress is not the same as a shear rate. So something is wrong. A few lines later the authors write Δ p = mu tau_loc, so they really mean tau_loc is a shear rate. Nevertheless, the authors then write "It is therefore related to the network's architecture as pressure senses the entire network's morphology" but this is incorrect since (as their equation indicated) tau_loc only knows about the local pressure drop, which is distinct from the actual values of the pressure (I believe all the arguments in the paper assume very reasonably that the liquid is incompressible).

(7) The authors continue with the confusion of shear stress and shear rate in equation (2): "the shear sensed by a vein wall taus and fluid shear rate " – I have no idea what this could mean. The fluid shear rate in the vessel is tau so what do they mean when they write "the shear sensed by a vein wall"? Note: I think the confusion is that the authors wrote "shear stress" above but they really mean "shear rate"?

(8) Then when I read some of this the authors also write (P. 3, column 1) "In the force balance perspective, pressure, circumferential stress and active stresses are typically constant during an adaptation event [8] and vary smoothly throughout the network (Appendix Figure A.2). In contrast, shear stress (proportional to shear rate) is observed to change significantly over time and space throughout the network, and is therefore expected to contribute significantly to long time scale dynamics" but this seems inconsistent with their statement in the next column that Δ p = mu tau_loc.

Of course, just after they write "Thus, tau_loc, which also includes a contribution from active stress generation," – so I think the big jumble here is that the authors are not helping a reader (or themselves) unravel rates from stresses, but surely this is important since the cell sensing mechanisms may be different for a rate versus a stress.

(9) Figure 2B: monotonous → monotonic.

(10) What do all the colors correspond to the panels in Figure 2B? The colors seem to be defined in the main text but they should also be identified in the figure caption.

(11) p. 4: "non-monotonous shear rate decrease (light pink Figure 2-B.i). Without the time delay (3), instantaneous shear sensing as …  (similarly as in e.g. Ref. [27]) can produce neither circling nor non-monotonous …" – so have previous authors highlighted some of these points (ref 27 is from 1996)? On p. 5, the authors similarly note results already identified in ref. 27. The authors are unclear about what might have already been demonstrated in advance of their work. Moreover, in several places, they write "Our force balance model challenges previous concepts showing that vein fate is not only determined through shear magnitude but also through network architecture dependent parameters." But again this seems to be disingenuous to some other authors (e.g. ref. 27 if I understand the paper) who have already made this point?

(12) "Overall, we find a remarkable agreement between t and data (Figure 2-D…" – but there is no figure 2d?)

(13) The sensitivity of my eyes to color is not good enough to make sense of Figure 3.

(14) p. 4-5: I do not understand why Q_net, which is dimensional should be, on its own, a relevant parameter.

*Reviewer #2 (Recommendations for the authors):*

I enjoyed reading this manuscript, which describes some impressive experiments and analysis, together with what seems to be some very original ideas about how vascular adaptation can work in P. polycepharum. The paper is rather dense, though, and although this is of course a question of style, I personally would think the many interesting results might more easily be appreciated by a broader audience if they were packaged as at least two separate papers – in particular, the detailed force-balance model (in Appendix A) based on the unique property of actin networks is very interesting and would seem to stand alone as a paper on its own, while its main role in the main text is to furnish one of the three equations (Equation 1) needed for the analysis of experimental data. I could conceive that most of the story of the main text could then be told in a paper that doesn't go into the details of Appendix A but provides Equation 1 as a phenomenological equation that satisfies Murray's law. In any case, the authors have made the choice to package it in this way, and so I will try to provide constructive suggestions below to improve readability/accessibility to the broad readership of *eLife*.

Overall, I have confidence in the authors' technical abilities (both experimental and theoretical), but in many places, the descriptions were less than transparent (and in some cases inaccurate; see below). Apart from those issues of presentation, I have identified only two major concerns – one regarding model fits the data, and another regarding the interpretation of the observed time delay – which I believe need to be addressed before publication:

1. The description of the model fits gives the impression that the adaptation timescale (t_adapt) and the target shear rate (tau_loc) were allowed to vary freely when fitting experimental data. But according to the force-balance model of Appendix A, the two are not independent, but rather are related in a specific way (via Equation 5 of main text). So allowing these two parameters to vary freely seems to implicitly assume that the other parameters appearing in Equation 5 (specifically, viscosity mu and line-friction coefficient γ) are changing across data sets. I am not familiar enough with the system to know whether this is a reasonable assumption, but I was under the impression for the Appendix A text that mu and γ were both assumed constant everywhere, and in any case, some discussion of this point (i.e. what is happening with the effectively assumed differences in mu and γ across these fits) seems necessary.

2. In the analysis and interpretation of the time delay, it seems the sign of the time delay was assumed to be positive (i.e. radius changes following shear-rate changes in time, rather than the other way around). For example, the cross-correlation analysis in Appendix D shows the cross-correlation only in the positive time direction. This assumption is also implicit in the statement (p.2) that "Clockwise circling corresponds to an in/decrease in shear followed by an in/decrease in vein radius pointing to a shear rate feedback on local vein adaptation." This is of course the expected direction of causality for shear-rate feedback, but the presented data and analyses in my view do not conclusively establish this direction of causality. This is because both the dynamics of the (time-averaged) shear rate and radius tend to be oscillatory, and with oscillatory signals, it is possible that the time delay associated with the cross-correlation peak can become positive or negative (and likewise oscillations in the phase plane can be CW or CCW), depending on the phase delay. So it does not seem so straightforward to establish the direction of causality based on the phase-plane rotation direction or the kinds of cross-correlation data presented in Appendix D. I do understand, however, that shear-rate feedback is a very natural expectation as a vein radius adaptation mechanism, especially given that Murray's law has already been established in the system. So perhaps a good choice could be to simply state something to that effect, and say that the direction of causality is an assumption, rather than a result of the analysis?

Apart from the above, my suggestions are mainly at the level of presentation, but I note that overall, the number of confusing or inaccurate statements was rather large, so the list below is not exhaustive and the authors should consider careful proofreading (perhaps with the aid of non-author colleagues) before finalizing the paper for publication.

p. 2, last paragraph of the left column: it is stated that the shear rate has twice the frequency of the radius. This statement is confusing because there are many nonlinear transformations of sinusoids that do not lead to frequency doubling. As I understand it, the frequency doubling is rather due to the fact that what is considered is the *absolute* (i.e. unsinged) shear rate, which is indeed a nonlinear transformation, but a very specific one. This means it is the absolute-flow-rate factor (|Q|) coming into the given expression for the shear rate, rather than the radius itself (which appears as a^3^) within that expression that is actually controlling the frequency doubling.

p.3, right column: where Figure D.3 is cited, that figure does not appear to provide relevant data for this context.

p.3, I found t_ave, appearing in Equation 3 as a time delay, rather confusing. In the Appendix, it is explained that this is a data average, but the data averaging procedure explained in Materials and Methods seems to use a window centered on the current time point, so it is unclear how data averaging imposes a delay. This is especially confusing because there are already two other timescales, t_delay and t_adapt, that could contribute to the delay. This should be clarified.

p.9 just under Equation (6): the term 'resistance' is used to describe R in Equation (6), but that equation and the expression given for R here imply that R here actually is conductance (i.e. inverse of resistance). I noticed this error also elsewhere, although in most places R does indeed correspond to resistance. Please check all instances where R appears.

p.15, near the top of the page: t_delay is used twice, where correctly I believe what is being discussed is in fact t_adapt.

p.18, under Equation B13: two expressions are given for λ_+. I guess this is a typo (the second one should be λ_-?).

p.20-25: Appendix D is very dense, and is lacking a narrative text. Currently, many important statements/derivations occur within the legends of the figures provided within this section, for example, the expression for how Q is computed from the vein radius change I believe only appears in the legend of Figure D.1 and I could not find it anywhere else in the main text or the text narrative of the appendices. Consider including this in the main text.

---

## [Author Response]

Reviewer #1 (Recommendations for the authors):I will admit that I found that paper hard going for a variety of reasons documented below. I think the authors may be writing an important paper but the writing is sufficiently vague in places that I can't unravel (a) was some of this in fact already recognized several decades ago (reference 27), even if not fully appreciated by the community? (b) how much of the model is really a "force balance" in contrast to a plausible model based on approximate ideas? (c) the time delay, while motivated by the data, is apparently not understood, even in terms of the origin of the magnitude of the time scale, which is fine since the idea seems new but again the authors seem to obfuscate some of the understanding and interpretation. Finally, in some places, the authors seem to confuse shear stress and shear rate, and in other places seem to imply that pressure drops do not change significantly but shear rates do change significantly, yet they also state that pressure drop is proportional to shear rate. Perhaps the authors will write in response that I have misunderstood but then I can only say that their writing is unclear. Also, several figure captions seem to lack important information, e.g., what do colors mean, and in other cases, the choice of the color scale, combined with my inability to appreciate shades of darkness means I can't understand what the authors are trying to convey in a figure.Other remarks:(1) The title is much too general. The opening sentence of the abstract ("Vascular networks continuously reorganize their morphology by growing new or shrinking existing veins to optimize function.") similarly sounds too general.

We have changed the title and the introduction to be more specific.

(2) The abstract seems not so clearly written – "dependent on shear rate" is not inconsistent with being dependent on shear rate with a time delay. Indeed, the authors even write on p. 1: "shear rate indeed feeds back on vein radii, interestingly with a time delay" (though shear rate values may cause veins either to grow or to shrink).

We have clarified the abstract and now avoid inconsistent sentences.

(3) p. 2: it looks like the authors are using Δ t and t_delay to denote the same time delay.

We apologize for this initially confusing notation and now just use *t*_delay_.

(4) The authors write that the time delay "consistent with measured data on the contractile response of active fibers" – but this is so vague that I have no idea what they are talking about and whether or I not should think that there is a relation.

We have clarified this statement. The statement now reads “Measured data on the contractile response of cross-linked fibers [44-46] exhibits a time delay of about 1 − 30 s for in vitro gels.”

(5) In many places the text with long paragraphs would benefit from paragraph breaks.

We have edited the paper significantly such that each paragraph opens up to an easy message and reads in a simpler, short and efficient way. We now include numerous paragraph breaks.

(6) There is something wrong with equation (1) and the ensuing discussion: "taus is shear stress sensed by the vessel wall, tau_loc is a shear rate reference" – but since these appear as a ratio in equation (1), and are compared to unity, they must have the same dimensions, which they do not. A shear stress is not the same as a shear rate. So something is wrong. A few lines later the authors write Δ p = mu tau_loc, so they really mean tau_loc is a shear rate. Nevertheless, the authors then write "It is therefore related to the network's architecture as pressure senses the entire network's morphology" but this is incorrect since (as their equation indicated) tau_loc only knows about the local pressure drop, which is distinct from the actual values of the pressure (I believe all the arguments in the paper assume very reasonably that the liquid is incompressible).

We apologize for our typo referring to shear stress when *τ_s_*, indeed, is simply the shear *rate* sensed by the vein after a time delay. We have clarified this quantity.

In addition, thanks to our bottom-up derivation based on force balance at the tube wall, we find that *τ*_loc_ (now *τ*_0_ in the manuscript) is, in fact, proportional to *P* − *P*_0_, *i.e.*, proportional to the difference between the local pressure, *P*, and the pressure *outside* the vein, *P*_0_, *i.e.* the atmospheric pressure. *P*, indeed, is a function of the entire network architecture. In fact, as the pressure, *P*, is obtained from a matrix inversion of Kirchhoff’s laws at every node, it integrates the entire network architecture via fluid flow.

We have clarified the notations (removing ∆*P*, which was confusing) and added this explanation in the main text.

(7) The authors continue with the confusion of shear stress and shear rate in equation (2): "the shear sensed by a vein wall taus and fluid shear rate " – I have no idea what this could mean. The fluid shear rate in the vessel is tau so what do they mean when they write "the shear sensed by a vein wall"? Note: I think the confusion is that the authors wrote "shear stress" above but they really mean "shear rate"?

The referee is correct that we meant shear rate for *τ_s_* and not shear stress. The letter *τ* is now used consistently to denote a shear rate. We have added this distinction in the main text.

(8) Then when I read some of this the authors also write (P. 3, column 1) "In the force balance perspective, pressure, circumferential stress and active stresses are typically constant during an adaptation event [8] and vary smoothly throughout the network (Appendix Figure A.2). In contrast, shear stress (proportional to shear rate) is observed to change significantly over time and space throughout the network, and is therefore expected to contribute significantly to long time scale dynamics" but this seems inconsistent with their statement in the next column that Δ p = mu tau_loc.

We are sorry to realize that our choice of variable name here caused confusion. In fact, *τ*_loc_ is not the local shear rate. Instead, it is the (local) *target* shear rate, or steady state value, for a specific vein. With the subscript loc we meant to indicate that this quantity is a local property that may smoothly depend on space and time. In contrast, h*τ*i is the actual shear rate in the vein. Again, if the vein were not evolving in time, we would observe that h*τ*i plateaus to a value that is *τ*_loc_. But since the network, here, never reaches steady-state, h*τ*i can vary broadly while *τ*_loc_ does not as much. Our choice of notation was confusing since *τ*_loc_ could be read as “local shear rate” which it is not; we now renamed this quantity *τ*_0_ in the revised manuscript. We now clarify, particularly in Sec. III-B, which quantities vary smoothly across the network and why. We also provide a table (Table I) to recapitulate the variable names, signification, and dependence on space and time.

Of course, just after they write "Thus, tau_loc, which also includes a contribution from active stress generation," – so I think the big jumble here is that the authors are not helping a reader (or themselves) unravel rates from stresses, but surely this is important since the cell sensing mechanisms may be different for a rate versus a stress.

We apologize again for this initial lack of clarity. We have now clarified and streamlined the text to only use shear rates. In particular, this statement now reads,

“*τ*_0_ is related to the local fluid pressure *P*, as *τ*_0_ ' *τ*_active_ − h*P* − *P*_0_i*/µ* (see short derivation in Appendix B 1). Here *P* − *P*_0_ characterizes the pressure imbalance between the fluid pressure inside the vein, *P*, and the pressure outside, *P*_0_, namely the atmospheric pressure. We recall that *µ* is the fluid viscosity. Finally, *τ*_active_ = *σ*_active_*/µ* is a shear rate related to the active stress *σ*_active_ generated by the acto-myosin cortex [57,58] to sustain the contractile activity of the vein, and is, therefore, an indirect measure of the metabolic or energetic consumption in the vein.”

Note that experimentally velocity and radius are well resolved in the organism, allowing us to infer shear rates from the data. However, the fluid’s viscosity may slightly depend on the veins position, and hence is not as resolved as other parameters [18]. Hence we focus here on shear rates rather than shear stresses which could contain less resolved data.

(9) Figure 2B: monotonous → monotonic.

We have modified monotonous to monotonic.

(10) What do all the colors correspond to the panels in Figure 2B? The colors seem to be defined in the main text but they should also be identified in the figure caption.

We provide an indication of the color code for Figure 2B (now Figure 3B) in the caption and other captions where suited.

(11) p. 4: "non-monotonous shear rate decrease (light pink Figure 2-B.i). Without the time delay (3), instantaneous shear sensing as …  (similarly as in e.g. Ref. [27]) can produce neither circling nor non-monotonous …" – so have previous authors highlighted some of these points (ref 27 is from 1996)? On p. 5, the authors similarly note results already identified in ref. 27. The authors are unclear about what might have already been demonstrated in advance of their work. Moreover, in several places, they write "Our force balance model challenges previous concepts showing that vein fate is not only determined through shear magnitude but also through network architecture dependent parameters." But again this seems to be disingenuous to some other authors (e.g. ref. 27 if I understand the paper) who have already made this point?

We thank the referee for highlighting that our placement of references to earlier work was confusing. To clarify, Ref. [1] is a theoretical study that investigates the previous phenomenological model resembling Equation (1), yet without any time delay, and examines the stability of two parallel veins in a network where those two veins are connected to a pressure and flow source.

We go beyond this theoretical work by (1) our experimental approach but also within the model itself as we (2) investigate two parallel veins embedded in a full network, with many other connected veins represented by an equivalent network resistance in our model *R*_net_. Finally, we (3) investigate numerous network configurations beyond two parallel veins like dangling ends and loops and (4) cascading dynamics of vein fate.

We now have clarified what previous models did and showed at each possible instance, and what we add here.

(12) "Overall, we find a remarkable agreement between t and data (Figure 2-D…" – but there is no figure 2d?)

We have updated the reference.

(13) The sensitivity of my eyes to color is not good enough to make sense of Figure 3.

To make more sense of Figure 3 (now Figure 4), we have updated the scale of the small veins to enhance the contrast. Additionally, we have removed *Q*_net_ from the figure to gain focus on other more relevant parameters such as the relative resistance *R/R*_net_ and the pressure *P*. Finally, we have now detailed in the caption and in the text what to look for and what to pick up on these graphs. We include in particular arrows to characteristic veins to focus on and comment on their behavior in the caption.

14) p. 4-5: I do not understand why Q_net, which is dimensional should be, on its own, a relevant parameter.

We apologize that in wanting to expose all possible relevant quantities, we did so at the expense of clarity and conciseness.

As *Q*_net_ is involved in Equation (3), it was a priori relevant to investigate how it might change the shear rate *τ*. Yet since *Q*_net_ is simply a measure of vein size, as Q_net_=Q≃a^2^ϵL/T

(where *a* is the vein radius, ϵ≃0.1 the contraction amplitude ratio, *L* the length of the vein and *T* the period of the contractions), it is not a trigger for adaptation.

We have removed *Q*_net_ from this figure. We keep it in the Appendix A in Figure A.6-F if some readers are interested in knowing the typical flow magnitudes observed here, and briefly comment on the graph.

Reviewer #2 (Recommendations for the authors):I enjoyed reading this manuscript, which describes some impressive experiments and analysis, together with what seems to be some very original ideas about how vascular adaptation can work in P. polycepharum. The paper is rather dense, though, and although this is of course a question of style, I personally would think the many interesting results might more easily be appreciated by a broader audience if they were packaged as at least two separate papers – in particular, the detailed force-balance model (in Appendix A) based on the unique property of actin networks is very interesting and would seem to stand alone as a paper on its own, while its main role in the main text is to furnish one of the three equations (Equation 1) needed for the analysis of experimental data. I could conceive that most of the story of the main text could then be told in a paper that doesn't go into the details of Appendix A but provides Equation 1 as a phenomenological equation that satisfies Murray's law. In any case, the authors have made the choice to package it in this way, and so I will try to provide constructive suggestions below to improve readability/accessibility to the broad readership of eLife.

We thank the referee for this insight into our work and their feedback on the presentation of our manuscript.

Indeed, our force balance approach (leading to Equation (1)) was originally built as a support for our experimental work – arising out of our frustration of the broadly used but only *phenomenologically* motivated law [1, 6, 7, 8, 9, 10, 11, 12, 13, 14]. For the sake of accuracy, the derivation ended up being a significant amount of work and material, becoming the rather dense (now former) Appendix A. Based on further thought on the presentation, on the convincing arguments of the referee, and for the sake of pedagogy and accessibility of the present manuscript, we have now prepared Appendix A as an accompanying manuscript that we are ready to submit if granted unanimous approval from all reviewers.

Overall, I have confidence in the authors' technical abilities (both experimental and theoretical), but in many places, the descriptions were less than transparent (and in some cases inaccurate; see below).

We are sorry to realize that the dense content of our manuscript kept us from rising also to our standards of meticulously consistent and clear writing. We have significantly modified the paper and believe our efforts have made descriptions and presentation more transparent. This included, for example, reworking figures, titles and text to decrease the number of variables used and adding a table (Table I) to highlight the main variables. We also have split apart Appendix A (in a separate manuscript to be approved by the reviewers) in an effort to increase readability.

Apart from those issues of presentation, I have identified only two major concerns – one regarding model fits the data, and another regarding the interpretation of the observed time delay – which I believe need to be addressed before publication:1. The description of the model fits gives the impression that the adaptation timescale (t_adapt) and the target shear rate (tau_loc) were allowed to vary freely when fitting experimental data. But according to the force-balance model of Appendix A, the two are not independent, but rather are related in a specific way (via Equation 5 of main text). So allowing these two parameters to vary freely seems to implicitly assume that the other parameters appearing in Equation 5 (specifically, viscosity mu and line-friction coefficient γ) are changing across data sets. I am not familiar enough with the system to know whether this is a reasonable assumption, but I was under the impression for the Appendix A text that mu and γ were both assumed constant everywhere, and in any case, some discussion of this point (i.e. what is happening with the effectively assumed differences in mu and γ across these fits) seems necessary.

We thank the referee for raising this highly relevant question. Before answering, let us first recall the main issue raised using updated notations and carefully revised equations. There was, sadly, a typo in the submitted manuscript, as detailed below, which we have now corrected. We are discussing the adaptation rule,

d⟨a⟩dt=⟨a⟩tadapt(Ts2T02−1).

(3)

obtained from force balance, and where *t*_adapt_ = tadapt=γ2πμTloc and T0=TcTloc. Note, we, here, made changes of notation *τ*_loc_ → *τ*_0_ and ˜*τ*_loc_ → *τ*_loc_. The typo in the previous manuscript consisted of calling the wrong *τ* variable when writing *t*_adapt =_
=γ2πμTloc, see now corrected notation above. In these equations, *γ* corresponds to a friction coefficient characterizing the resistance of the cortex material to plastic deformation, *µ* the inner fluid’s viscosity, *τ_c_* a shear rate characterizing the anisotropic response of the material to shear, and *τ*_loc_ the shear rate target value for the vein, that is a measure of both metabolic consumption for actin cortex contractility and local pressure, up to viscosity prefactor, in the network.

To put things in context again, the force balance derivation shows that *τ*_loc_ encapsulates the local pressure in the network and varies smoothly and slowly in time and throughout the network. This insight led us to fit different trajectories with a priori different values of *τ*_0_, in time and throughout a network, since *τ*_0_ is related to *τ*_loc_, as *τ*_0_ = *τ_c_τ*_loc_. Similarly, we fit different trajectories with different values of *t*_adapt_ since *t*_adapt_ is also related to *τ*_loc_, *t*_adapt_ = _2*πµτ*_*^γ^* loc. The question raised by the referee is that both quantities, *τ*_0_ and *t*_adapt_, within the scope of the equations resulting from our force balance derivation, are related by three parameters: *γ*, *µ*, and *τ_c_*, that at first sight appear to be organism-specific and constant, questioning if *τ*_0_ and *t*_adapt_ are independent parameters.

To now answer the question. Indeed *γ*, *µ* and *τ_c_* do depend on mechanical and fluidic properties that vary across an individual organism as a function of both vein maturation and size [19, 18, 20, 21], as well as integrated exposure to light [22], as well as among different specimen due to the responsiveness to ambient conditions, such as humidity [23, 24], light conditions [23, 25, 26, 27] and temperature [28, 29]. For example, the cytoplasm viscosity *µ* can vary depending on local content of salt concentration or dispersed particles [18]. Furthermore, both *γ* and *τ_c_* are related to the cortex mechanical properties, whose structure varies both within a specimen and over time [20, 21]. Thus, variations both among and within an organism are expected in all three parameters. This validates that *τ*_0_ and *t*_adapt_ are independent parameters. In the revised manuscript, we added important context regarding the expected variability of the mechanical and fluidic parameters. To further support the significance of the time delay, we now perform data fits without a time delay, resulting in a prediction error that is systematically larger than when fitting with the time delay (see Table B.3).

2. In the analysis and interpretation of the time delay, it seems the sign of the time delay was assumed to be positive (i.e. radius changes following shear-rate changes in time, rather than the other way around). For example, the cross-correlation analysis in Appendix D shows the cross-correlation only in the positive time direction. This assumption is also implicit in the statement (p.2) that "Clockwise circling corresponds to an in/decrease in shear followed by an in/decrease in vein radius pointing to a shear rate feedback on local vein adaptation." This is of course the expected direction of causality for shear-rate feedback, but the presented data and analyses in my view do not conclusively establish this direction of causality. This is because both the dynamics of the (time-averaged) shear rate and radius tend to be oscillatory, and with oscillatory signals, it is possible that the time delay associated with the cross-correlation peak can become positive or negative (and likewise oscillations in the phase plane can be CW or CCW), depending on the phase delay. So it does not seem so straightforward to establish the direction of causality based on the phase-plane rotation direction or the kinds of cross-correlation data presented in Appendix D. I do understand, however, that shear-rate feedback is a very natural expectation as a vein radius adaptation mechanism, especially given that Murray's law has already been established in the system. So perhaps a good choice could be to simply state something to that effect, and say that the direction of causality is an assumption, rather than a result of the analysis?

We thank the referee for highlighting that the causality link was not clearly established. We have now added further analysis based on our data to support the causality claim.

– Firstly, the observation of trajectories in the phase space h*a*i and h*τ*i allows us to establish a first claim on causality relations. Indeed, if circulatory trajectories are observed, clockwise trajectories in time (reciprocally anticlockwise) indicate that shear rate changes precede vein radius adaptation (reciprocally follow). This was not clearly stated, so we stress it now in the main text. Out of the 12 close-up veins investigated, 6

veins show stable clockwise feedback, 4 shrink and vanish, 1 shows stable anticlockwise feedback, and 1 is not classifiable (see Appendix Figure S2). For the specimen of Figure 1B, out of 200 randomly picked veins, 80 show stable clockwise feedback, 80 shrink and vanish, 20 show stable anticlockwise feedback, and 20 are not classifiable. From the dominant majority of stable clockwise trajectories, it appears to us reasonable to assume that shear rate changes precede vein radius adaptation. Hence, shear rate may cause vein radius adaptation. We initially included most of this discussion only in the methods, but we now see that it is key to understand causality and include it in the main text.

– Secondly, we now conduct the delay feedback statistical analysis on the full networks without assuming causality. We retain the positive *and* negative time delays that correspond to a maximum in the cross-correlation of the averaged shear rate h*τ*i(*t*+∆*t*) and average vein adaptation d⟨a⟩dt(t), if that maximum is significant, *i.e.* if the crosscorrelation is greater than 0*.*5. We obtain 15 times more veins with a relevant crosscorrelation with positive time delay than with a negative time delay. We show these results in the updated Figure B-2 in Appendix B.

– Finally, indeed, some trajectories are oscillating, say with period *T*_osc_, and hence it may not be obvious by cross-correlation for these specific trajectories to determine whether the delay is, say *t*_delay_ or −*T*_osc_ + *t*_delay_, or another superposition of both. *T*_osc_ is the approximate period of the oscillations in the long-time adaptation dynamics, and typically is *T*_osc_ = 20 min. In contrast, the apparent phase lag (between h*τ*i and h*a*i) is usually of the order of a few minutes in the samples where the delay can be inferred unambiguously (*t*_delay_ 1 − 5min). We may, thus, expect that the time delay is indeed *t*_delay_ 1 − 5min (and not −*T*_osc_ + *t*_delay_ which would be much longer). Additionally, most signals are actually not strictly oscillatory; for example, Figure 1 B iii clearly shows a lag time (between 7−15 min) that allows one to resolve the causality relation unambiguously.

We now make all these details appear either in the main text or in the appendices.

Apart from the above, my suggestions are mainly at the level of presentation, but I note that overall, the number of confusing or inaccurate statements was rather large, so the list below is not exhaustive and the authors should consider careful proofreading (perhaps with the aid of non-author colleagues) before finalizing the paper for publication.

We have significantly revised our manuscript, with the help of non-co-authors, with the constant goal of making a clearer and more accurate text.

p. 2, last paragraph of the left column: it is stated that the shear rate has twice the frequency of the radius. This statement is confusing because there are many nonlinear transformations of sinusoids that do not lead to frequency doubling. As I understand it, the frequency doubling is rather due to the fact that what is considered is the absolute (i.e. unsinged) shear rate, which is indeed a nonlinear transformation, but a very specific one. This means it is the absolute-flow-rate factor (|Q|) coming into the given expression for the shear rate, rather than the radius itself (which appears as a^3^) within that expression that is actually controlling the frequency doubling.

We apologize again for the confusion. The referee is correct in their understanding that the frequency doubling of h*τ*i is due to taking the absolute value of the flow rate |*Q*|. In addition, the oscillations of a≃a0(1=ϵcos⁡(wt)) contribute only at second order compared to the oscillations of *Q*≃.*.*cos(*ωt*), since *Q* oscillates around 0 while *a* oscillates around a nonzero value. We have made that statement more precise in the main text.

p.3, right column: where Figure D.3 is cited, that figure does not appear to provide relevant data for this context.

We have clarified that part of the text and the ensuing reference to the figure by simplifying it.

p.3, I found t_ave, appearing in Equation 3 as a time delay, rather confusing. In the Appendix, it is explained that this is a data average, but the data averaging procedure explained in Materials and Methods seems to use a window centered on the current time point, so it is unclear how data averaging imposes a delay. This is especially confusing because there are already two other timescales, t_delay and t_adapt, that could contribute to the delay. This should be clarified.

We apologize for the initial presentation with *t*_ave_, which was, in fact, incorrect, since, indeed, the averaging procedure is centered and, hence, does not impose a time delay. To correct the mistake, we have removed *t*_ave_, which leaves the model’s conclusions unaltered.

p.9 just under Equation (6): the term 'resistance' is used to describe R in Equation (6), but that equation and the expression given for R here imply that R here actually is conductance (i.e. inverse of resistance). I noticed this error also elsewhere, although in most places R does indeed correspond to resistance. Please check all instances where R appears.

We thank the referee for spotting this inconsistency, and we have carefully defined *R* as the flow resistance everywhere.

p.15, near the top of the page: t_delay is used twice, where correctly I believe what is being discussed is in fact t_adapt.

We have clarified the distinction between *t*_delay_ and *t*_adapt_ and took care of consistent usage.

p.18, under Equation B13: two expressions are given for λ_+. I guess this is a typo (the second one should be λ_-?).

Yes, it should read *λ*_−_; we have corrected it.

p.20-25: Appendix D is very dense, and is lacking a narrative text. Currently, many important statements/derivations occur within the legends of the figures provided within this section, for example, the expression for how Q is computed from the vein radius change I believe only appears in the legend of Figure D.1 and I could not find it anywhere else in the main text or the text narrative of the appendices. Consider including this in the main text.

We have restructured all appendices significantly to include narrative text and reflect the paper’s flow as it is currently set.

References

1. Hacking, W., VanBavel, E. and Spaan, J. Shear stress is not sufficient to control growth of vascular networks: a model study. American Journal of Physiology-Heart and Circulatory Physiology 270, H364–H375 (1996).

2. Janmey, P. A. et al. Negative normal stress in semiflexible biopolymer gels. Nature materials 6, 48–51 (2007).

3. Gardel, M. L., Kasza, K. E., Brangwynne, C. P., Liu, J. and Weitz, D. A. Mechanical response of cytoskeletal networks. Methods in cell biology 89, 487–519 (2008).

4. Kang, H. et al. Nonlinear elasticity of stiff filament networks: strain stiffening, negative normal stress, and filament alignment in fibrin gels. The Journal of Physical Chemistry B 113, 3799–3805 (2009).

5. Conti, E. and MacKintosh, F. C. Cross-linked networks of stiff filaments exhibit negative normal stress. Physical review letters 102, 088102 (2009).

6. Taber, L. A. A Model for Aortic Growth Based on Fluid Shear and Fiber Stresses. Journal of Biomechanical Engineering 120, 348–354 (1998). URL https://doi.org/10.1115/1.2798001.

7. Hu, D., Cai, D. and Rangan, A. V. Blood vessel adaptation with fluctuations in capillary flow distribution. PloS one 7, e45444 (2012).

8. Secomb, T. W., Alberding, J. P., Hsu, R., Dewhirst, M. W. and Pries, A. R. Angiogenesis: an adaptive dynamic biological patterning problem. PLoS computational biology 9 (2013).

9. Pries, A., Secomb, T. and Gaehtgens, P. Structural adaptation and stability of microvascular networks: theory and simulations. American Journal of Physiology-Heart and Circulatory Physiology 275, H349–H360 (1998).

10. Pries, A. R., Reglin, B. and Secomb, T. W. Remodeling of blood vessels: responses of diameter and wall thickness to hemodynamic and metabolic stimuli. Hypertension 46, 725–731 (2005).

11. Hu, D. and Cai, D. Adaptation and optimization of biological transport networks. Physical review letters 111, 138701 (2013).

12. Tero, A., Kobayashi, R. and Nakagaki, T. A mathematical model for adaptive transport network in path finding by true slime mold. Journal of theoretical biology 244, 553–564 (2007).

13. Akita, D. et al. Experimental models for murray’s law. Journal of Physics D: Applied Physics 50, 024001 (2016).

14. Baumgarten, W. and Hauser, M. J. Functional organization of the vascular network of physarum polycephalum. Physical biology 10, 026003 (2013).

15. Radszuweit, M., Alonso, S., Engel, H. and B¨ar, M. Intracellular mechanochemical waves in an active poroelastic model. Physical review letters 110, 138102 (2013).

16. Alonso, S., Radszuweit, M., Engel, H. and B¨ar, M. Mechanochemical pattern formation in simple models of active viscoelastic fluids and solids. Journal of Physics D: Applied Physics 50, 434004 (2017).

17. Alim, K., Amselem, G., Peaudecerf, F., Brenner, M. P. and Pringle, A. Random network peristalsis in Physarum polycephalum organizes fluid flows across an individual. Proc. Natl. Acad. Sci. U. S. A. (2013). arXiv:1408.1149.

18. Puchkov, E. Intracellular viscosity: Methods of measurement and role in metabolism. Biochemistry (Moscow) Supplement Series A: Membrane and Cell Biology 7, 270–279

(2013).

19. Swaminathan, R., Hoang, C. P. and Verkman, A. Photobleaching recovery and anisotropy decay of green fluorescent protein gfp-s65t in solution and cells: cytoplasmic viscosity probed by green fluorescent protein translational and rotational diffusion. Biophysical journal 72, 1900–1907 (1997).

20. Fessel, A., Oettmeier, C., Wechsler, K. and D¨obereiner, H.-G. Indentation analysis of active viscoelastic microplasmodia of P. polycephalum. Journal of Physics D: Applied Physics 51, 024005 (2017).

21. Lewis, O. L., Zhang, S., Guy, R. D. and Alamo, J. C. d. Coordination of contractility,´ adhesion and flow in migrating Physarum amoebae. Journal of The Royal Society Interface 12, 20141359 (2015).

22. B¨auerle, F. K. On mass transport in Physarum polycephalum submitted by (2019).

23. Rakoczy, L. The myxomycete physarum nudum as a model organism for photobiological studies. Berichte der Deutschen Botanischen Gesellschaft 86, 141–164 (1973).

24. Takahashi, K., Takamatsu, A., Hu, Z.-S. and Tsuchiya, Y. Asymmetry in the selfsustained oscillation ofphysarum plasmodial strands. Protoplasma 197, 132–135 (1997).

25. Hato, M., Ueda, T., Kurihara, K. and Kobatake, Y. Phototaxis in true slime mold physarum polycephalum. Cell Structure and function 1, 269–278 (1976).

26. Nakagaki, T., Umemura, S., Kakiuchi, Y. and Ueda, T. Action spectrum for sporulation and photoavoidance in the plasmodium of physarum polycephalum, as modified differentially by temperature and starvation. Photochemistry and photobiology 64, 859–862 (1996).

27. Rodiek, B. and Hauser, M. Migratory behaviour of physarum polycephalum microplasmodia. The European Physical Journal Special Topics 224, 1199–1214 (2015).

28. Wohlfarth-Bottermann, K. Oscillating contractions in protoplasmic strands of physarum: Simultaneous tensiometry of longitudinal and radial rhythms, periodicity analysis and temperature dependence. Journal of Experimental Biology 67, 49–59

(1977).

29. Hejnowicz, Z. and Wohlfarth-Bottermann, K. Propagated waves induced by gradients of physiological factors within plasmodia ofphysarum polycephalum. Planta 150, 144–152 (1980).